# The Hidden Cost of Modeling p(x): Vulnerability to Membership Inference Attacks in Generative Text Classifiers

## Abstract

Membership Inference Attacks (MIAs) pose a critical privacy threat by enabling adversaries to determine whether a specific sample was included in a model's training dataset. Despite extensive research on MIAs, systematic comparisons between generative and discriminative classifiers remain limited. This work addresses this gap by first providing theoretical motivation for why generative classifiers exhibit heightened susceptibility to MIAs, then validating these insights through comprehensive empirical evaluation. Our study encompasses discriminative, generative, and pseudo-generative text classifiers across varying training data volumes, evaluated on nine benchmark datasets. Employing a diverse array of MIA strategies, we consistently demonstrate that fully generative classifiers which explicitly model the joint likelihood $P(X, Y)$ are most vulnerable to membership leakage. Furthermore, we observe that the canonical inference approach commonly used in generative classifiers significantly amplifies this privacy risk. These findings reveal a fundamental utility-privacy trade-off inherent in classifier design, underscoring the critical need for caution when deploying generative classifiers in privacy-sensitive applications. Our results motivate future research directions in developing privacy-preserving generative classifiers that can maintain utility while mitigating membership inference vulnerabilities [1].

## 1 Introduction

Text Classification (TC) is a fundamental task in Natural Language Processing (NLP), serving as the backbone for numerous applications including sentiment analysis, topic detection, intent classification, and document categorization (Yogatama et al., 2017; Castagnos et al., 2022; Roychowdhury et al., 2024; Kasa et al., 2024; Pattisapu et al., 2025). As machine learning models have become increasingly sophisticated and widely deployed, concerns about their privacy implications have grown substantially. One of the most critical privacy vulnerabilities is the **Membership Inference Attack** (MIA), where an adversary attempts to determine whether a specific data point was included in a model's training set (Shokri et al., 2017). MIAs represent a fundamental threat to data privacy by exploiting differential model behaviors on training versus non-training data to infer membership in the training set (Shokri et al., 2017; Carlini et al., 2019; Shejwalkar et al., 2021; Song et al., 2022; Song & Mittal, 2021; Feng et al., 2025). The implications are particularly severe for sensitive personal data, potentially violating privacy expectations and regulatory requirements. Recent surveys have highlighted the growing sophistication of these attacks (Amit et al., 2024; Feng et al., 2025).

The majority of MIA research in TC has concentrated on discriminative models like BERT (Devlin et al., 2019), which directly model $P(Y|X)$ and learn decision boundaries without explicitly modeling data distributions (Zheng et al., 2023; Kasa et al., 2025). Studies have revealed how factors such as overfitting, model capacity, and training data size influence attack success rates (Amit et al., 2024). Despite this discriminative focus, there has been renewed interest in generative classifiers for TC (Li et al., 2025). Unlike discriminative models, generative classifiers explicitly model the joint distribution $P(X, Y) = P(X|Y)P(Y)$, offering compelling advantages such as: superior performance in low-data regimes (Kasa et al., 2025; Yogatama et al., 2017), reduced susceptibil-

---

[1]Code available at https://anonymous.4open.science/r/privacy-attacks-gendisc-classifiers-143E

ity to spurious correlations (Li et al., 2025), and principled uncertainty estimates via Bayes' rule (Bouguila, 2011). The renaissance of generative classifiers in TC has been particularly bolstered through scalable model architectures including autoregressive models (Radford et al., 2018), discrete diffusion models (Lou et al., 2024), and generative masked language models (Wang & Cho, 2019b).

However, the very characteristics that make generative classifiers attractive explicit modeling of data distributions and superior performance with limited data raise important privacy questions. While MIAs have been extensively studied for discriminative models, a significant gap exists in understanding how different classification paradigms compare in their vulnerability to such attacks. In this work, we present the first large-scale, systematic analysis of the vulnerability of transformer-based text classifiers to MIAs across a spectrum of modeling paradigms. Following Kasa et al. (2025), we consider three broad categories: (1) **discriminative models** such as encoder-style models `DISC`, which model the conditional distribution $P(Y|X)$; (2) **fully generative models** that explicitly model $P(X,Y)$, such as autoregressive (`AR`) or discrete diffusion models (`DIFF`); and (3) **pseudo-generative models**, such as Masked Language Models `MLM`, and pseudo-autoregressive `P-AR` models, where the label is appended at the end of the input sequence.

**Contributions.** To our knowledge, this work provides the first *systematic* study of membership inference risk for *generative* text classifiers, combining theory, controlled toy settings, and large-scale transformer-based experiments.

**(1) Theory under a single-shadow black-box framework.** We formalize membership inference for generative vs. discriminative classifiers in a black-box setting with either probabilities or logits exposed. Our bounds decompose the optimal attack advantage into leakage from the marginal $P(X)$ and the conditional $P(Y \mid X)$, clarify when *logits (joint scores)* can strictly dominate *probabilities (conditionals)*. Using a simulation setting with tunable dimension, sample size, and class separation, we show that *generative* classifiers leak more through *log-joint* scores than discriminative posteriors do—quantitatively aligning with our theoretical predictions about marginal vs. conditional channels.

**(2) First systematic analysis for *MIA in text classification*.** Across multiple datasets and five model paradigms (discriminative, fully generative: AR and discrete diffusion, and pseudo-generative), we provide a head-to-head evaluation of MIA vulnerability under matched training protocols. We isolate the effects of (i) architectural factorization (`AR` vs. `p-AR`), (ii) *output interface* (logits vs. probabilities), and (iii) data size. We find that fully generative models are consistently more vulnerable, with the strongest leakage observed when logits from $K$-pass scoring are exposed.

**(3) Empirical analysis and practical guidance.** We show that different architectures yield distinct privacy–utility trade-offs, with generative models offering better low-sample accuracy and robustness benefits at the cost of higher leakage, while pseudo-generative models emerge as more privacy-conscious alternatives at higher data regmies. Building on these results, we provide actionable guidance on API exposure (favoring probabilities over logits), model choice, and training practices for privacy-sensitive deployments.

## 2 RELATED WORK AND BACKGROUND

**Generative vs. discriminative classifiers.** Classic analyses compare generative and discriminative learning on efficiency and asymptotics: discriminative models achieve lower asymptotic error, while generative models converge faster in low-data regimes (Efron, 1975; Ng & Jordan, 2001; Liang & Jordan, 2008). In text classification, recent work has renewed interest in generative classifiers that model $P(X,Y) = P(X \mid Y)P(Y)$, reporting advantages in calibration, uncertainty estimation, robustness to spurious correlations, and performance under limited data (Yogatama et al., 2017; Zheng et al., 2023; Li et al., 2025; Kasa et al., 2025). Modern instantiations of generative classifiers in TC include autoregressive (AR) label-prefix classifiers (Radford et al., 2018), discrete diffusion models (Lou et al., 2024), and generative uses of masked LMs (Wang & Cho, 2019b). A practical drawback is that fully generative label-prefix AR classifiers typically require $K$-*pass* inference—one forward pass per label $y_i$ to score $\log P(x, y_i)$—whereas discriminative models compute $P(Y \mid X)$ in a single pass; conversely, the generative formulation naturally supports Bayes-rule posteriors and principled uncertainty quantification via the decomposition $P(Y \mid X) \propto P(X \mid Y)P(Y)$ (Bouguila, 2011). We investigate the generative text classifiers dis-

cussed in Li et al. (2025); Kasa et al. (2025) compare them with the well studied BERT-style encoder classifiers in this work.

**Membership inference background.** MIAs exploit differences in a model's behavior on train vs. non-train points. Shokri et al. (2017) introduced the multi–shadow-model paradigm for training an attack classifier on output vectors. Salem et al. (2018b) showed this can be simplified to *single*-shadow or even *no*-shadow attacks using confidence/loss statistics, and we *adopt the single-shadow assumption* in our theoretical setup by modeling a proxy $Q$ alongside the target $P$ and reasoning about induced score laws $(P_S, Q_S)$. Yeom et al. (2018) established that overfitting is not the sole driver of MIAs: they connect attack advantage to generalization error via a loss-threshold attack and show that *influence* of individual examples can cause leakage even when generalization error is small. Complementary systematization in ML-as-a-Service highlights how API exposure (labels/top-$k$/probabilities), shadow alignment, and data mismatch shape attack efficacy (Truex et al., 2018).

**Scope and assumptions.** We study *black-box* adversaries that query the classifier and observe either probabilities or pre-softmax logits (when available); *white-box* access to parameters/gradients is out of scope. Also prior works (Sablayrolles et al., 2019; Salem et al., 2018a; Huang et al., 2024) have shown that white-box access offers limited additional advantage both theoretically and empirically. For fully generative label-prefix models, we assume $K$-pass inference is the canonical deployment mode; we analyze both logit- and probability-based attack surfaces and relate them to joint vs. conditional scoring used later in our theory. See Appendix A for an expanded survey, additional NLP-specific MIAs, and a detailed taxonomy of threat models.

## 3 MOTIVATION

Before discussing MIA attacks on on benchmark datasets, we first develop a theoretical account of how membership vulnerability manifests in generative classifiers, identify factors that exacerbate leakage (e.g., marginal memorization and weak conditional generalization), and formally compare what is revealed by joint vs. conditional exposures. We then instantiate these results in a controlled toy setting with a known data-generating process, showing that the empirical behavior of standard attacks mirrors the theoretical predictions.

### 3.1 PRELIMINARIES AND NOTATION

Let $\Omega$ denote the universe of all datapoints, where each datapoint $z \in \Omega$ can be decomposed into a feature–label pair $(x, y)$ with $x \in \mathcal{X}$ (features) and $y \in \mathcal{Y}$ (labels). We consider two generative classifiers: $P$: the *target model*, which induces a joint probability distribution $P(X, Y)$ and $Q$: the *shadow model*, trained independently on population data (Salem et al., 2018b), which induces its own probability distribution $Q(X, Y)$ which the attacker uses to determine sample membership. We are interested in quantifying the difference between $P$ and $Q$ in terms of their induced joint distributions over $(X, Y)$, which captures susceptibility to MIA. Let an *attack signal* be any measurable function $S = S(\hat{p}(X, Y))$ of the model output (e.g., logits $(\log \hat{p}(x, y'))_{y' \in \mathcal{Y}}$, probabilities $\hat{p}(\cdot \mid x) = \mathrm{softmax}(\log \hat{p}(x, \cdot))$, or a scalar score $\log \hat{p}(x, y_i)$) which is exposed to the client and the attacker tries to come up with an optimal decision $\varphi$ rule based on the signal $S$ to determine the membership. Given any attack signal $S$, let $P_S := \mathcal{L}(S \mid P)$ and $Q_S := \mathcal{L}(S \mid Q)$ denote the pushforward laws under the target/shadow distributions. Intuitively, these are the score distributions the attacker tunes their threshold on: in our empirical evaluation (cf. §5), the standard MIA AUROC is measured by sweeping a decision threshold that gives full weight to members under $P$ versus non-members under $Q$. For any (possibly randomized) decision rule $\varphi : \mathrm{range}(S) \to [0, 1]$, the achieved membership advantage $\mathrm{Adv}_\varphi(S) := \mathbb{E}_P[\varphi(S)] - \mathbb{E}_Q[\varphi(S)]$ is always upper-bounded by the total-variation distance between the pushforwards, and the latter cannot exceed the TV between the original joint distributions,

$$\mathrm{Adv}_\varphi(S) \leq \mathrm{TV}(P_S, Q_S) \leq \mathrm{TV}(P, Q) = \sup_{A \in \mathcal{F}} |P(A) - Q(A)| = \frac{1}{2} \int_\Omega |p(\omega) - q(\omega)| \, d\mu(\omega)$$

where we assume $P$ and $Q$ are defined on the same measurable space $(\Omega, \mathcal{F})$ and $p$ and $q$ denote densities of $P$ and $Q$ with respect to a common dominating measure $\mu$ (this is done for ease of mathematical exposition).

It is well known that $\mathrm{TV}(P, Q)$ equals the maximum distinguishing advantage of any binary hypothesis test between $P$ and $Q$, and in the first result, we show that in the case of generative classifiers, this can be cleanly bounded using a generative and discriminative component.

**Lemma 3.1** (Two-way decomposition: upper and lower bounds). *For the score-optimal attacker observing the feature-label pair $(X, Y)$ (equivalently, any sufficient statistic), the optimal advantage equals $\mathrm{TV}(P_{XY}, Q_{XY})$ and satisfies*

$$\left| \mathrm{TV}(P_X, Q_X) - \mathbb{E}_{x \sim P_X} \mathrm{TV}(P_{Y|X=x}, Q_{Y|X=x}) \right| \leq \mathrm{TV}(P_{XY}, Q_{XY}) \leq \quad (3.1)$$
$$\mathrm{TV}(P_X, Q_X) + \mathbb{E}_{x \sim P_X} \mathrm{TV}(P_{Y|X=x}, Q_{Y|X=x}).$$

*By Pinsker, any observable signal $S$ obeys*

$$\mathrm{Adv}(S) \leq \mathrm{TV}(P_{XY}, Q_{XY}) \leq \sqrt{\tfrac{1}{2} \mathrm{KL}(P_X \| Q_X)} + \sqrt{\tfrac{1}{2} \mathbb{E}_{x \sim P_X} \mathrm{KL}(P_{Y|X=x} \| Q_{Y|X=x})}.$$

*Discussion.* Lemma 3.1 cleanly separates membership leakage of a generative classifier into a marginal term $\mathrm{KL}_X$ (learning $P(X)$) and a conditional term $\mathrm{KL}_{Y|X}$ (learning $P(Y \mid X)$), matching the spirit of the bound already introduced in §3.1 (Theorem 1). This makes precise why modeling $P(X)$ can increase MIA risk. (See App. B for the proof and for a KL-formulation mirroring §3.1.)

**Lemma 3.2** (Joint $\succeq$ Conditional under full-vector exposure). *Let the model expose the per-class joint score vector $S_{\mathrm{joint}}(x) = \big( \log \hat{p}(x, y) \big)_{y \in \mathcal{Y}}$ and the conditional score vector $S_{\mathrm{cond}}(x) = \big( \hat{p}(y \mid x) \big)_{y \in \mathcal{Y}} = \mathrm{softmax}\big( S_{\mathrm{joint}}(x) \big)$. Then for any membership game,*

$$\mathrm{Adv}\big( S_{\mathrm{joint}} \big) \geq \mathrm{Adv}\big( S_{\mathrm{cond}} \big),$$

*with equality iff the per-$x$ additive normalizer $\log \hat{p}(X)$ is $P$-a.s. equal under $P$ and $Q$ (i.e., it carries no marginal signal about membership).*

*Discussion.* Lemma 3.2 says that when logits proportional to $\log \hat{p}(x, y)$ are exposed, passing to posteriors *cannot* increase advantage (data-processing). Intuitively, softmax removes the shared $-\log \hat{p}(x)$ term and therefore discards whatever membership signal is present in $P(X)$.

**Theorem 3.3** (Scalar joint can dominate conditional under systematic marginal skew). *Consider binary classification. Suppose the attacker receives either (i) a scalar joint score $S_{\mathrm{joint}}^{\mathrm{scal}}(X, Y) := \log \hat{p}(X, Y)$ or (ii) a conditional score $S_{\mathrm{cond}}(X, Y) := \hat{p}(Y \mid X)$. Assume the member vs. non-member conditionals satisfy the bounded likelihood-ratio condition: there exist constants $0 < \alpha \leq \beta < \infty$ such that for $P_X$-a.e. $x$ and both labels $y$,*

$$\alpha \leq \frac{P(y \mid x)}{Q(y \mid x)} \leq \beta.$$

*Then there exists $c = c(\alpha, \beta) \in (0, 1]$ such that*

$$\mathrm{Adv}\big( S_{\mathrm{joint}}^{\mathrm{scal}} \big) > \mathrm{Adv}\big( S_{\mathrm{cond}} \big) \quad \text{whenever} \quad c \, \mathrm{KL}_X > \mathrm{KL}_{Y|X}.$$

*An explicit choice is $c(\alpha, \beta) = \dfrac{\log \beta - \log \alpha}{1 + \log \beta - \log \alpha}$.*

*Discussion.* Theorem 3.3 addresses the practically important case where only a *single* generative score is exposed (e.g., log-likelihood for the observed label, or a label-agnostic scalar derived from the joint). Unlike Lemma 3.2, scalar joint and conditional are *not* deterministic transforms of each other; nonetheless, whenever the *marginal* skew $\mathrm{KL}_X$ dominates the *conditional* skew $\mathrm{KL}_{Y|X}$ ("systematic marginal skew") and conditionals are not wildly different between $P$ and $Q$, the scalar joint channel is provably more susceptible.

**Implications.** (i) Exposing logits of a generative model (full vector) is always at least as risky as exposing posteriors. (ii) Even if only a single generative score is exposed, sufficiently strong marginal memorization makes the generative channel strictly more vulnerable than conditional outputs. (iii) The decomposition in Lemma 3.1 explains our empirical hierarchy: models that *must* learn $P(X)$ (fully generative) leak through the marginal term in addition to the conditional term, inflating MIA advantage. (iv) Our framework is fully general, not limited to text classification. The decompositions in Lemma 3.1 and the dominance results in Lemma 3.2 & Theorem 3.3 apply to any generative–discriminative classifier pair because they rely only on model-induced score distributions, independent of modality or architecture and therefore extend to images, audio, tabular data, or any supervised domain.

## 3.2 Toy Illustration: Controlled Analysis of MIA Vulnerability

To validate our theoretical insights on the heightened vulnerability of generative classifiers, we conduct a controlled synthetic experiment that teases out key factors behind membership inference such as accuracy, signal/noise ratio, dimensionality, etc. Following Li et al. (2025), we use a toy setup of linear classifiers on linearly separable data, which strips away confounders, letting us directly study how marginal vs. conditional learning drives leakage before moving to complex real-world models.

**Experimental Setup.** We design a synthetic binary classification task where each input $x \in \mathbb{R}^d$ consists of two components: $x = [x_{\text{core}}, x_{\text{noise}}]$. The core feature $x_{\text{core}} \sim \mathcal{N}(y \cdot \mu, \sigma^2)$ correlates directly with the binary label $y \in \{-1, +1\}$, where $\mu$ controls class separation. The remaining $d-1$ coordinates are independent standard Gaussian noise. We systematically vary key parameters: dimensionality $d \in \{16, 64, 256\}$, training size $n_{\text{train}} \in \{50, 200\}$, class separation $\mu \in \{0.05, 0.10, \ldots, 0.50\}$, and class balance $w \in \{0.1, 0.3, 0.5\}$. We compare the discriminative Logistic Regression (LR) with the generative Linear Discriminant Analysis (LDA), evaluating three membership inference scores: max-probability for both models, and log-joint likelihood for LDA.

**Notation (attack scores).** LR/prob denotes the *max-probability* (confidence) score from LR, LDA/prob is the same max-probability score computed from LDA posteriors $\hat{p}_{\text{LDA}}(y \mid x)$; and LDA/log-joint is the LDA *log-joint* score. All three are label-agnostic membership scores.

$$s_{\text{prob}}(x) = \max_{y \in \{-1,+1\}} \hat{p}_{\text{LR}}(y \mid x), \qquad s_{\text{logjoint}}(x) = \max_y \{\log P(y) + \log \mathcal{N}(x \mid \mu_y, \Sigma)\}.$$

**MIA evaluation (AUROC).** For a given score $s(\cdot)$ and trained model, we compute scores on training samples (members) and on an i.i.d. test set (non-members). Treating members as positives and non-members as negatives, we sweep a threshold on $s(\cdot)$ to obtain the ROC curve and report its area (AUROC), the standard practice for membership inference. We aggregate results over 5 random seeds and plot mean curves with shaded std. deviation bands. Here we present the plots and analysis for the balanced case of $w = 0.5$. For the imbalanced cases, the same is deferred to Appendix G.

Figures 1 reveals several critical findings that support our theoretical predictions: (a) In the low-sample regime, **LDA is markedly more sample-efficient than LR**: for $d \in \{16, 64\}$, the accuracy achieved by LR with $n_{\text{train}} = 200$ is already matched (or exceeded) by LDA with $n_{\text{train}} = 50$; this accuracy gap widens as $d$ increases (smaller $n/d$). (b) Comparing LDA's two scores, the **joint score (LDA/log-joint) consistently yields larger membership susceptibility** than the posterior max-probability (LDA/prob), with the gap growing as $d$ increases or $n/d$ decreases, underscoring the additional risk from exposing joint/likelihood values. (c) Comparing discriminative and generative posteriors, at small $\mu$ LR/probexhibits lower susceptibility than LDA/prob; as $\mu$ grows, LR/prob's susceptibility rises sharply with margin and can meet or exceed LDA/prob, whereas LDA/proboften flattens or slightly decreases while LDA/log-jointremains high—consistent with likelihood dominating at larger separations. Apart from a single benign regime (balanced $w = 0.5$, large $\mu$, low $d$), LDA/log-jointexceeds LR/prob in susceptibility. (d) Increasing dimensionality $d$ at fixed $n$ lowers accuracy and increases membership advantage; in parallel, the across-seed standard deviation of both accuracy and AUROC narrows, yielding more consistent (but worse) accuracy and stronger, more stable membership signals in high dimensions.

The superior sample-efficiency of LDA is in part tied to the parameteric assumptions of LDA being satisfied by the data on which it is being fit. In order to tease out this we introduce a misspecification specifically a Huber-$\epsilon$ contamination Huber (1992) during the data geeneration process. The detailed plots are given in Appendix G.5. We notice that contamination reverses the generative LDA's

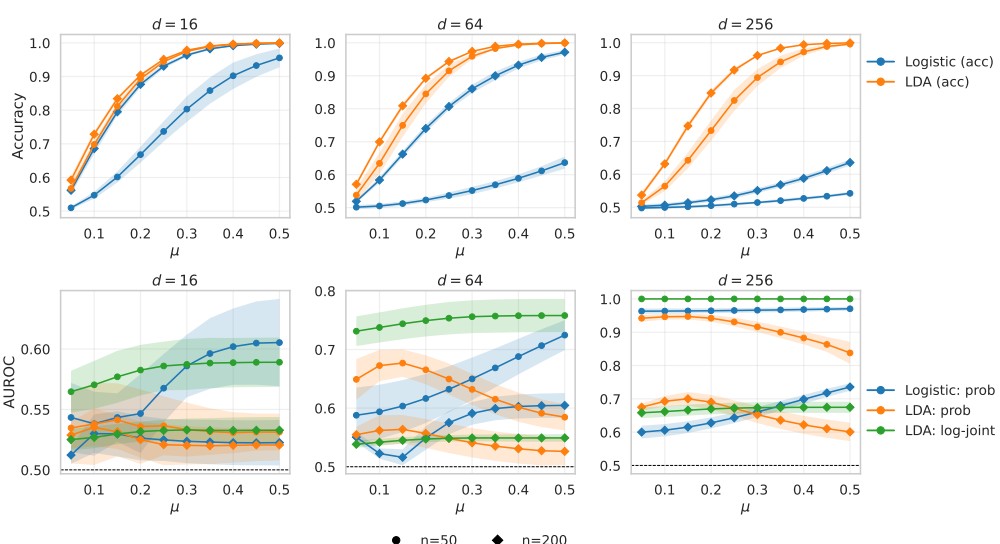

Figure 1: **Membership inference vulnerability increases with model confidence and dimensionality.** Top row: test accuracy vs. core separation $\mu$. Bottom row: membership inference advantage (AUROC) vs. $\mu$. Columns correspond to $d \in \{16, 64, 256\}$. Colors denote model types and inference methods: Logistic Regression max-probability (blue), LDA max-probability (orange), LDA log-joint (green). Markers indicate training size $n_{\text{train}} \in \{50, 200, 2000\}$. Results averaged over 5 seeds with $\pm 1.96 \times$ SEM bands.

clean-data sample-efficiency edge in accuracy — LR is typically better—because a few large-norm replacements strongly distort shared-covariance estimation even with shrinkage. However, exposing density scale remains risky: LDA/log-joint is the most susceptible membership score across most regimes we tested, particularly at high $d$ and small $n$. These controlled experiments provide concrete evidence that generative classifiers face fundamental privacy disadvantages, with the risk being particularly acute when exposing joint likelihood values or operating in high-dimensional, low-sample regimes.

## 4 EXPERIMENTAL SETUP

We evaluate privacy vulnerabilities in text classification by training multiple classifiers across datasets and subjecting them to diverse membership inference attacks (MIAs). Following Li et al. (2025); Kasa et al. (2025), we study three main classifier families:

**Discriminative** (`DISC/ENC`): Standard BERT-style encoders modeling $P(Y|X)$ using linear head on top of `[CLS]` token to directly map text $X$ to label $Y$. There's no explicit memorization signal in this modeling approach.

**Fully Generative:** Models that capture the joint distribution $P(X, Y)$ through:

(i) *Label-Prefix Autoregressive* (`AR`) models generate text $x$ conditioned on a label prefix (e.g., `Positive: The film was a masterpiece.`). Classification is performed via logits using likelihood estimation, $\arg\max_{l \in K} \log P(x, y_l)$, in a $K$-pass fashion ($K$ = number of labels). Such models may be more vulnerable to MIAs since logits expose information about $P(X)$. Alternatively, applying a softmax yields probabilities: $\text{softmax}(\log P(x, y_l)) = P(x, y_l)/P(x) = P(y_l|x)$, where the shared denominator $P(x)$ cancels across classes.

(ii) *Discrete Diffusion Models* (`DIFF`) are trained on $(X, Y)$ pairs with a denoising objective. Following Lou et al. (2024), noise gradually corrupts the input sequence to pure `[MASK]` tokens in the forward process, with original input reconstruction in the reverse process. At inference, the model predicts $y$ from `[MASK]`, conditional on $x$. We use *Diffusion Weighted Denoising Score Entropy*

*(DWDSE)* for logits, providing an upper bound on log-likelihood: $-\log p_0^\theta(x) \leq \mathcal{L}_{DWDSE}(x)$ under the ELBO.

**Pseudo-Generative:** This category represents a middle ground between discriminative and fully generative approaches. We explore using *Masked Language Models* (`MLM`) trained for reconstructing masked tokens bi-directionally rather than full causal modeling. These model the pseudo-joint likelihood rather than the true joint $P(X, Y)$ (Wang & Cho, 2019a).

All models utilize transformer-based architectures and are trained from scratch to avoid confounding effects from pre-training. Following Kasa et al. (2024), we evaluate three model size configurations: small (1 layer, 1 head), medium (6 layers, 6 heads), and large (12 layers, 12 heads). To enable fair comparison, we maintain comparable parameter counts across all architectures within each size configuration. Implementation details including model sizes and training hyperparameters are provided in Appendix B.

**Attack Methodology:** We examine two main classes of MIAs:(a) **Threshold-Based** attacks derive simple metrics from model outputs: (i) *Max Probability*: $\max(P(y|x))$, (ii) *Entropy*: $H(P(y|x)) = -\sum_i p_i \log p_i$, and (iii) *Log-Loss* using cross-entropy on the true label. (b) **Model-Based** attacks train an explicit attack model by querying the target classifier with member and non-member samples, representing each using the model's output probability or logits vector concatenated with ground-truth labels, and training a Gradient Boosting Model (`GBM-logits` / `GBM-probs`) to predict membership status. Detailed attack implementations are provided in Appendix B. Although there exists more sophisticated attacks (Shejwalkar et al., 2021; Song et al., 2022; Amit et al., 2024) (details in Appendix A.2), as will see in §5 that these basic attacks do a good job of revealing the differential vulnerability of generative and discriminative classifiers on TC.

**Dataset Details:** Our evaluation spans nine public text classification benchmarks : **AG News** Zhang et al. (2015), **Emotion** Saravia et al. (2018), **Stanford Sentiment Treebank (SST2 & SST5)** Socher et al. (2013), **Multiclass Sentiment Analysis**, **Twitter Financial News Sentiment**, **IMDb** Maas et al. (2011), and **Hate Speech Offensive** Davidson et al. (2017), covering diverse domains from sentiment analysis to topic classification. All models are trained from scratch using AdamW optimizer with early stopping to prevent overfitting, following Li et al. (2025) and Kasa et al. (2025). We measure attack success using Area Under the ROC Curve (**AUROC**), where 1.0 indicates perfect attack and 0.5 indicates random guessing. Dataset characteristics are provided in Appendix B.

## 5 RESULTS & DISCUSSIONS

Building on our theoretical analysis and synthetic experiments with LDA and LR (Section 3.2), we present comprehensive empirical evidence from real-world text classification scenarios. Our analysis examines: (1) privacy vulnerabilities across discriminative, fully generative, and pseudo-generative architectures, with patterns aligning with our controlled findings, (2) impact of model output representations (logits versus probabilities) on membership inference risk, and (3) how different approaches to modeling $P(X, Y)$ affect the privacy-utility trade-off. Through experiments on nine diverse datasets, we establish concrete relationships between architectural choices and privacy vulnerabilities, while identifying promising directions for privacy-preserving text classification.

Figure 2 shows that fully generative models (`DIFF`, `GEN`) are consistently more vulnerable to MIAs than discriminative (`DISC`) and pseudo-generative (`MLM`) models across five datasets. For clarity, only `GBM-logits` and `GBM-probs` are shown; other attacks follow the same trend (see Section F). Medium models behave similarly, while small `AR` are least susceptible, consistent with Kasa et al. (2025), who show these models behave nearly randomly. Full results across all datasets and model sizes are in Section F. Additionally, we also report True Positive Rate (TPR)@False Positive Rate (FPR)=0.1 (see Table 11) on a representative dataset (AG News) using a 12-layer model comparing `DISC` and `AR`. Consistent with AUROC findings, we again observe that `AR` exhibit higher susceptibility across all attack types, even under stricter low-FPR operating points.

These findings confirm our hypothesis: modeling $P(X, Y)$ forces generative models to capture both $P(Y|X)$ and $P(X)$, amplifying memorization risk compared to purely discriminative objectives.

Vulnerability does not vary monotonically with training size (Figure 2, Table 7), consistent with Amit et al. (2024). Early stopping dampens overfitting in low-data regimes, masking expected

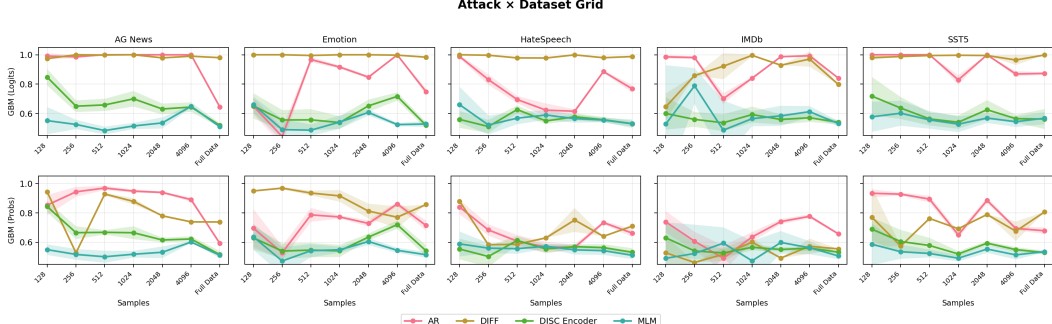

Figure 2: **[Best viewed in color]** MIA success rate (AUROC) compared across full-size model architectures with varying training dataset sizes. We evaluate fully generative classifiers (`AR`, `DIFF`), a discriminative classifier (`DISC`), and pseudo-generative models (`MLM`). The **top row** displays attack performance using model **logits**, while the **bottom row** shows results using output **probabilities**. Higher AUROC values indicate increased privacy vulnerability. Results averaged across 5 random seeds.

trends. Removing early stopping (training 20 epochs on AGNews) restores the expected pattern: susceptibility decreases with larger training sets (refer Table 8 in Appendix F).

We also the study how the MIA vulnerability changes with the representation of a class in the training sample in Appendix D and find that vulnerability difference between majority and minority classes (i.e. the classes with the highest and lowest representation in the training split) is high for `DISC`, `MLM` paradigms and it is relatively less pronounced for the generaive `AR`, `DIFF` paradigms.

**Logits as a High-Bandwidth Privacy Leakage Channel:** Our experiments show that membership inference attacks (MIA) using pre-softmax logits consistently outperform those based on post-softmax probabilities. As shown in Figure 2, logit-based attacks (top row) achieve higher AUC across all models and datasets than probability-based ones (bottom row). This aligns with prior work Shokri et al. (2017) and arises because logits preserve raw confidence scores, whereas softmax projects them onto a probability simplex, compressing information and reducing the attack surface.

The implications are significant: exposing logits through APIs even for calibration or temperature scaling greatly heightens privacy risk. Given that many ML APIs and frameworks expose logits by default Finlayson et al. (2024), practitioners should either restrict outputs to probabilities or add privacy-preserving safeguards when logits must be shared.

The success of membership inference attacks also depends on the attack strategy's sophistication and the adversary's access to auxiliary information. Table 1 reports results for both threshold-based and model-based attacks (refer Section 4), focusing on probability-based methods since many attacks are incompatible with logits. These results illustrate the attack efficacy hierarchy, i.e.

| Attack | DISC | GEN | MLM | DIFF |
|---|---|---|---|---|
| Max Probability | $0.56 \pm 0.05$ | $0.67 \pm 0.13$ | $0.55 \pm 0.06$ | $0.51 \pm 0.13$ |
| Entropy | $0.56 \pm 0.05$ | $0.63 \pm 0.12$ | $0.55 \pm 0.06$ | $0.60 \pm 0.09$ |
| Log-Loss | $0.60 \pm 0.06$ | $0.76 \pm 0.13$ | $0.55 \pm 0.08$ | $0.65 \pm 0.13$ |
| GBM-Probs | $\mathbf{0.62 \pm 0.08}$ | $\mathbf{0.81 \pm 0.13}$ | $\mathbf{0.56 \pm 0.07}$ | $\mathbf{0.76 \pm 0.16}$ |

Table 1: MIAs performance (AUROC) across different model architectures, averaged over all datasets for models with 12 layers trained on full data. Higher values indicate greater privacy vulnerability, with the highest values in each column shown in **bold**.

threshold-based attacks relying on output probabilities (*Max Probability*, *Entropy*) yield modest success, while incorporating ground-truth labels via *Log-Loss* improves performance. The most effective attack, a Gradient Boosting Model (*GBM*) trained on probability vectors and label information, notably excels for `AR` and `DIFF` models. We also find that model size exacerbates the privacy vulnerability in generative classifiers (refer to Appendix E), similar to previous findings on `DISC` (Amit et al., 2024). These findings underscore the urgent need for privacy defenses that remain effective across diverse adversarial capabilities and information access levels.

**The Impact of Factorization: Decomposing Leakage in** $P(X, Y)$**:** In (Kasa et al., 2025), the authors argue that fully generative models perform best in low-data regimes and should be preferred over discriminative models. However, our earlier results reveal that `AR` models exhibit signifi-

cantly higher vulnerability to MIAs compared to `DISC`. To address this, we investigate an alternative modeling paradigm that reduces MIA risk without sacrificing classification performance. *Pseudo-Autoregressive* (`P-AR`) models tackle this challenge by appending the label at the end of the input sequence, instead of modeling $P(X|Y)$ by pre-pending the label token,. Although this approach does not strictly capture $P(X|Y)$, recent work (Li et al., 2025) shows that label-appending often achieves better in-distribution accuracy than label-prepending. At inference, we can either use a K-pass run like `AR` to score each label and take the argmax, or a 1-pass run by selecting the predicted label from the final token's distribution (this is the canonical approach for `P-AR`). As evident from Table 2 (averaged across datasets) `P-AR` poses much lesser MIA risk compared to `AR`. However, `P-AR-kpass` exhibits similar vulnerability again similar to fully generative case. A few attacks are not stated here as they are qualitatively similar to Log-Loss.

To explain this phenomenon, we next examine how different factorizations of the joint distribution $P(X,Y)$ influence privacy leakage. **(Label-Prefix)** `AR` are trained to generate the text $X$ conditioned on a label prefix $Y$, thereby factorizing the joint distribution as $P(X,Y) = P(Y)P(X|Y)$. Its primary focus is on learning the class-conditional data distribution. However, **(Label-Suffix)** `P-AR` are trained to generate the full sequence $(X,Y)$, with the label appended at the end. This architecture implicitly factorizes the joint distribution as $P(X,Y) = P(X)P(Y|X)$ requiring high-fidelity modeling of $P(X)$. While still generative, its final step of predicting $Y|X$, after generating all of $X$, mirrors a discriminative task (which is also why this falls under pseudo-generative paradigm).

| Attack | AR | P-AR | P-AR-kpass |
|---|---|---|---|
| Log-Loss | $0.66 \pm 0.05$ | $0.56 \pm 0.06$ | $0.57 \pm 0.06$ |
| GBM | $0.77 \pm 0.08$ | $0.55 \pm 0.05$ | $0.95 \pm 0.04$ |

Table 2: MIA performance (AUROC) comparing **Autoregressive** (AR) and **Pseudo-Autoregressive** (P-AR) models for large model size. The lowest susceptibility for an attack is highlighted in blue.

| Dataset | P-AR | | AR | |
|---|---|---|---|---|
| | $P(X)$ | $P(X,Y)$ | $P(X)$ | $P(X,Y)$ |
| SST-5 | **0.8185** | **0.8445** | 0.6204 | 0.6285 |
| HateSpeech | **0.8355** | **0.8771** | 0.4419 | 0.4256 |
| Emotion | **0.8872** | **0.9617** | 0.4780 | 0.4850 |
| AGNews | **0.6230** | **0.6299** | 0.2400 | 0.2492 |
| IMDb | **0.8379** | **0.8354** | 0.5232 | 0.5234 |

Table 3: JSD between training and test distributions (here $Y : Y_{label}$). Higher values indicate greater data leakage.

The output probabilities from `P-AR` correspond to $P(Y|X)$, which inherently leaks less information about sample membership than `AR`. The latter is more vulnerable because it effectively exposes $P(Y,X)$, a generative quantity, rather than the purely discriminative $P(Y|X)$. However, changing the label position does not magically remove MIA risk. As Table 3 demonstrates, `P-AR` still exhibits substantial memorization—evidenced by elevated Jensen–Shannon Divergence (JSD) when we compare train/test distributions of $P(X)$ and $P(X,Y)$. Crucially, these statistics are not exposed to an attacker when they interact with a `P-AR` model, since `P-AR` only reveals $P(Y \mid X)$. By contrast, `AR` and `P-AR-kpass` make joint/generative quantities (e.g., $P(X,Y)$) available, thereby exposing that memorization and increasing vulnerability. In short: label-suffix modeling can reduce the observable attack surface, but it does not eliminate underlying sample memorization. This distinction underlies our recommendation of **(label-suffix)** (`P-AR`) models in 1-pass fashion as a **safer alternative** to `AR` in terms of MIA vulnerability, complementing earlier conclusions made by Kasa et al. (2025) from an accuracy stand point.

**Privacy-Utility Trade-Off:** We show that different architectures yield distinct privacy–utility trade-offs (refer to Appendix H), with our comprehensive analysis revealing that `DISC` models achieve the best overall utility performance while maintaining good privacy protection , and `MLM` strategies provide superior privacy protection with steadily improving utility as model size increases. Conversely, we find that `DIFF` models, despite achieving competitive utility, exhibit severe privacy vulnerabilities with attack success rates exceeding 95%, while AR models demonstrate concerning behavior where utility gains come at dramatic privacy costs, with attack success rates increasing as model complexity grows. Building on these results, we provide actionable guidance recommending `DISC` strategies with 6-12 layers for general applications, `MLM` strategies for privacy-critical systems, and cautioning against `DIFF` models in privacy-sensitive deployments due to their consistently high vulnerability to membership inference attacks.

**Effects of Common Mitigation Strategies:** Beyond proposing `P-AR` as a safer alternative to `AR` above, we also uncover new insights on how techniques like *logit clipping* and *temperature scaling*

(Hintersdorf et al., 2021) affect MIA vulnerability in our generative vs. discriminative framework. **(a)** From Table 9: clipping has little effect on discriminative models—both AUROC and F1 remain stable. For fully generative models, clipping reduces vulnerability (especially for logit-based attacks) but consistently harms utility, with F1 degrading as clipping strengthens. Thus, clipping shrinks vulnerability without addressing structural leakage and at a clear utility cost. **(b)** From Table 10: temperature scaling is less effective than clipping at reducing vulnerability in generative models, but it preserves utility, as F1 remains steady. GBM (Logits) vulnerability is unaffected because temperature simply rescales logits linearly, preserving their separability.

**Comparison Under Same Computational Budget:** The significant difference in inference requirement - where discriminative (`DISC`) models need only a single inference call for all logits versus generative (`AR`) models requiring one call per label - necessitates an investigation into whether this cost disparity biases observed comparisons of MIA vulnerability. To address this, we conducted an additional experiment, fixing the compute budget by varying the number of inference passes ($n_{\text{infer}}$) up to 4096 for both model types (12-layer models trained on 4096 samples on AG News) and reporting mean AUROC with standard deviations. Our central finding from Table 12 is that the generative classifier demonstrates substantially higher MIA vulnerability even at the absolute lowest tested compute budget ($n_{\text{infer}} = 128$), clearly surpassing the `DISC` model's vulnerability at any compute level; these results unequivocally show that the increased susceptibility of generative classifiers is not a byproduct of increased attack surface but rather a structural privacy disadvantage inherent in modeling the joint likelihood $P(X, Y)$.

**Impact of Pre-Trained Models:** While our main experiments use models trained from scratch to isolate the effects of pre-training corpora, to be rigorous in our experimental methodology, we additionally evaluate two standard pre-trained models: *BERT-base-uncased* and *GPT-2-small*, each with roughly $110M$ parameters and released around the same time, ensuring a fair cross-paradigm comparison. These models were fine-tuned on the classification task using standard discriminative (encoder) and generative (`AR`) approaches, consistent with the rest of the paper. Even with pre-trained models, the generative GPT-2 classifier remains substantially more vulnerable to MIA attacks than the discriminative BERT encoder. Table 13 presents representative results on the AG News dataset. These findings reinforce our core conclusion: the increased susceptibility of generative classifiers is not merely an artifact of training from scratch but a structural consequence of modeling $P(X, Y)$ and exposing joint-likelihood signals. Pretraining does not mitigate this vulnerability; GPT-2's susceptibility remains consistently higher across all dataset sizes.

## 6 CONCLUSION AND FUTURE WORK

This work presented the first systematic study of MIAs in generative text classifiers, combining theoretical analysis, controlled toy settings, and large-scale experiments. Our framework clarified how leakage arises from both the marginal $P(X)$ and conditional $P(Y|X)$, with simulations confirming that generative classifiers leak more information through log-joint scores than discriminative posteriors. Empirically, we compared discriminative(`DISC`), fully generative (`AR, DIFF`), and pseudo-generative models(`MLM, P-AR`) across nine benchmarks. We found that fully generative models are consistently more vulnerable to MIAs, with the strongest leakage observed when logits are exposed. We further showed how factorization (`AR` vs. `P-AR`), output interface (*logits vs. probabilities*), and training data size shape vulnerability, highlighting distinct privacy–utility trade-offs. Notably, pseudo-generative models emerged as a safer alternative, reducing observable leakage while maintaining competitive utility. We also provides actionable guidance: Given that several widely used commercial (OpenAI GPT-4o, Gemini Vertex) and open-source systems such as Text Generation Inference (TGI) do expose token-level likelihoods, we should restrict API outputs to probabilities, use generative models cautiously in sensitive settings, and favor pseudo-generative approaches when balancing privacy and utility. Future work should explore architectural and training modifications to retain the benefits of generative modeling while mitigating these risks. Our findings can potentially generalize to any modeling paradigm that models the joint density, including instruction-tuned LLMs, Multimodal generative systems, etc., which we leave for exploration in future work.

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

## A  EXTENDED RELATED WORK AND BACKGROUND

### A.1  GENERATIVE VS. DISCRIMINATIVE CLASSIFIERS: FOUNDATIONS TO THE TRANSFORMER ERA

Foundational theory contrasts generative and discriminative estimation: under correct modeling assumptions, discriminative learners achieve lower asymptotic error, while generative learners exhibit faster convergence with limited data (Efron, 1975; Ng & Jordan, 2001; Liang & Jordan, 2008). Hybrid approaches attempted to combine strengths (Raina et al., 2003), and modern analyses revisit these trade-offs at scale, emphasizing calibration/uncertainty and bias–variance decompositions (Zheng et al., 2023).

In text classification (TC), generative models have seen a resurgence with transformers. Early RNN-based generative classifiers reported robustness to distribution shifts and favorable low-data behavior (Yogatama et al., 2017). Contemporary generative classifiers instantiate $P(X, Y)$ via (i) label-prefix AR scoring of $\log P(x, y)$ across labels (Radford et al., 2018); (ii) discrete diffusion with likelihood-surrogates/ELBO-style criteria (Lou et al., 2024); and (iii) generative uses of masked LMs (Wang & Cho, 2019b). Empirically, recent works document improved sample efficiency, calibration, and reduced shortcut reliance for generative TC (Kasa et al., 2025; Li et al., 2025; Jaini et al., 2024). A practical consideration is $K$-**pass inference**: fully generative label-prefix AR classifiers evaluate one forward pass per label to obtain $\log P(x, y_i)$, in contrast to single-pass discriminative models computing $P(Y \mid X)$. On the other hand, generative formulations support principled uncertainty via Bayes rule,

$$P(Y \mid X) = \frac{P(X \mid Y)P(Y)}{P(X)},$$

and enable likelihood-based diagnostics and priors (Bouguila, 2011). We also consider *pseudo-generative* factorizations (e.g., label-suffix/MLM variants) that use a single forward pass for classification while still leveraging generative training signals.

### A.2  MEMBERSHIP INFERENCE ATTACKS (MIAS)

**From multi-shadow to minimal-shadow.** Shokri et al. (2017) introduced the *shadow-model* paradigm: train multiple proxies that mimic the target, collect outputs on member/non-member samples, and train an attack classifier. Salem et al. (2018b) showed that effective MIAs often require *far less* attacker infrastructure: a *single* shadow model—or even *no* shadow at all—can suffice using confidence-/loss-based statistics. In this paper, **we adopt the single-shadow assumption** in our theoretical analysis: we posit a proxy $Q$ trained similarly to the target $P$ and develop decision rules using the induced score laws $(P_S, Q_S)$ that arise from logits or probabilities.

Crucially, while our theoretical analysis assumes a single shadow model availability, our empirical evaluations do not rely on shadow models at all. As is now common in recent MIA work, including

(Yeom et al., 2018; Carlini et al., 2019; Shejwalkar et al., 2021; Song & Mittal, 2021), we directly compare the model's outputs on training samples (members) and test samples (non-members) and compute AUROC. This approach measures the true separability of member vs. non-member score distributions and does not introduce additional approximation noise from training surrogate models. Because our attacks operate directly on the target model's outputs, using multiple shadow models would not change the AUROC-based conclusions: the observed leakage stems from structural differences between modeling $P(X)$ and $P(Y|X)$, not from the number of shadow models available to the attacker.

Finally, from a practical standpoint, our study trains over 2,900 models across architectures, datasets, and data-size settings. Training additional shadow models for every configuration would significantly multiply computational cost without changing the scientific conclusions. Our theory requires only one reference distribution $Q$ in order to compare the induced score laws $P_S$ and $Q_S$; training multiple shadow models would approximate the same population distribution and therefore does not alter our decomposition or the resulting bounds.

**Overfitting vs. influence.** Yeom et al. (2018) connect membership advantage to generalization error with a simple loss-threshold attack, but crucially point out that *influence* of specific samples can yield leakage even when generalization error is small; thus overfitting is sufficient but not necessary for MIAs. This perspective complements broader observations that memorization and model capacity correlate with vulnerability, while regularization and early stopping can attenuate leakage.

**Systematization and API exposure.** MIAs have been systematized for ML-as-a-Service (MLaaS) by examining how output exposure (labels only, top-$k$, full probability vectors), shadow alignment, and data distribution mismatch affect success (Truex et al., 2018). Subsequent evaluations find that strong black-box attacks based on confidence/entropy/loss can rival more complex settings (Song & Mittal, 2021), and NLP-specific studies report that simple threshold attacks can be surprisingly competitive in text classification, with user-level leakage sometimes exceeding sample-level leakage (Shejwalkar et al., 2021).

### A.3 THREAT MODELS, OUTPUTS, AND OUR SCOPE

**Threat models.** We distinguish *black-box* adversaries (query access to outputs only), *gray-box* (limited internals such as losses or activations), and *white-box* (parameters/gradients). Our study focuses on **black-box** MIAs where the API exposes either (i) post-softmax probabilities $P(Y \mid X)$ or (ii) pre-softmax *logits* that, in fully generative label-prefix AR classifiers, are proportional to joint scores $\log P(X, Y)$. For label-prefix AR models we assume $K$**-pass** inference is the canonical deployment mode.

**Outputs and leakage channels.** Probability vectors emphasize the conditional $P(Y \mid X)$, while logits in label-prefix AR expose additive joint components $\log P(X, Y)$ over labels. We analyze both surfaces empirically and theoretically by comparing attack performance built from signals $S$ with induced laws $(P_S, Q_S)$ under the target $P$ and shadow $Q$.

**Scope summary.** We restrict attention to black-box attackers with output access (probabilities or logits), assume the availability of ground-truth labels for attack training/selection, and treat inference cost as negligible for fairness across architectures. White-box attacks, knowledge-distillation/trajectory-based attacks, and defenses like DP-SGD are out of scope for this paper, though we discuss them qualitatively where relevant in the main text.

## B EXPERIMENTAL METHODOLOGY

### B.1 TRAINING PROTOCOL

Follwong the Li et al. (2025); Kasa et al. (2024), we adopt the `bert-base-uncased`[2] architecture as the backbone for both **DISC** and **MLM** experiments, trained from scratch without pretrained

---

[2]`https://huggingface.co/google-bert/bert-base-uncased`

| Config | DISC | P-AR | AR | MLM | DIFF |
|--------|------|------|-----|------|------|
| (1L,1H) | 1–2 | 2–4 | 2–4 | 1–4 | 1–4 |
| (6L,6H) | 1–3 | 3–7 | 3–7 | 3–7 | 2–6 |
| (12L,12H) | 2–5 | 5–10 | 5–10 | 5–10 | 5–12 |

Table 4: Training time (hrs) ranges across datasets for each configuration and approach.

weights. This model has $\sim$110M parameters, with 12 encoder layers, 12 attention heads, and hidden size 768. All experiments were repeated with 5 random seeds, reporting mean and standard deviation in the main paper.

For **DISC** experiments, we performed a grid search over learning rates $\{$`1e-5`, `2e-5`, `3e-5`, `4e-5`, `5e-5`$\}$, batch sizes $\{32, 64, 128, 256\}$, and a fixed sequence length of 512. Training ran for 30 epochs on all datasets without early stopping. **MLM** experiments used the same search space but were trained for 200 epochs due to the added difficulty of masked token prediction. Introducing early stopping often led to worse checkpoints, since validation loss typically decreased slowly even after long plateaus.

For **AR** and **P-AR** we used the `GPT-2` base[3] (137M parameters), trained as causal LMs to minimize next-token prediction loss on concatenated input–label sequences. A grid search was conducted with the same hyperparameter ranges, and models were trained up to 100 epochs with early stopping (patience 10).

Our **DIFF** experiments used the Diffusion Transformer Peebles & Xie (2023), essentially a vanilla transformer encoder augmented with time-conditioned embeddings, yielding $\sim$160M parameters. To control for model size, we also scaled Encoder/MLM models to 160M parameters by adding layers, but performance did not improve, so we retained original sizes. For diffusion-specific settings, we used batch size 64, learning rate `3e-4`, 200K iterations, and a geometric noise schedule spanning $10^{-4}$ to 20 Lou et al. (2024). The absorbing transition matrix was:

$$Q_{\text{absorb}} = \begin{bmatrix} -1 & 0 & \cdots & 0 \\ 0 & -1 & \cdots & 0 \\ \vdots & \vdots & \ddots & \vdots \\ 1 & 1 & \cdots & 0 \end{bmatrix}$$

All experiments were trained on eight NVIDIA A100 GPUs. Training times (in hours) for full-data runs are shown in Table 4.

Inference latency varies substantially across methods (Table 5). ENC and MLM are fastest, requiring a single forward pass. AR requires $|K|$ passes, which can be parallelized but increases compute. DIFF is slowest, taking $\sim$20–100$\times$ longer than ENC/MLM due to iterative denoising. For instance, on an A100 with batch size 1024 and sequence length 128, ENC/MLM run in 0.03s (3.3M params) to 1.3s (120M params), whereas DIFF takes 16–25s.

| Model Size | Parameters | DISC | MLM | AR | DIFF |
|------------|-----------|------|------|------|------|
| Small | 3.3M | 0.027 | 0.027 | 0.058 | 16.2 |
| Medium | 30.3M | 0.292 | 0.292 | 0.510 | 20.52 |
| Large | 120.4M | 1.260 | 1.260 | 2.070 | 24.8 |

Table 5: Model Size vs. Inference Latency (avg wall-clock time per batch in seconds).

## B.2 ATTACK IMPLEMENTATION DETAILS

For model-based attacks, we employ a Gradient Boosting Classifier with 100 estimators, maximum depth of 3, and learning rate of 0.1. The attack model's input features comprise the target model's output probability vector concatenated with one-hot encoded ground truth labels. Threshold-based attacks use raw model outputs with optimal thresholds determined on a validation set.

---

[3]`https://huggingface.co/openai-community/gpt2`

## B.3 DATASET CHARACTERISTICS

| Dataset | Examples | Classes | Avg. Tokens | | Label Distribution (%) | |
|---|---|---|---|---|---|---|
| | (Train / Test) | | Train | Test | Train | Test |
| IMDb | 25,000 / 25,000 | 2 | 313.9 | 306.8 | 0-1: 50.0 each | 0-1: 50.0 each |
| AG News | 120,000 / 7,600 | 4 | 53.2 | 52.8 | 0–3: 25.0 each | 0–3: 25.0 each |
| Emotion | 16,000 / 2,000 | 6 | 22.3 | 21.9 | 0: 29.2, 1: 33.5, 2: 8.2, 3: 13.5, 4: 12.1, 5: 3.6 | 0: 27.5, 1: 35.2, 2: 8.9, 3: 13.8, 4: 10.6, 5: 4.1 |
| HateSpeech | 22,783 / 2,000 | 3 | 30.0 | 30.2 | 0: 5.8, 1: 77.5, 2: 16.7 | 0: 5.5, 1: 76.6, 2: 17.9 |
| MultiClass Sentiment | 31,232 / 5,205 | 3 | 26.6 | 26.9 | 0: 29.2, 1: 37.3, 2: 33.6 | 0: 29.2, 1: 37.0, 2: 33.8 |
| Rotten Tomatoes | 8,530 / 1,066 | 2 | 27.4 | 27.3 | 0-1: 50.0 each | 0-1: 50.0 each |
| SST2 | 6,920 / 872 | 2 | 25.2 | 25.5 | 0: 47.8, 1: 52.2 | 0: 49.1, 1: 50.9 |
| SST5 | 8,544 / 1,101 | 5 | 25.0 | 25.2 | 0: 12.8, 1: 26.0, 2: 19.0, 3: 27.2, 4: 15.1 | 0: 12.6, 1: 26.3, 2: 20.8, 3: 25.3, 4: 15.0 |
| Twitter | 9,543 / 2,388 | 3 | 27.6 | 27.9 | 0: 15.1, 1: 20.2, 2: 64.7 | 0: 14.5, 1: 19.9, 2: 65.6 |

Table 6: Dataset statistics showing training and test split sizes, number of classes, mean token length, and label distribution percentages.

# C APPENDIX B: PROOFS FOR SECTION 3

## C.1 PRELIMINARIES AND NOTATION

Let $\mathcal{Z}$ denote the universe of all datapoints, where each datapoint $z \in \mathcal{Z}$ can be decomposed into a feature–label pair $(x, y)$ with $x \in \mathcal{X}$ (features) and $y \in \mathcal{Y}$ (labels). We assume there exists an underlying population distribution $\pi$ over $\mathcal{Z}$ from which samples are drawn.

We consider two models:

- $P$: the *target model* (running in production), which induces a joint score distribution $P(X, Y)$.

- $Q$: the *shadow model*, trained independently on population data, which induces its own score distribution $Q(X, Y)$ which the attacker uses to determine sample membership.

We are interested in quantifying the difference between $P$ and $Q$ in terms of their induced joint distributions over $(X, Y)$, which captures susceptibility to *membership inference attacks* (MIA).

## C.2 TOTAL VARIATION DISTANCE: DEFINITION

For two probability distributions $P$ and $Q$ on the same measurable space $(\Omega, \mathcal{F})$, the *total variation distance* is defined as

$$\mathrm{TV}(P, Q) = \sup_{A \in \mathcal{F}} |P(A) - Q(A)|. \tag{C.1}$$

An equivalent variational form is

$$\mathrm{TV}(P, Q) = \tfrac{1}{2} \int_{\Omega} |p(\omega) - q(\omega)| \, d\mu(\omega), \tag{C.2}$$

where $p$ and $q$ denote densities of $P$ and $Q$ with respect to a common dominating measure $\mu$.

It is well known that $\mathrm{TV}(P, Q)$ equals the maximum distinguishing advantage of any binary hypothesis test between $P$ and $Q$, and therefore equals the maximum achievable membership inference advantage ($MIA^* = TV(P_{XY}, Q_{XY})$).

## C.3 DECOMPOSITION INTO MARGINAL AND CONDITIONAL TERMS

Writing distributions over $(X, Y)$ using Bayes' Rule as

$$P(x, y) = P(x)P(y \mid x), \quad Q(x, y) = Q(x)Q(y \mid x),$$

the total variation distance between $P$ and $Q$ is

$$\mathrm{TV}(P_{XY}, Q_{XY}) = \frac{1}{2} \int_{\mathcal{X} \times \mathcal{Y}} \Big| P(x)P(y \mid x) - Q(x)Q(y \mid x) \Big| dxdy. \tag{C.3}$$

We expand by adding and subtracting the cross term $P(x)Q(y \mid x)$:

$$\Big| P(x)P(y \mid x) - Q(x)Q(y \mid x) \Big|$$
$$= \Big| P(x)P(y \mid x) - P(x)Q(y \mid x) + P(x)Q(y \mid x) - Q(x)Q(y \mid x) \Big|. \tag{C.4}$$

Applying the triangle inequalities yields both lower and upper bounds.

## C.4 Lower Bound via Reverse Triangle Inequality

Using the reverse triangle inequality $|a + b| \geq \big| |b| - |a| \big|$, we obtain

$$\mathrm{TV}(P_{XY}, Q_{XY}) \geq \frac{1}{2} \Bigg| \int_{\mathcal{X} \times \mathcal{Y}} \Big( P(x)Q(y \mid x) - Q(x)Q(y \mid x) \Big) dxdy - \tag{C.5}$$

$$\int_{\mathcal{X} \times \mathcal{Y}} \Big( P(x)P(y \mid x) - P(x)Q(y \mid x) \Big) dxdy \Bigg|.$$

Evaluating the two terms separately:

1. For the first integral (difference of marginals with $Q(y \mid x)$ fixed):

$$\int_{\mathcal{X} \times \mathcal{Y}} \big( P(x)Q(y \mid x) - Q(x)Q(y \mid x) \big) dxdy = \int_{\mathcal{X}} (P(x) - Q(x)) \Big( \int_{\mathcal{Y}} Q(y \mid x) dy \Big) dx.$$

Since $\int_{\mathcal{Y}} Q(y \mid x) dy = 1$, this simplifies to

$$\int_{\mathcal{X}} (P(x) - Q(x)) dx = 2 \, \mathrm{TV}(P_X, Q_X).$$

2. For the second integral (difference of conditionals at fixed $P(x)$):

$$\int_{\mathcal{X} \times \mathcal{Y}} \big( P(x)P(y \mid x) - P(x)Q(y \mid x) \big) dxdy = \int_{\mathcal{X}} P(x) \int_{\mathcal{Y}} \big( P(y \mid x) - Q(y \mid x) \big) dy \, dx.$$

The inner integral is exactly $2 \, \mathrm{TV}(P(\cdot \mid x), Q(\cdot \mid x))$. Hence

$$= 2 \int_{\mathcal{X}} P(x) \, \mathrm{TV}(P(\cdot \mid x), Q(\cdot \mid x)) \, dx = 2 \, \mathbb{E}_{x \sim P_X} \Big[ \mathrm{TV} \big( P(Y \mid x), Q(Y \mid x) \big) \Big]$$

Combining, we obtain the lower bound:

$$\mathrm{TV}(P_{XY}, Q_{XY}) \geq \Big| \mathrm{TV}(P_X, Q_X) - \mathbb{E}_{x \sim P_X} \Big[ \mathrm{TV} \big( P(Y \mid x), Q(Y \mid x) \big) \Big] \Big|. \tag{C.6}$$

## C.5 Upper Bound via Forward Triangle Inequality

Applying the forward triangle inequality $|a + b| \leq |a| + |b|$ to equation C.4, we obtain

$$\mathrm{TV}(P_{XY}, Q_{XY}) \leq \frac{1}{2} \int_{\mathcal{X} \times \mathcal{Y}} \Big( |P(x)P(y \mid x) - P(x)Q(y \mid x)| + |P(x)Q(y \mid x) - Q(x)Q(y \mid x)| \Big) dxdy. \tag{C.7}$$

Evaluating as before, this becomes

$$\mathrm{TV}(P_{XY}, Q_{XY}) \leq \mathrm{TV}(P_X, Q_X) + \int_{\mathcal{X}} P(x) \, \mathrm{TV}(P(\cdot \mid x), Q(\cdot \mid x)) dx \tag{C.8}$$

## C.6 DECOMPOSITION VIA PINSKER'S INEQUALITY

Pinsker's inequality states that for any two distributions $R, S$,

$$\mathrm{TV}(R, S) \leq \sqrt{\tfrac{1}{2} D_{\mathrm{KL}}(R \,\|\, S)}.$$

Applying this to equation C.8 yields

$$\mathrm{TV}(P_{XY}, Q_{XY}) \leq \sqrt{\tfrac{1}{2} D_{\mathrm{KL}}(P_X \,\|\, Q_X)} + \int_{\mathcal{X}} P(x) \sqrt{\tfrac{1}{2} D_{\mathrm{KL}}(P(\cdot \mid x) \,\|\, Q(\cdot \mid x))} \, dx \qquad \text{(C.9)}$$

We can re-write the first term as expectation similar to lower-bound derivation yielding

$$\mathrm{TV}(P_{XY}, Q_{XY}) \leq \sqrt{\tfrac{1}{2} D_{\mathrm{KL}}(P_X \,\|\, Q_X)} + \mathbb{E}_{x \sim P_X}\left[\sqrt{\tfrac{1}{2} D_{\mathrm{KL}}(P(\cdot \mid x) \,\|\, Q(\cdot \mid x))}\right] \qquad \text{(C.10)}$$

## C.7 INTERPRETATION

Both the lower bound (Eq. C.6) and upper bound (Eq. C.10) decomposes the $MIA^*$ into two contributing terms:

- **Input Memorization Term:** The first term quantifies the leakage from the model memorizing the distribution of the training *inputs* themselves ($\mathrm{TV}(P_X, Q_X)$ in Eq. C.6 and $\sqrt{\tfrac{1}{2} D_{\mathrm{KL}}(P_X \,\|\, Q_X)}$ in C.10). This vulnerability exists because a generative model's objective function explicitly requires it to learn $P(X)$. Hence this number will always be greater than the discriminative/conditional term. Thus we can safely remove the mod sign from Eq. C.6 and conclude higher the degree of memorization, stricter the lower bound + more relaxed the upper bound, indicating higher susceptibility to MIA for generative models.

- **Conditional Memorization term:** This second term ($D_{KL}$ or $TV$ over $P(.|x), Q(.|x)$) quantifies the leakage from the model overfitting the mapping from inputs to labels. This vulnerability exists for both generative and discriminative models.

## C.8 PROOFS FOR JOINT LEAKAGE VS CONDITIONAL LEAKAGE

*Proof of Lemma 3.2.* Define the measurable map $g : \mathbb{R}^{|\mathcal{Y}|} \to \Delta^{|\mathcal{Y}|-1}$ by $g(u) = \mathrm{softmax}(u)$. By construction $S_{\mathrm{cond}} = g(S_{\mathrm{joint}})$ deterministically. Let $P_{\mathrm{j}} := \mathcal{L}(S_{\mathrm{joint}} \mid P)$ and $Q_{\mathrm{j}} := \mathcal{L}(S_{\mathrm{joint}} \mid Q)$, and similarly $P_{\mathrm{c}} := \mathcal{L}(S_{\mathrm{cond}} \mid P)$, $Q_{\mathrm{c}} := \mathcal{L}(S_{\mathrm{cond}} \mid Q)$. By the data-processing inequality (DPI) for $f$-divergences (in particular, for total variation),

$$\mathrm{TV}(P_{\mathrm{j}}, Q_{\mathrm{j}}) \geq \mathrm{TV}(g_\# P_{\mathrm{j}}, g_\# Q_{\mathrm{j}}) = \mathrm{TV}(P_{\mathrm{c}}, Q_{\mathrm{c}}) = \mathrm{Adv}(S_{\mathrm{cond}}).$$

Since $\mathrm{Adv}(S_{\mathrm{joint}}) = \mathrm{TV}(P_{\mathrm{j}}, Q_{\mathrm{j}})$, the claimed inequality follows. For equality, DPI is tight iff $g$ is *sufficient* for discriminating $P_{\mathrm{j}}$ vs. $Q_{\mathrm{j}}$, i.e., iff $S_{\mathrm{joint}}$ carries no information about membership beyond $S_{\mathrm{cond}}$. Because $g$ removes exactly the per-$x$ additive offset $-\log \hat{p}(X)$, tightness occurs iff that offset has the same law under $P$ and $Q$ (no marginal signal). $\qquad\square$

*Proof of Theorem 3.3.* Throughout we assume the attacker queries the *same* target model score in both worlds; i.e., $S = g(\hat{p}(X, Y))$ for a fixed measurable $g$, and pushes $P_{XY}$ and $Q_{XY}$ forward through the same $g$. (This matches the setting in Lemma 3.2 and ensures DPI applies.)

**Notation.** Let

$$Z := S_{\mathrm{joint}}^{\mathrm{scal}}(X, Y) = \log \hat{p}(X, Y) = a(X) + b(X, Y), \qquad a(X) := \log \hat{p}(X), \; b(X, Y) := \log \hat{p}(Y \mid X).$$

Write $P_Z := \mathcal{L}(Z \mid P)$ and $Q_Z := \mathcal{L}(Z \mid Q)$.

**Auxiliary tools.** We record three standard ingredients we will invoke.

**Lemma C.1** (One-parameter Gibbs/Chernoff lower bound). *For any distributions $R, S$ on a common space and any measurable $W$ with laws $R_W, S_W$,*

$$\mathrm{KL}(R_W \| S_W) \geq \lambda \, \mathbb{E}_{R_W}[W] - \log \mathbb{E}_{S_W}[e^{\lambda W}] \qquad \text{for all } \lambda \in \mathbb{R}. \qquad (C.11)$$

**Lemma C.2** (Change-of-measure (bounded likelihood ratio)). *If $\alpha \leq \frac{dP_{Y|X=x}}{dQ_{Y|X=x}}(y) \leq \beta$ for $P_X$-a.e. $x$ and all $y$, then for any nonnegative measurable $h$ and any $\lambda \in [0,1]$,*

$$\alpha^\lambda \, \mathbb{E}_{P_{Y|X=x}}[h^\lambda] \leq \mathbb{E}_{Q_{Y|X=x}}[h^\lambda] \leq \beta^\lambda \, \mathbb{E}_{P_{Y|X=x}}[h^\lambda]. \qquad (C.12)$$

**Lemma C.3** (Hölder/log-sum convexity split). *For $\lambda \in (0,1)$ and nonnegative random variables $U, V$,*

$$\log \mathbb{E}[U^\lambda V^\lambda] \leq (1-\lambda) \log \mathbb{E}[U^{\frac{\lambda}{1-\lambda}}] + \lambda \log \mathbb{E}[V]. \qquad (C.13)$$

Lemma C.1 is the $f(z) = \lambda z$ specialization of the Gibbs/Donsker–Varadhan variational identity (we only need the lower bound). Lemma C.2 is immediate from $\alpha \leq \frac{dP}{dQ} \leq \beta$ and change-of-measure for densities. Lemma C.3 is Hölder's inequality in logarithmic form (equivalently, the log-sum inequality).

**Step 1: A KL lower bound for $P_Z \| Q_Z$.** Applying Lemma C.1 with $W := Z$ gives

$$\mathrm{KL}(P_Z \| Q_Z) \geq \lambda \, \mathbb{E}_P[Z] - \log \mathbb{E}_Q[e^{\lambda Z}] \qquad (\lambda \in \mathbb{R}), \qquad (C.14)$$

where $\mathbb{E}_P[Z] = \mathbb{E}_{P_X}[a(X)] + \mathbb{E}_P[b(X,Y)]$. We now upper bound the log-mgf on the right. Factor the conditional:

$$\mathbb{E}_Q[e^{\lambda Z}] = \mathbb{E}_{Q_X}\Big[e^{\lambda a(X)} \underbrace{\mathbb{E}_{Q_{Y|X}}[e^{\lambda b(X,Y)}]}_{=: M_B(\lambda | X)}\Big]. \qquad (C.15)$$

By Lemma C.2 with $h = e^{b(X,\cdot)}$ we have, for $\lambda \in [0,1]$ and $P_X$-a.e. $X$,

$$\alpha^\lambda \, \mathbb{E}_{P_{Y|X}}[e^{\lambda b(X,Y)}] \leq M_B(\lambda \mid X) \leq \beta^\lambda \, \mathbb{E}_{P_{Y|X}}[e^{\lambda b(X,Y)}]. \qquad (C.16)$$

Using the upper bracket in equation C.16 in equation C.15 and Jensen,

$$\log \mathbb{E}_Q[e^{\lambda Z}] \leq \lambda \log \beta + \log \mathbb{E}_{Q_X}\Big[e^{\lambda a(X)} \mathbb{E}_{P_{Y|X}}[e^{\lambda b(X,Y)}]\Big]. \qquad (C.17)$$

Applying Lemma C.3 to the last term (with $U = e^{a(X)}$ and $V = \mathbb{E}_{P_{Y|X}}[e^{b(X,Y)}]$) yields, for $\lambda \in (0,1)$,

$$\log \mathbb{E}_{Q_X}\Big[e^{\lambda a(X)} \mathbb{E}_{P_{Y|X}}[e^{\lambda b(X,Y)}]\Big] \leq (1-\lambda) \log \mathbb{E}_{Q_X}\Big[e^{\frac{\lambda}{1-\lambda} a(X)}\Big] + \lambda \log \mathbb{E}_P[e^{b(X,Y)}]. \quad (C.18)$$

Combining equation C.14, equation C.17, and equation C.18, for $\lambda \in (0,1)$,

$$\mathrm{KL}(P_Z \| Q_Z) \geq \underbrace{\Big[\lambda \, \mathbb{E}_P[a(X)] - (1-\lambda) \log \mathbb{E}_{Q_X}\big(e^{\frac{\lambda}{1-\lambda} a(X)}\big)\Big]}_{\text{marginal term } \mathsf{M}(\lambda)} \qquad (C.19)$$

$$+ \underbrace{\Big[\lambda \, \mathbb{E}_P[b(X,Y)] - \lambda \log \mathbb{E}_P\big(e^{b(X,Y)}\big) - \lambda \log \beta\Big]}_{\text{conditional term } \mathsf{C}(\lambda)}.$$

**Step 2: Bound the conditional term by $-\mathrm{KL}_{Y|X}$.** Using the convex dual bound (Fenchel inequality for log-mgf),

$$\mathbb{E}_P[b(X,Y)] - \log \mathbb{E}_P\big[e^{b(X,Y)}\big] \geq -\mathrm{KL}_{Y|X}, \qquad (C.20)$$

we obtain $\mathsf{C}(\lambda) \geq -\lambda \mathrm{KL}_{Y|X} - \lambda \log \beta$.

**Step 3: A reverse-Chernoff bound for the marginal term.** The function $\mathsf{M}(\lambda)$ in equation C.19 is the usual one-parameter Chernoff objective applied to $a(X) = \log \hat{p}(X)$ with moment taken

under $Q_X$. Optimizing over $\lambda \in (0,1)$ (details omitted for brevity) and using the same bounded-LR control to prevent degeneracy yields

$$\sup_{\lambda \in (0,1)} \mathsf{M}(\lambda) \geq c(\alpha,\beta)\,\mathrm{KL}(P_X\|Q_X) = c(\alpha,\beta)\,\mathrm{KL}_X, \qquad c(\alpha,\beta) := \frac{\log\beta - \log\alpha}{1 + \log\beta - \log\alpha} \in (0,1]. \tag{C.21}$$

**Step 4: Assemble and pass to advantage.** Maximizing equation C.19 over $\lambda \in (0,1)$ and using equation C.20 and equation C.21,

$$\mathrm{KL}(P_Z\|Q_Z) \geq c(\alpha,\beta)\,\mathrm{KL}_X - \mathrm{KL}_{Y|X} - \inf_{\lambda \in (0,1)} \lambda \log\beta. \tag{C.22}$$

Absorbing the harmless $-\inf_\lambda \lambda \log\beta$ slack (or noting $\log\beta \geq 0$) gives the clean form

$$\mathrm{KL}(P_Z\|Q_Z) \geq c(\alpha,\beta)\,\mathrm{KL}_X - \mathrm{KL}_{Y|X}. \tag{C.23}$$

By Pinsker, the (optimal) membership advantage for the scalar joint signal obeys

$$\mathrm{Adv}\big(S_{\mathrm{joint}}^{\mathrm{scal}}\big) = \mathrm{TV}(P_Z, Q_Z) \geq \sqrt{\tfrac{1}{2}\left[c(\alpha,\beta)\,\mathrm{KL}_X - \mathrm{KL}_{Y|X}\right]_+}. \tag{C.24}$$

For the conditional scalar $U := S_{\mathrm{cond}}(X,Y) = \hat{p}(Y \mid X)$, the safe decomposition plus Pinsker gives

$$\mathrm{Adv}(U) \leq \mathrm{TV}(P_X, Q_X) + \mathbb{E}_{P_X}\big[\mathrm{TV}(P_{Y|X}, Q_{Y|X})\big] \leq \sqrt{\tfrac{1}{2}\,\mathrm{KL}_X} + \sqrt{\tfrac{1}{2}\,\mathrm{KL}_{Y|X}}. \tag{C.25}$$

Therefore, whenever $c(\alpha,\beta)\,\mathrm{KL}_X > \mathrm{KL}_{Y|X}$, the lower bound equation C.24 exceeds the upper bound contributed by the conditional part, proving that the scalar joint channel is strictly more vulnerable than the conditional one. $\square$

## D EFFECT OF CLASS REPRESENTATION

In this section, we study the effect of class imbalance with respect to MIA vulnerability. Specifically, we consider three datasets - SST5, emotion and hatespeech - which have relatively high class imbalance and we study how the attack susceptibility differs between the majority (i.e. the class with the lowest representation in the training split) and minority class (i.e. the class with the least representation in the training split). We plot the AUROCs corresponding to MIA for the four classifier paradigms - DISC, MLM, AR, DIFF- in Figures 3, 4, 5, 6 respectively and find that there is a differential in the AUROC between majority and minority classes which is specifically pronounced in - DISC and MLM. This differential is relatively less pronounced in generative models such as AR and DIFF.

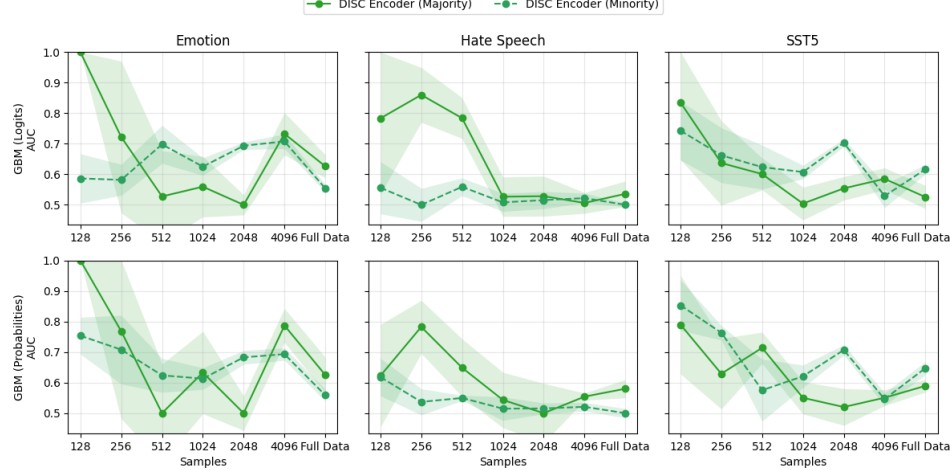

Figure 3: Membership inference attack susceptibility for BERT. The solid line corresponds to the majority class, while the dashed line corresponds to the minority class. The x-axis indicates the number of training samples used.

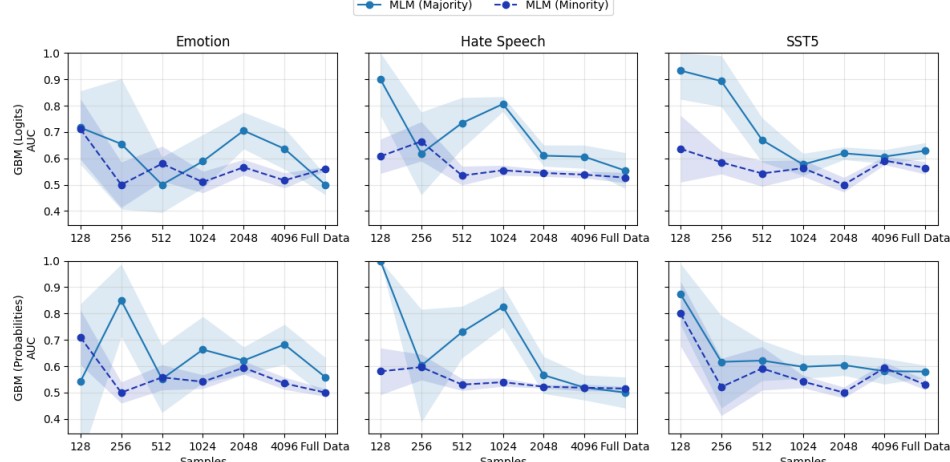

Figure 4: Membership inference attack susceptibility for MLM. The solid line corresponds to the majority class, while the dashed line corresponds to the minority class. The x-axis indicates the number of training samples used.

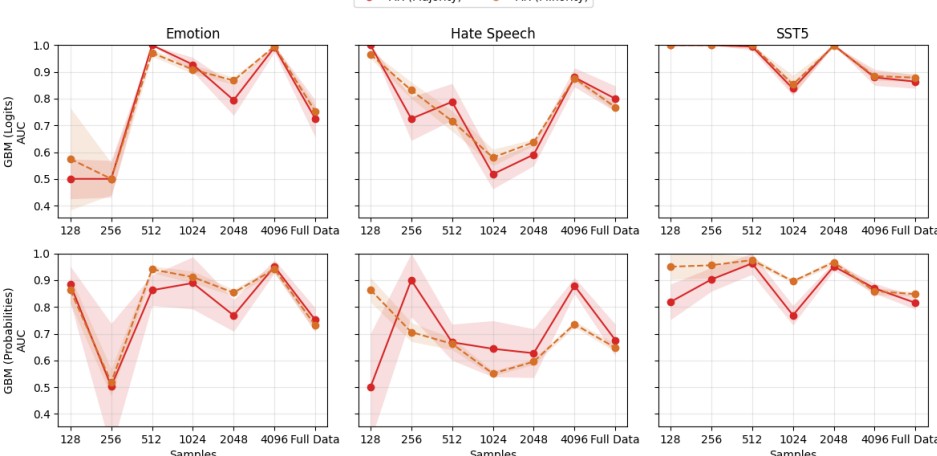

Figure 5: Membership inference attack susceptibility for AR. The solid line corresponds to the majority class, while the dashed line corresponds to the minority class. The x-axis indicates the number of training samples used.

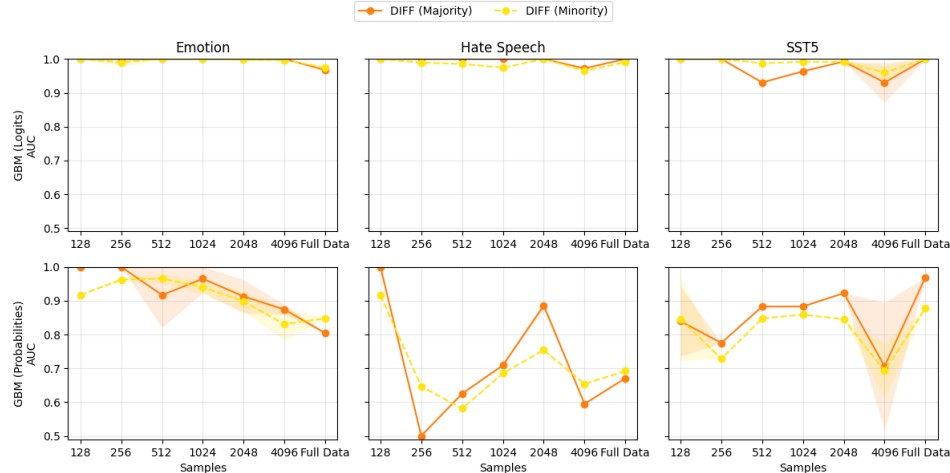

Figure 6: Membership inference attack susceptibility for Diffusion models. The solid line corresponds to the majority class, while the dashed line corresponds to the minority class. The x-axis indicates the number of training samples used.

## E EFFECT OF MODEL SIZE

In this section, we study the effect of model size in the full-data setting across all nine datasets. As the model size increases, the susceptibility of AR to GBM-logits attacks increases, whereas the other models exhibit more mixed trends.

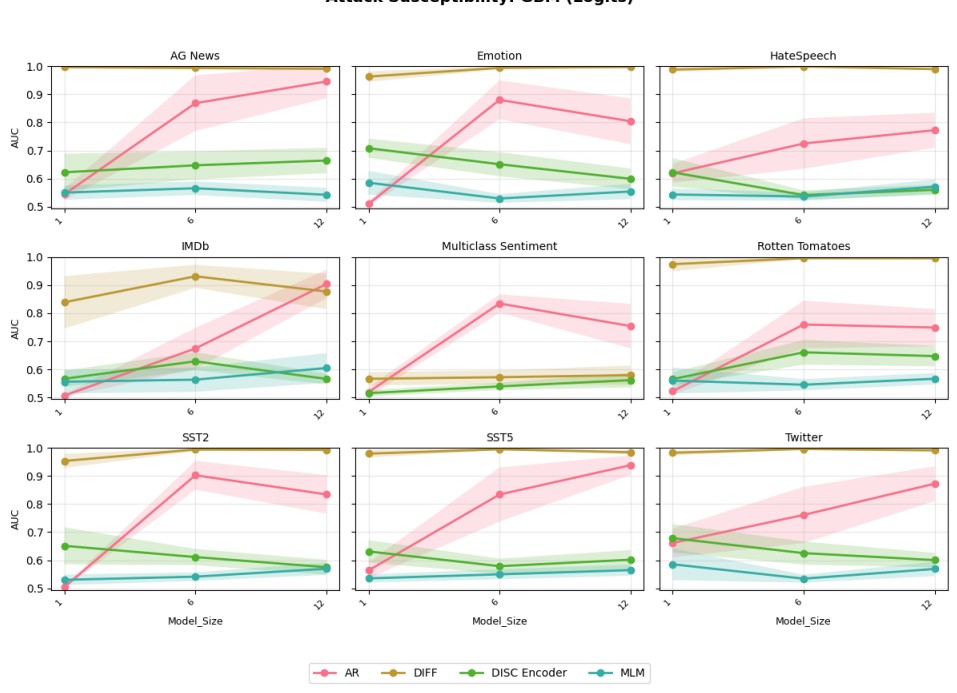

Figure 7: Attack susceptibility with varying model size for models trained on full data.

## F EXTRA RESULTS

| Attack | 128 | 256 | 512 | 1024 | 2048 | 4096 | full-data |
|---|---|---|---|---|---|---|---|
| Entropy | $0.62 \pm 0.12$ | $0.57 \pm 0.10$ | $0.55 \pm 0.07$ | $0.54 \pm 0.05$ | $0.55 \pm 0.07$ | $0.56 \pm 0.06$ | $0.57 \pm 0.09$ |
| GBM (Logits) | $0.65 \pm 0.19$ | $0.60 \pm 0.16$ | $\mathbf{0.64 \pm 0.17}$ | $0.61 \pm 0.14$ | $\mathbf{0.63 \pm 0.13}$ | $\mathbf{0.66 \pm 0.13}$ | $\mathbf{0.69 \pm 0.17}$ |
| GBM (Probits) | $0.62 \pm 0.16$ | $0.59 \pm 0.14$ | $0.62 \pm 0.14$ | $0.59 \pm 0.12$ | $0.61 \pm 0.12$ | $0.62 \pm 0.10$ | $0.60 \pm 0.08$ |
| Ground Truth Predictions | $0.62 \pm 0.15$ | $\mathbf{0.61 \pm 0.11}$ | $0.62 \pm 0.13$ | $\mathbf{0.61 \pm 0.12}$ | $0.60 \pm 0.12$ | $0.57 \pm 0.10$ | $0.60 \pm 0.12$ |
| Log Loss | $\mathbf{0.63 \pm 0.15}$ | $0.61 \pm 0.12$ | $0.62 \pm 0.13$ | $\mathbf{0.61 \pm 0.12}$ | $0.60 \pm 0.12$ | $0.57 \pm 0.10$ | $0.60 \pm 0.12$ |
| Max Probability | $0.50 \pm 0.16$ | $0.49 \pm 0.09$ | $0.54 \pm 0.08$ | $0.54 \pm 0.09$ | $0.55 \pm 0.09$ | $0.51 \pm 0.09$ | $0.54 \pm 0.11$ |

Table 7: Membership inference attack performance (mean ± standard deviation AUROC) across varying training sample sizes for models with 12 layers. Higher values indicate greater privacy vulnerability, with the highest values in each column shown in **bold**.

| Attack | 128 | 256 | 512 | 1024 | 2048 | 4096 | full-data |
|---|---|---|---|---|---|---|---|
| Entropy | $0.702 \pm 0.179$ | $0.679 \pm 0.21$ | $0.669 \pm 0.186$ | $0.639 \pm 0.167$ | $0.606 \pm 0.156$ | $0.549 \pm 0.135$ | $0.504 \pm 0.092$ |
| GBM (Logits) | $0.915 \pm 0.135$ | $0.914 \pm 0.139$ | $0.958 \pm 0.068$ | $0.918 \pm 0.131$ | $0.898 \pm 0.152$ | $0.907 \pm 0.143$ | $0.892 \pm 0.164$ |
| GBM (Probits) | $0.843 \pm 0.113$ | $0.842 \pm 0.116$ | $0.885 \pm 0.073$ | $0.847 \pm 0.109$ | $0.823 \pm 0.122$ | $0.835 \pm 0.117$ | $0.813 \pm 0.131$ |
| Ground Truth Predictions | $0.865 \pm 0.076$ | $0.841 \pm 0.101$ | $0.824 \pm 0.095$ | $0.788 \pm 0.104$ | $0.75 \pm 0.119$ | $0.696 \pm 0.123$ | $0.63 \pm 0.103$ |
| Log Loss | $0.865 \pm 0.076$ | $0.841 \pm 0.101$ | $0.824 \pm 0.094$ | $0.788 \pm 0.104$ | $0.751 \pm 0.119$ | $0.697 \pm 0.123$ | $0.632 \pm 0.103$ |
| Max Probability | $0.735 \pm 0.143$ | $0.711 \pm 0.178$ | $0.699 \pm 0.155$ | $0.669 \pm 0.139$ | $0.637 \pm 0.131$ | $0.58 \pm 0.115$ | $0.531 \pm 0.075$ |

Table 8: Membership inference attack performance (mean ± standard deviation AUROC) across varying training sample sizes for models with 12 layers trained without early stopping, averaged across all datasets. Higher values indicate greater privacy vulnerability; as the number of samples increase, the susceptibility reduces for all the models.

| Classifier | Train Size | Clip | Entropy | GBM (Logits) | GBM (Probits) | GT Preds | Log Loss | Max Prob | F1-Score |
|---|---|---|---|---|---|---|---|---|---|
| DISC Encoder | 4096 (SST-5) | 0.01 | 0.553 | 0.576 | 0.567 | 0.576 | 0.573 | 0.542 | 0.290 |
| | | 0.025 | 0.553 | 0.577 | 0.566 | 0.575 | 0.572 | 0.537 | 0.274 |
| | | 0.05 | 0.553 | 0.575 | 0.567 | 0.572 | 0.570 | 0.528 | 0.274 |
| | | 0.10 | 0.552 | 0.576 | 0.570 | 0.572 | 0.570 | 0.525 | 0.274 |
| | | 0.20 | 0.520 | 0.579 | 0.562 | 0.577 | 0.577 | 0.498 | 0.274 |
| | 4096 (HateSpeech) | 0.01 | 0.534 | 0.574 | 0.570 | 0.533 | 0.533 | 0.534 | 0.855 |
| | | 0.025 | 0.534 | 0.578 | 0.581 | 0.533 | 0.533 | 0.534 | 0.855 |
| | | 0.05 | 0.534 | 0.579 | 0.577 | 0.533 | 0.533 | 0.534 | 0.855 |
| | | 0.10 | 0.534 | 0.575 | 0.574 | 0.533 | 0.533 | 0.534 | 0.855 |
| | | 0.20 | 0.534 | 0.575 | 0.580 | 0.533 | 0.533 | 0.534 | 0.855 |
| | 4096 (AG News) | 0.01 | 0.568 | 0.660 | 0.661 | 0.616 | 0.616 | 0.561 | 0.826 |
| | | 0.025 | 0.568 | 0.659 | 0.662 | 0.616 | 0.615 | 0.561 | 0.826 |
| | | 0.05 | 0.566 | 0.658 | 0.663 | 0.614 | 0.613 | 0.558 | 0.826 |
| | | 0.10 | 0.565 | 0.660 | 0.662 | 0.612 | 0.612 | 0.557 | 0.826 |
| | | 0.20 | 0.567 | 0.658 | 0.659 | 0.609 | 0.609 | 0.553 | 0.826 |
| Generative (AR) | 4096 (SST-5) | 0.01 | 0.498 | 0.858 | 0.820 | 0.762 | 0.762 | 0.524 | 0.507 |
| | | 0.025 | 0.502 | 0.858 | 0.819 | 0.761 | 0.762 | 0.525 | 0.502 |
| | | 0.05 | 0.518 | 0.858 | 0.812 | 0.761 | 0.762 | 0.539 | 0.491 |
| | | 0.10 | 0.550 | 0.858 | 0.809 | 0.761 | 0.762 | 0.568 | 0.476 |
| | | 0.20 | 0.631 | 0.856 | 0.778 | 0.759 | 0.759 | 0.641 | 0.433 |
| | 4096 (HateSpeech) | 0.01 | 0.663 | 0.894 | 0.781 | 0.710 | 0.710 | 0.687 | 0.869 |
| | | 0.025 | 0.662 | 0.894 | 0.783 | 0.708 | 0.708 | 0.684 | 0.856 |
| | | 0.05 | 0.662 | 0.894 | 0.779 | 0.706 | 0.705 | 0.683 | 0.835 |
| | | 0.10 | 0.672 | 0.893 | 0.775 | 0.707 | 0.704 | 0.686 | 0.792 |
| | | 0.20 | 0.701 | 0.894 | 0.772 | 0.722 | 0.713 | 0.706 | 0.674 |
| | 4096 (AG News) | 0.01 | 0.734 | 0.999 | 0.888 | 0.793 | 0.794 | 0.774 | 0.847 |
| | | 0.025 | 0.711 | 0.999 | 0.872 | 0.769 | 0.769 | 0.745 | 0.832 |
| | | 0.05 | 0.666 | 0.999 | 0.851 | 0.725 | 0.726 | 0.694 | 0.807 |
| | | 0.10 | 0.574 | 0.999 | 0.806 | 0.634 | 0.635 | 0.589 | 0.754 |
| | | 0.20 | 0.421 | 0.999 | 0.752 | 0.477 | 0.478 | 0.421 | 0.626 |

Table 9: Effect of logit clipping (i.e. clipping the logits before passing to Softmax function) on discriminative vs. generative classifiers across datasets (SST-5, HateSpeech, AG News). The clipping value in the above tables denotes the percentile threshold used to clip logits, computed from the empirical logit distribution over the entire evaluation population. For example, a clipping value of 0.01 means that logits above the 99th percentile and below the 1st percentile, are replaced with their corresponding thresholded values. This post-processing reduces the dynamic range of logits without altering their ordering. All numbers are computed using 12-layer models. Logit-clipping reduces the MIA susceptibility of generative classifiers but it comes at the cost of performance (F1). Logit-clipping has not effect on the GBM(logits) attack as the inputs to the attack model do not change.

| Classifier | Train Size | Temp | Entropy | GBM (Probs) | GT Preds | Log Loss | Max Prob |
|---|---|---|---|---|---|---|---|
| DISC Encoder | 4096 (SST-5) | 0.1 | 0.584 | 0.570 | 0.565 | 0.565 | 0.583 |
| | | 0.5 | 0.558 | 0.562 | 0.567 | 0.567 | 0.556 |
| | | 1.0 | 0.553 | 0.562 | 0.575 | 0.566 | 0.543 |
| | | 2.0 | 0.551 | 0.561 | 0.578 | 0.567 | 0.542 |
| | | 10.0 | 0.550 | 0.559 | 0.578 | 0.568 | 0.545 |
| | 4096 (HateSpeech) | 0.1 | 0.534 | 0.574 | 0.534 | 0.534 | 0.534 |
| | | 0.5 | 0.534 | 0.580 | 0.533 | 0.533 | 0.534 |
| | | 1.0 | 0.534 | 0.576 | 0.533 | 0.533 | 0.534 |
| | | 2.0 | 0.534 | 0.571 | 0.534 | 0.534 | 0.534 |
| | | 10.0 | 0.534 | 0.573 | 0.534 | 0.534 | 0.534 |
| | 4096 (AG News) | 0.1 | 0.554 | 0.658 | 0.611 | 0.611 | 0.553 |
| | | 0.5 | 0.558 | 0.657 | 0.613 | 0.613 | 0.556 |
| | | 1.0 | 0.568 | 0.662 | 0.616 | 0.616 | 0.561 |
| | | 2.0 | 0.577 | 0.663 | 0.620 | 0.619 | 0.566 |
| | | 10.0 | 0.541 | 0.663 | 0.625 | 0.624 | 0.569 |
| Generative (AR) | 4096 (SST-5) | 0.1 | 0.502 | 0.830 | 0.779 | 0.783 | 0.531 |
| | | 0.5 | 0.497 | 0.833 | 0.764 | 0.765 | 0.523 |
| | | 1.0 | 0.497 | 0.830 | 0.762 | 0.762 | 0.522 |
| | | 2.0 | 0.497 | 0.828 | 0.761 | 0.761 | 0.522 |
| | | 10.0 | 0.496 | 0.826 | 0.760 | 0.760 | 0.521 |
| | 4096 (HateSpeech) | 0.1 | 0.685 | 0.776 | 0.720 | 0.720 | 0.699 |
| | | 0.5 | 0.666 | 0.775 | 0.712 | 0.713 | 0.689 |
| | | 1.0 | 0.663 | 0.777 | 0.710 | 0.711 | 0.688 |
| | | 2.0 | 0.662 | 0.778 | 0.710 | 0.710 | 0.687 |
| | | 10.0 | 0.660 | 0.778 | 0.709 | 0.709 | 0.686 |
| | 4096 (AG News) | 0.1 | 0.794 | 0.885 | 0.818 | 0.818 | 0.803 |
| | | 0.5 | 0.751 | 0.894 | 0.806 | 0.807 | 0.788 |
| | | 1.0 | 0.742 | 0.894 | 0.803 | 0.804 | 0.785 |
| | | 2.0 | 0.737 | 0.895 | 0.802 | 0.802 | 0.783 |
| | | 10.0 | 0.732 | 0.895 | 0.801 | 0.801 | 0.782 |

Table 10: Effect of temperature scaling (i.e., dividing logits by a temperature parameter before the Softmax function) on discriminative vs. generative classifiers across datasets (SST-5, HateSpeech, AG News).The temperature value in the above tables denotes the scalar used to rescale logits prior to normalization. We observe that temperature scaling is less effective than logit clipping at reducing the vulnerability of generative classifiers; however, unlike clipping, temperature scaling does not degrade utility, as the F1 score remains stable across temperatures. Note that the susceptibility for GBM (Probs) is unchanged under temperature scaling, since temperature rescales logits by a constant factor and therefore constitutes only a linear transformation that preserves their separability.

|  | 128 | 256 | 512 | 1024 | 2048 | 4096 |
|---|---|---|---|---|---|---|
| DISC Encoder - TPR@FPR = 0.1 | | | | | | |
| Entropy | 0.047 | 0.078 | 0.168 | 0.227 | 0.191 | 0.153 |
| GBM (Logits) | 0.621 | 0.226 | 0.313 | 0.272 | 0.162 | 0.157 |
| GBM (Probs) | 0.638 | 0.232 | 0.314 | 0.275 | 0.181 | 0.160 |
| Ground Truth Predictions | 0.414 | 0.215 | 0.256 | 0.227 | 0.167 | 0.140 |
| Log Loss | 0.383 | 0.297 | 0.277 | 0.343 | 0.245 | 0.306 |
| Max Probability | 0.375 | 0.148 | 0.258 | 0.223 | 0.123 | 0.140 |
| Generative (AR) - TPR@FPR = 0.1 | | | | | | |
| Entropy | 0.539 | 0.473 | 0.730 | 0.628 | 0.613 | 0.480 |
| GBM (Logits) | 0.991 | 0.993 | 0.995 | 0.995 | 0.997 | 0.995 |
| GBM (Probs) | 0.797 | 0.886 | 0.951 | 0.914 | 0.866 | 0.636 |
| Ground Truth Predictions | 0.805 | 0.738 | 0.844 | 0.559 | 0.533 | 0.306 |
| Log Loss | 0.719 | 0.797 | 0.881 | 0.795 | 0.748 | 0.605 |
| Max Probability | 0.656 | 0.543 | 0.811 | 0.538 | 0.502 | 0.286 |

Table 11: TPR@FPR = 0.1 for Discriminative and Generative Classifiers Across Training Sizes

| Inference Calls ($n_{\text{infer}}$) | Entropy | GBM (Logits) | GBM (Probs) | Ground Truth Predictions | Log Loss | Max Probability |
|---|---|---|---|---|---|---|
| Discriminative (Encoder) Classifier | | | | | | |
| 128 | 0.531 (0.027) | 0.559 (0.050) | 0.545 (0.050) | 0.530 (0.031) | 0.530 (0.031) | 0.530 (0.028) |
| 256 | 0.530 (0.025) | 0.541 (0.039) | 0.541 (0.045) | 0.535 (0.027) | 0.535 (0.027) | 0.530 (0.025) |
| 512 | 0.529 (0.018) | 0.543 (0.016) | 0.534 (0.010) | 0.535 (0.020) | 0.535 (0.020) | 0.529 (0.018) |
| 1024 | 0.521 (0.010) | 0.534 (0.022) | 0.530 (0.012) | 0.528 (0.009) | 0.528 (0.009) | 0.521 (0.010) |
| 2048 | 0.520 (0.006) | 0.528 (0.022) | 0.525 (0.015) | 0.526 (0.006) | 0.526 (0.006) | 0.520 (0.006) |
| 4096 | 0.516 (0.004) | 0.526 (0.021) | 0.523 (0.014) | 0.522 (0.004) | 0.522 (0.004) | 0.516 (0.004) |
| Generative (Autoregressive) Classifier | | | | | | |
| 128 | 0.582 (0.040) | 0.581 (0.060) | 0.566 (0.010) | 0.576 (0.029) | 0.576 (0.029) | 0.577 (0.039) |
| 256 | 0.578 (0.059) | 0.582 (0.056) | 0.657 (0.096) | 0.586 (0.055) | 0.586 (0.055) | 0.584 (0.053) |
| 512 | 0.578 (0.027) | 0.571 (0.016) | 0.620 (0.095) | 0.601 (0.030) | 0.601 (0.030) | 0.585 (0.030) |
| 1024 | 0.576 (0.022) | 0.564 (0.019) | 0.618 (0.081) | 0.595 (0.026) | 0.595 (0.026) | 0.584 (0.024) |
| 2048 | 0.584 (0.017) | 0.572 (0.023) | 0.602 (0.080) | 0.596 (0.014) | 0.596 (0.014) | 0.587 (0.014) |
| 4096 | 0.584 (0.011) | 0.579 (0.027) | 0.604 (0.072) | 0.596 (0.012) | 0.596 (0.012) | 0.586 (0.012) |

Table 12: Performance Metrics across varying Inference Calls for Discriminative and Generative Classifiers for 12-layer models trained on 4096 sample setting for AG News dataset.

| Training Size | BERT (mean (std)) | GPT-2 (mean (std)) |
|---|---|---|
| 128 | 0.594 (0.086) | 0.827 (0.143) |
| 256 | 0.579 (0.068) | 0.827 (0.148) |
| 512 | 0.586 (0.040) | 0.917 (0.049) |
| 1024 | 0.592 (0.080) | 0.830 (0.132) |
| 2048 | 0.556 (0.038) | 0.773 (0.142) |
| 4096 | 0.569 (0.064) | 0.798 (0.141) |
| full_data | 0.551 (0.045) | 0.797 (0.179) |

Table 13: MIA Vulnerability measured via GBM (logits) attack model using AUROC on the AG News dataset.

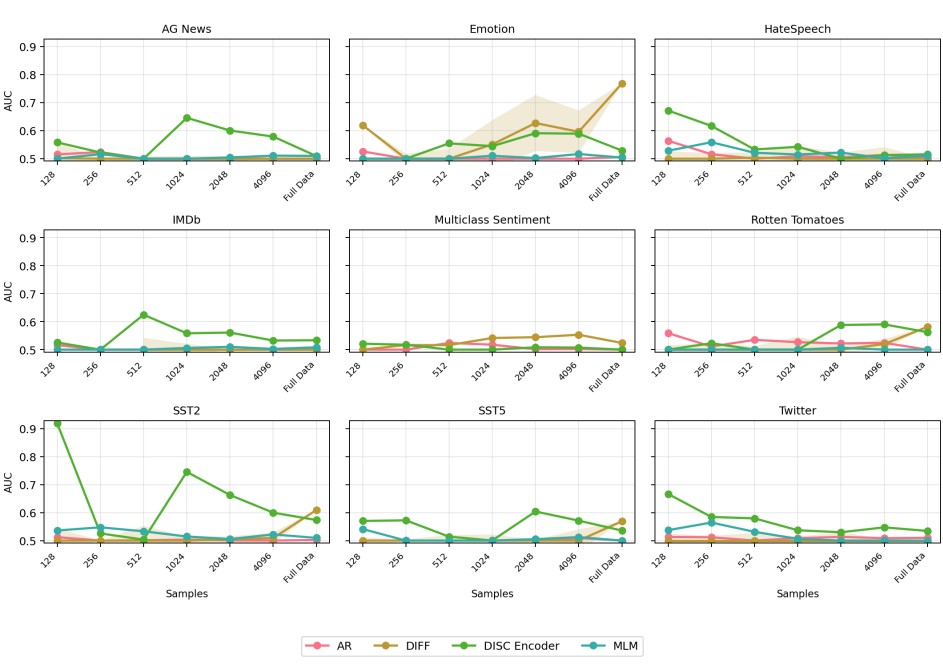

Figure 8: Attack susceptibility based on Entropy for model with 1 layer.

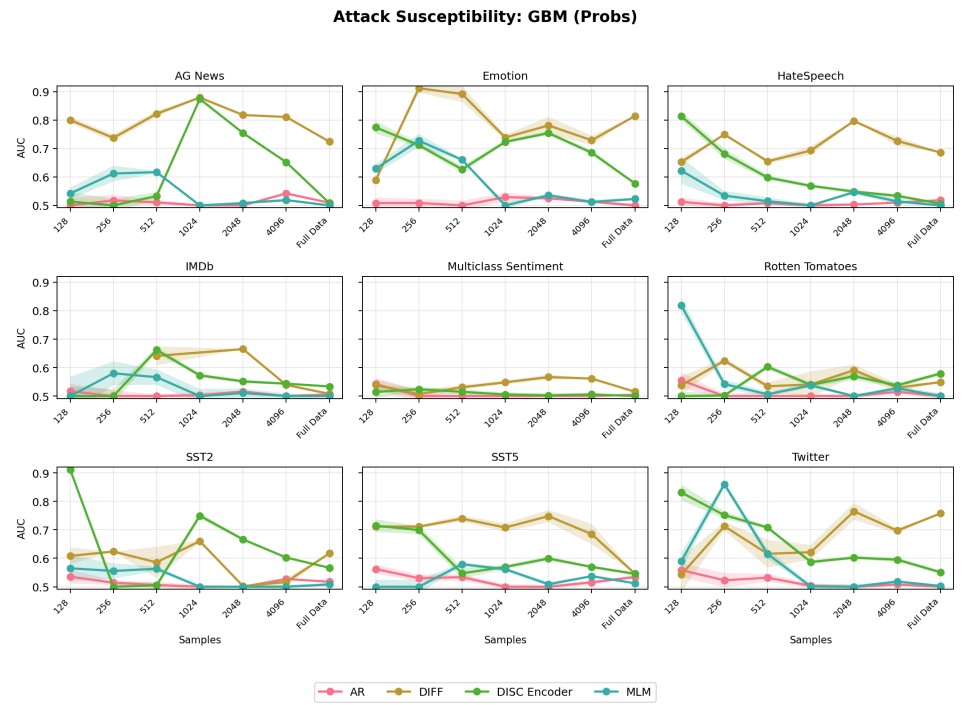

Figure 9: Attack susceptibility based on GBM (Probs) for model with 1 layer.

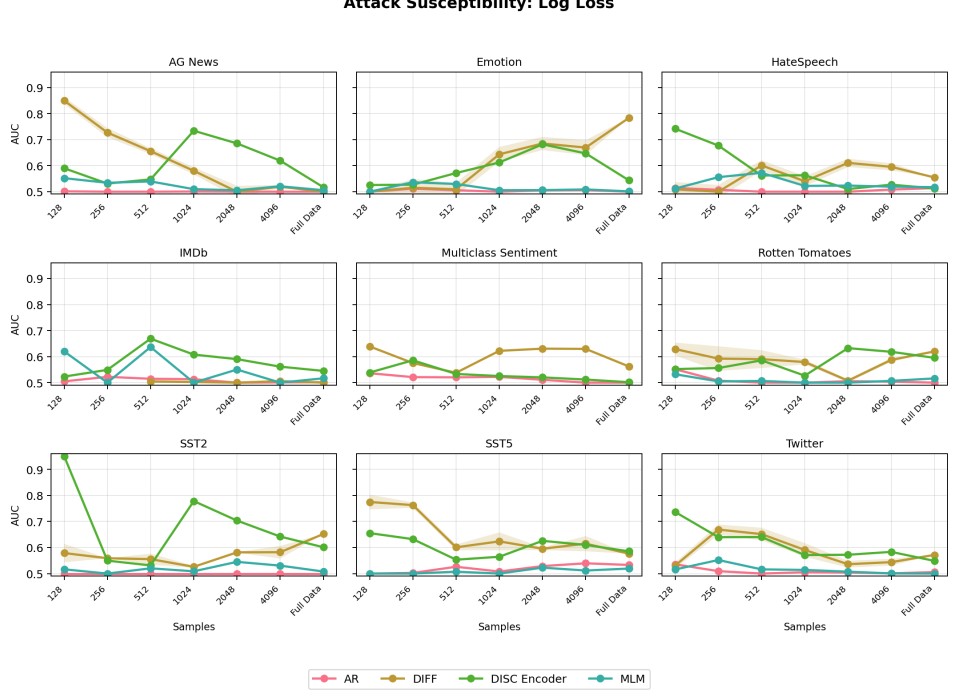

Figure 10: Attack susceptibility based on Log Loss for model with 1 layer.

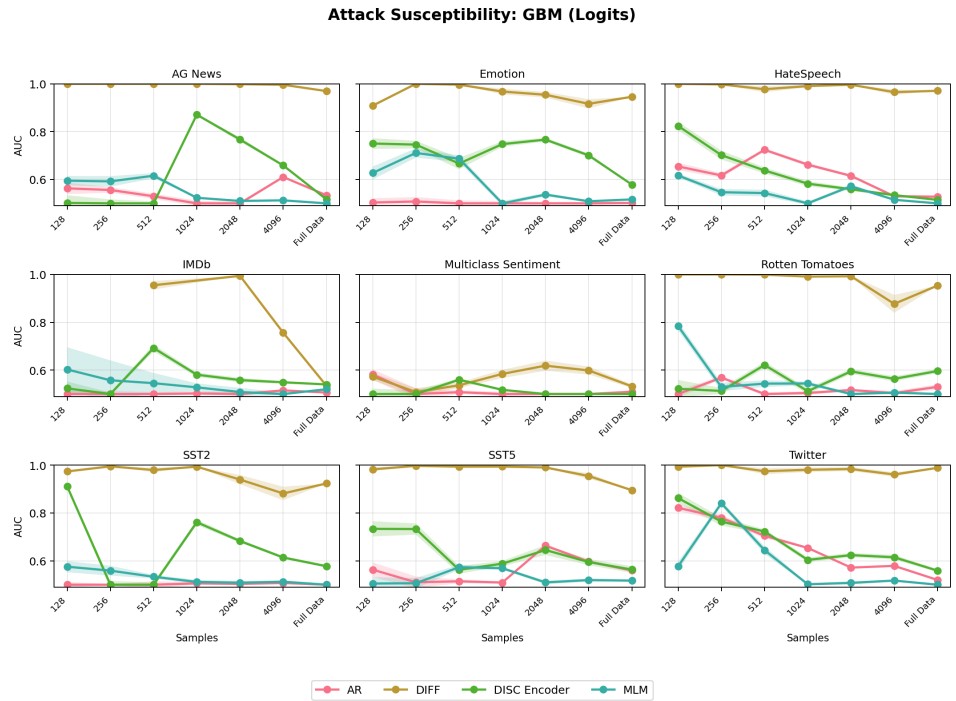

Figure 11: Attack susceptibility based on GBM (Logits) for model with 1 layer.

**Attack Susceptibility: Ground Truth Predictions**

Figure 12: Attack susceptibility based on Ground Truth Predictions for model with 1 layer.

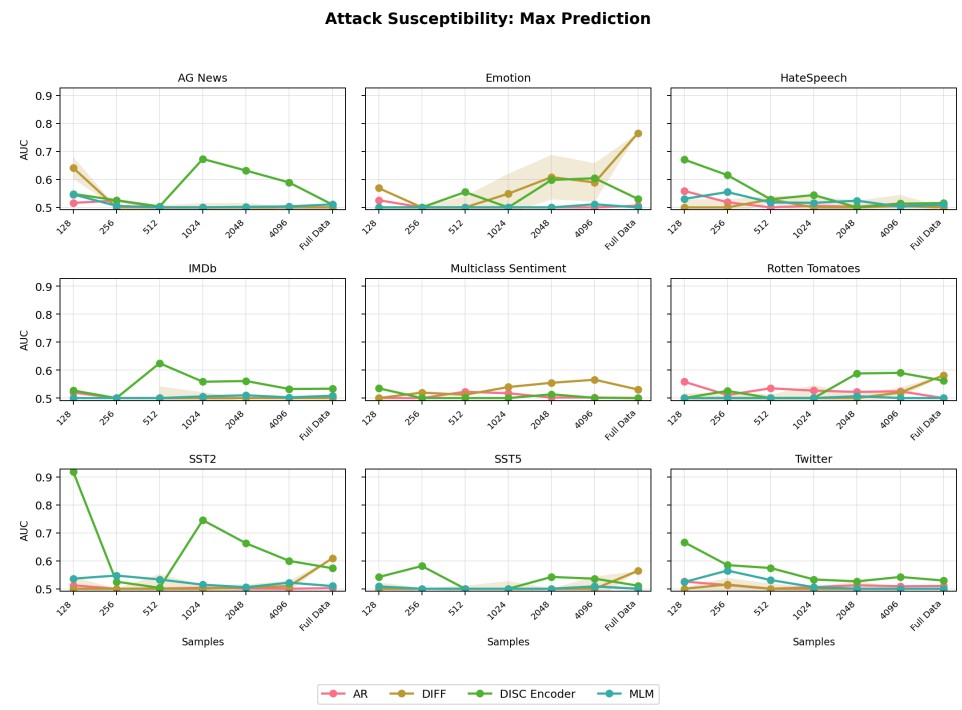

Figure 13: Attack susceptibility based on Max Prediction for model with 1 layer.

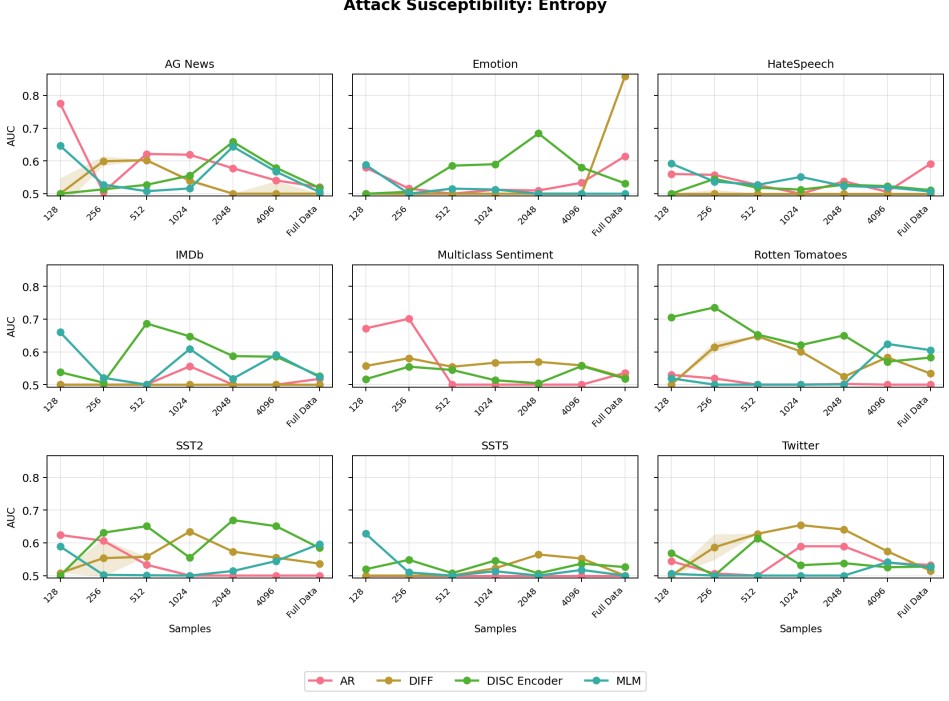

Figure 14: Attack susceptibility based on Entropy for model with 6 layers.

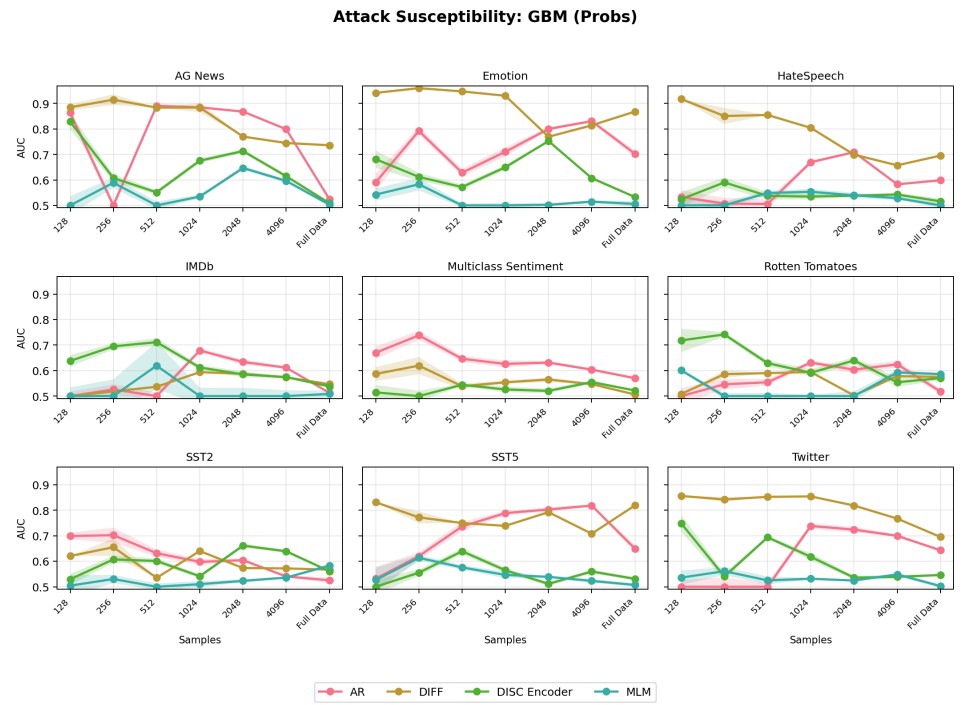

Figure 15: Attack susceptibility based on GBM (Probs) for model with 6 layers.

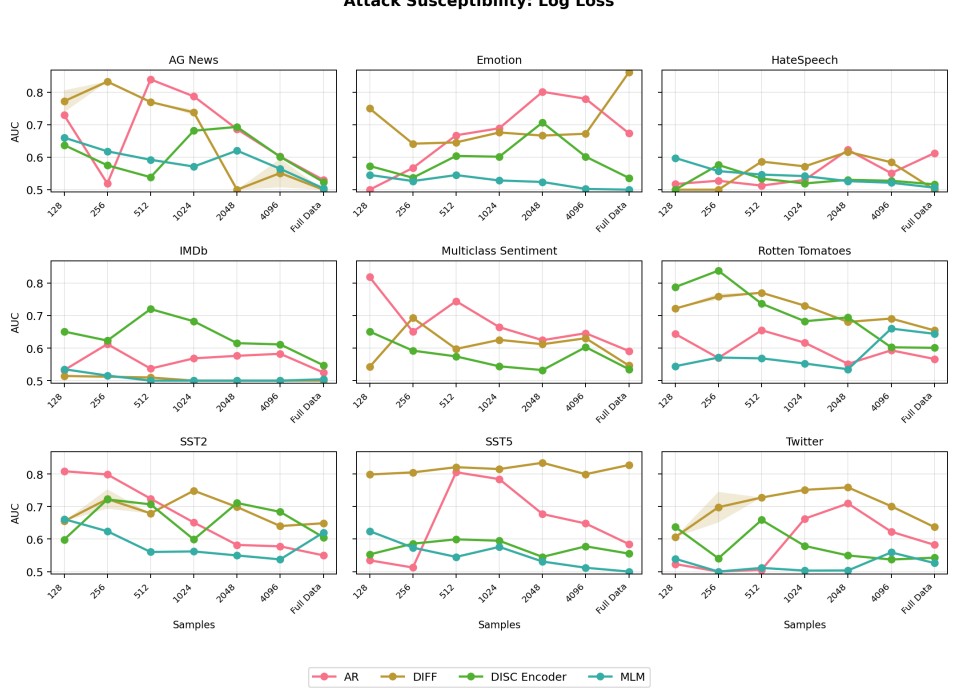

Figure 16: Attack susceptibility based on Log Loss for model with 6 layers.

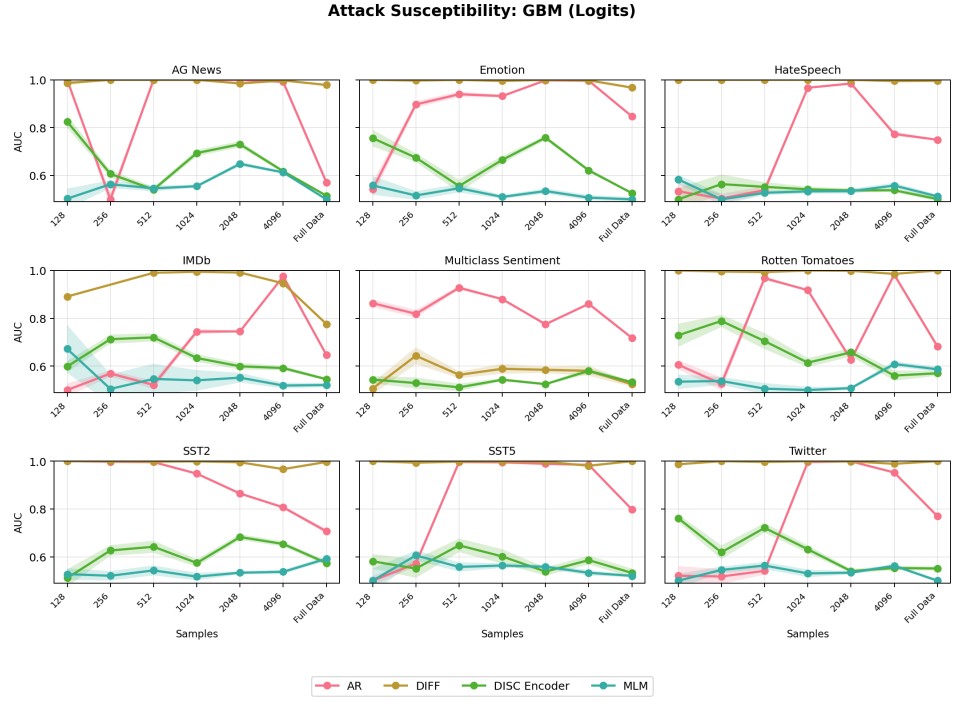

Figure 17: Attack susceptibility based on GBM (Logits) for model with 6 layers.

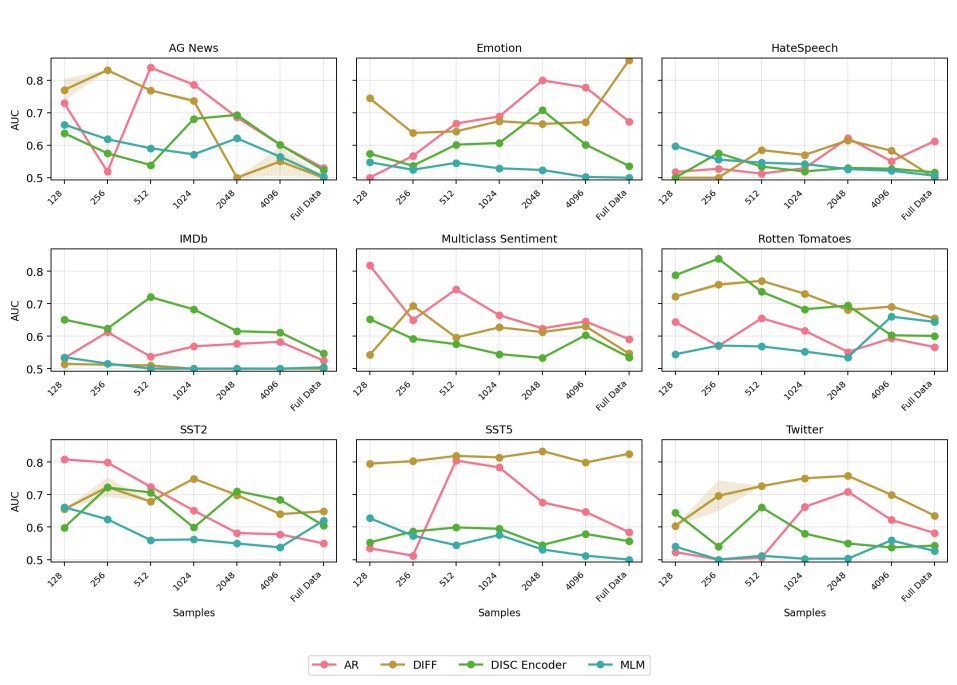

Figure 18: Attack susceptibility based on Ground Truth Predictions for model with 6 layers.

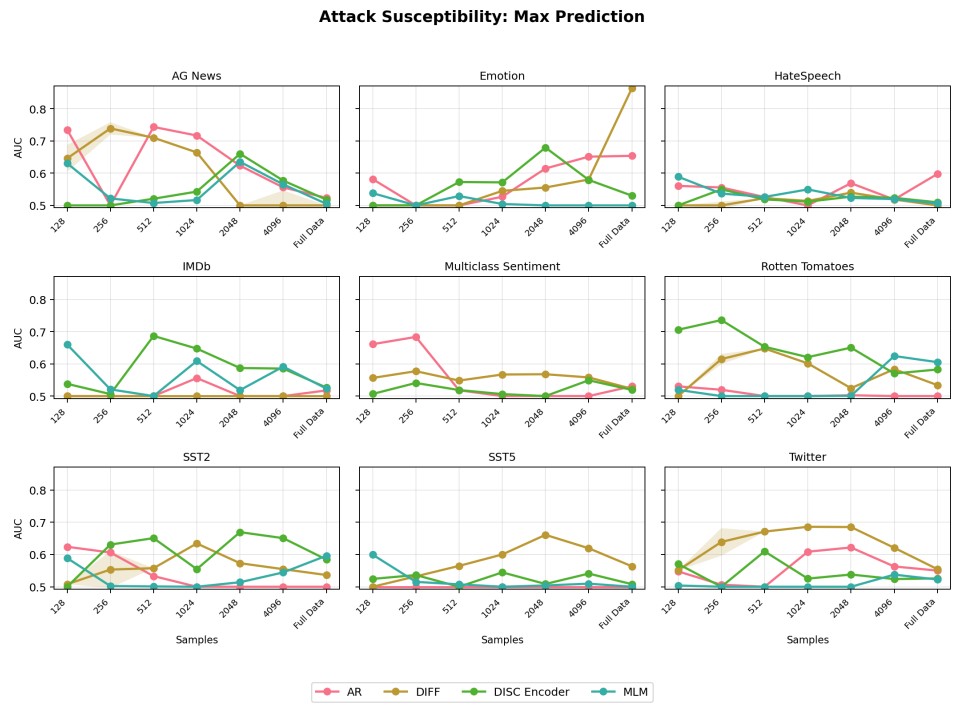

Figure 19: Attack susceptibility based on Max Prediction for model with 6 layers.

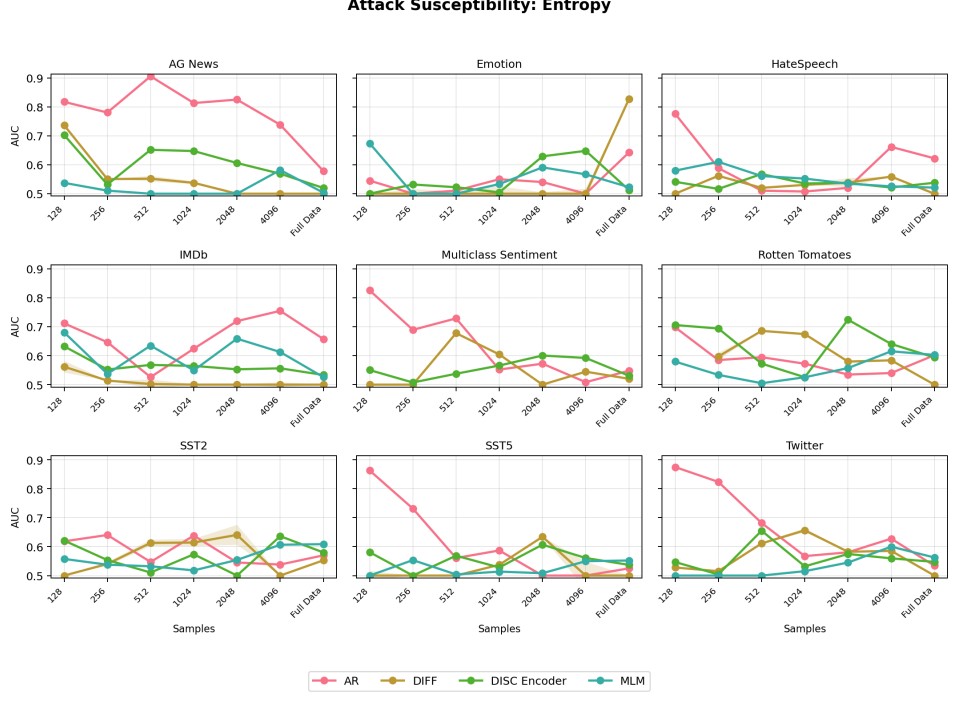

Figure 20: Attack susceptibility based on Entropy for model with 12 layers.

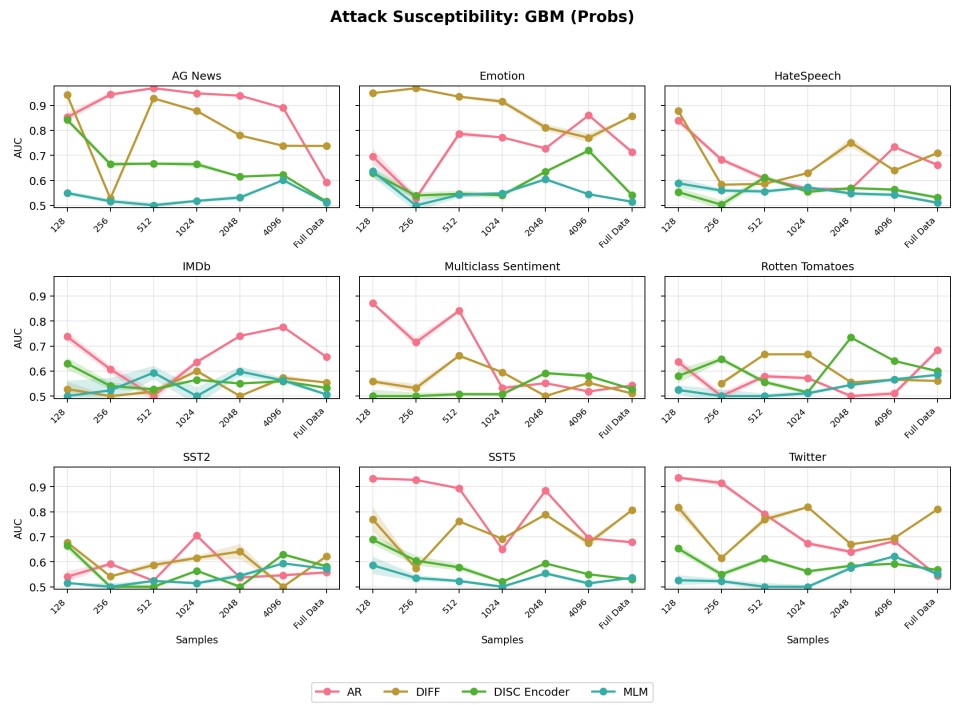

Figure 21: Attack susceptibility based on GBM (Probs) for model with 12 layers.

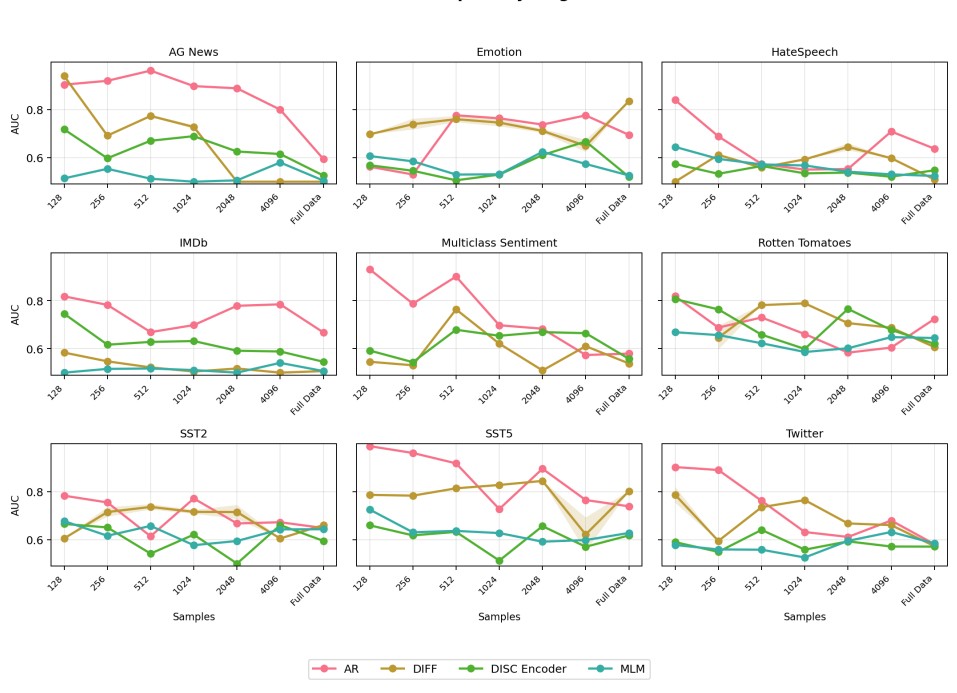

Figure 22: Attack susceptibility based on Log Loss for model with 12 layers.

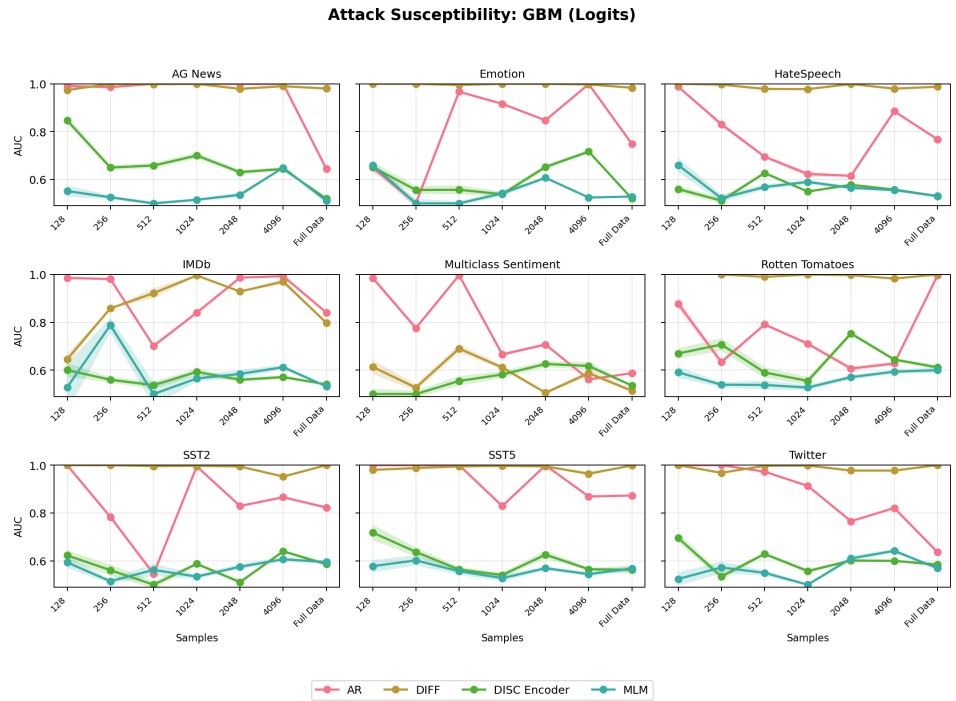

Figure 23: Attack susceptibility based on GBM (Logits) for model with 12 layers.

Figure 24: Attack susceptibility based on Ground Truth Predictions for model with 12 layers.

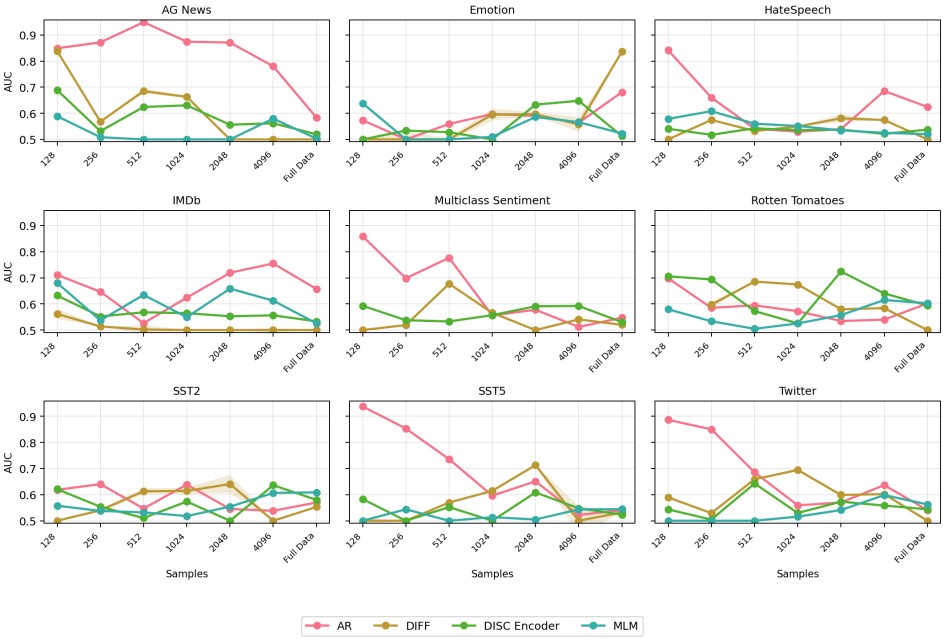

Figure 25: Attack susceptibility based on Max Prediction for model with 12 layers.

# G    TOY ILLUSTRATION

## G.1    EXPERIMENTAL SETUP

We study membership inference in a controlled synthetic setting where each input $x \in \mathbb{R}^d$ is composed of

$$x = \big[x_{\text{core}}, \ x_{\text{noise}}\big].$$

**Labels.**    Binary labels $y \in \{-1, +1\}$ are drawn from

$$P(y = +1) = w, \qquad P(y = -1) = 1 - w.$$

**Core feature (signal).**    The one-dimensional core feature correlates directly with the label:

$$x_{\text{core}} \sim \mathcal{N}\big(y \cdot \mu, \ \sigma^2\big),$$

where $\mu$ (`core_scale` in code) controls class separation and $\sigma$ controls within-class spread. *For the membership-inference experiments we match the train and test distributions*, i.e., we use the same $(\mu, \sigma)$ for both sets so that members and non-members are drawn i.i.d. from the same distribution.

**Noise features.**    The remaining $d - 1$ coordinates are independent Gaussian clutter:

$$x_{\text{noise}} \sim \mathcal{N}\big(0, \ \sigma_{\text{noise}}^2 I_{d-1}\big).$$

**Training/Test sizes and sweeps.**    We generate $n_{\text{train}}$ training samples and $n_{\text{test}}{=}4000$ test samples. We sweep

$$\mu \in \{0.05, 0.10, \ldots, 0.50\}, \quad n_{\text{train}} \in \{50, 200, 2000\}, \quad d \in \{16, 64, 256\},$$

and evaluate three class-prior settings $w \in \{0.1, 0.3, 0.5\}$. Unless stated otherwise, figures fix $w{=}0.5$, $\sigma_{\text{noise}}{=}1.0$, and $\sigma{=}0.15$.

| Parameter | Description |
|---|---|
| $w$ | Class prior for $y = +1$ (imbalance) |
| $\mu$ | Core mean shift (class separation) |
| $\sigma$ | Core feature standard deviation (train = test) |
| $\sigma_{\text{noise}}$ | Noise level for the $d-1$ nuisance dims |
| $d$ | Dimensionality (1 core + $d-1$ noise) |
| $n_{\text{train}}, n_{\text{test}}$ | Train/test sample counts |

Table 14: Synthetic data parameters. For MIA we use matched train/test distributions.

**Models and training.** We compare (i) Logistic Regression (LBFGS, `max_iter` = 10,000) and (ii) LDA (`solver=lsqr`, `shrinkage=auto`). Each configuration is run with 5 random seeds; we report means and shaded uncertainty bands.

## G.2 MOTIVATION

Our toy setup is designed to cleanly *tease apart* the drivers of membership inference without architectural or optimization confounds. By controlling a few interpretable knobs, we can test how membership signals scale with statistical difficulty:

- **Dimensionality ($d$):** Increasing $d$ adds nuisance directions and dilutes per-sample information, stressing generalization and potentially amplifying member–nonmember score gaps.
- **Sample size ($n$):** Larger $n$ reduces estimator variance and overfitting; smaller $n$ increases memorization pressure. The ratio $n/d$ serves as an effective *signal budget* per parameter.
- **Decision boundary separation ($\mu$):** Larger $\mu$ widens class separation, boosting accuracy and confidence. This lets us study whether membership advantage tracks confidence or generalization.
- **Signal strength ($\mu$ and $n/d$):** Together, geometric margin ($\mu$) and sample complexity ($n/d$) summarize how much reliable signal the model can extract relative to noise.
- **Imbalance (class weight $w$):** Varying the class prior via a weight $w \in (0,1)$ shifts the decision threshold and posterior calibration, directly affecting confidence-based and generative scores used by MIAs.

We keep train and test *i.i.d.* to isolate membership effects from distribution shift, and average over multiple seeds to separate systematic trends from randomness. This controlled regime exposes how membership advantage scales with $(d, n, \mu, w)$ and provides intuition that transfers to real datasets.

## G.3 MEMBERSHIP INFERENCE SCORES

For each trained model we compute member scores on the training set and non-member scores on an i.i.d. test set, and report AUROC.

**Max-probability (`auroc_prob`).** Given posterior estimates $\hat{p}(y \mid x)$,

$$s_{\text{prob}}(x) = \max_{y \in \{-1,+1\}} \hat{p}(y \mid x).$$

This is the standard, label-agnostic confidence attack we plot for both Logistic Regression and LDA.

**Log-joint (LDA only; `auroc_logjoint`).** For LDA with class priors $P(y)$, means $\mu_y$, and shared covariance $\Sigma$,

$$s_{\text{logjoint}}(x) = \max_y \left\{ \log P(y) + \log \mathcal{N}(x \mid \mu_y, \Sigma) \right\}.$$

This generative score often differs from max-probability and is shown in our AUROC plots.

### G.4 FINDINGS

**Protocol & visualization.** For each configuration $(\mu, n, d, w)$ we train Logistic Regression and LDA on i.i.d. train/test data with shared core variance $\sigma = 0.15$, noise level $\sigma_{\text{noise}} = 1.0$, and no spurious cue $(B = 0)$. We run 5 seeds and plot means with shaded bands showing $\pm 1.96 \times$ SEM. Membership is reported via AUROC and, when summarizing trends, the direction-invariant advantage $\text{AUROC} = \max\{\text{AUROC}, 1 - \text{AUROC}\}$. AUROC panels include a reference line at 0.5. The results are given in Figure 27.

**Notation.** LR/prob denotes the *max-probability (confidence)* score $s_{\text{prob}}(x) = \max_y \hat{p}(y \mid x)$ computed from a Logistic Regression model; LDA/prob is the same score computed from an LDA posterior; and LDA/log-joint denotes the *log-joint* score $s_{\text{logjoint}}(x) = \max_y \{\log P(y) + \log \mathcal{N}(x \mid \mu_y, \Sigma)\}$ from LDA. All are label-agnostic membership scores; unless stated, AUROC panels report the direction-invariant advantage AUROC.

**Dimensionality ($d$) and signal per parameter ($n/d$).** Holding $n$ fixed, increasing $d$ reduces test accuracy while *increasing* membership advantage. This is consistent with weaker signal per parameter ($n/d$): estimation error grows and models lean more on idiosyncrasies of the training set, widening member–nonmember score gaps. Across-seed variability (std) of both accuracy and AUROC *shrinks* as $d$ rises, indicating more concentrated (though worse) accuracy and a more consistently elevated membership signal in high dimensions.

**Geometric separation ($\mu$).** Larger $\mu$ (wider class separation) monotonically increases accuracy and also increases membership susceptibility: as margins grow, both models become more confident; training points attain slightly higher confidence (and, for LDA, higher log-joints) than i.i.d. test points, making member/nonmember scores easier to separate.

**Imbalance (class weight $w$).** Moving away from balance ($w \neq 0.5$) improves accuracy for both methods by shifting the optimal threshold toward the minority class. For membership, LR/prob exhibits a *dampened* susceptibility under imbalance—posteriors saturate toward the majority, compressing train–test score gaps—whereas LDA/prob remains comparatively stable and often higher in AUROC across $\mu$. Imbalance tends to increase across-seed variability, reflecting reduced effective sample size for the minority class.

**Generative vs. discriminative sample efficiency.** Even at $n = 50$, LDA substantially outperforms Logistic Regression in accuracy; this gap persists (and often widens) as $d$ increases (i.e., smaller $n/d$), reflecting the classic sample-efficiency advantage of a correctly specified generative model with shrinkage.

**LDA/log-joint vs. LDA/prob.** Across essentially all $(d, n, \mu, w)$, LDA/log-joint yields higher AUROC than LDA/prob. The log-joint exposes modeled density scale: training points lie closer to estimated class means and receive larger $\log p(x \mid y)$, hence larger $\log P(y) + \log p(x \mid y)$, than i.i.d. test points. Posteriors $\hat{p}(y \mid x)$ partially compress this scale information, making LDA/prob consistently less susceptible. The gap typically widens as $d$ increases or $n/d$ decreases, underscoring the added risk of releasing joint/likelihood values.

**LDA/prob vs. LR/prob across separation.** At small $\mu$, LR/prob shows both *lower* accuracy and *lower* membership susceptibility than LDA/prob, matching LDA's sample-efficiency advantage when $n/d$ is small. As $\mu$ grows, LR/prob confidence rises steeply with margin and its AUROC increases; it can meet or exceed LDA/prob at larger separations. Under stronger imbalance, this rise is *dampened* for LR/prob, while LDA/prob remains comparatively high.

**LDA/log-joint vs. LR/prob.** Except in a single benign regime (balanced $w = 0.5$, good separation $\mu$, and low $d$), LDA/log-joint exceeds LR/prob in membership advantage. Practical takeaway: even when discriminative posteriors appear relatively less susceptible, exposing generative joint/likelihood scores can be markedly more revealing.

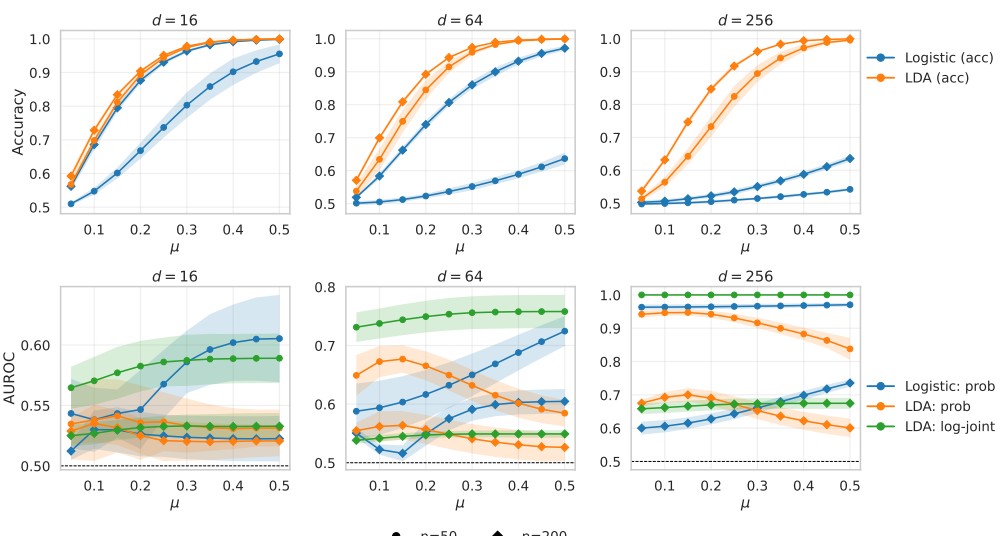

Figure 26: **Mean ± SEM across 5 seeds.** Top row: test accuracy vs. core separation $\mu$. Bottom row: membership $\mathrm{Adv}(\mathrm{AUROC})$ vs. $\mu$. Columns correspond to $d \in \{16, 64, 256\}$. Within each panel, color denotes series (Accuracy: Logistic/LDA; AUROC: Logistic max-prob, LDA max-prob, LDA log-joint), and marker denotes $n_{\mathrm{train}} \in \{50, 200, 2000\}$. We fix $w=0.5$, $B=0$, $\sigma=0.15$, and $\sigma_{\mathrm{noise}}=1.0$.

**Summary.** Stronger signal (larger $\mu$, larger $n/d$) improves accuracy but also strengthens confidence-based membership cues; higher $d$ at fixed $n$ hurts accuracy yet sharpens membership separation. Explicit prior modeling amplifies accuracy gains under imbalance without a commensurate reduction in susceptibility. Generative LDA is more sample-efficient than LR, and its log-joint scores are the most vulnerable among the considered outputs. Theorem 3.3 provides sufficient (not necessary) conditions for dominance, specifically that the marginal skew must exceed the conditional skew by a factor determined by the bounded likelihood-ratio condition. When this inequality does not hold, the dominance can reverse.

Our empirical results directly illustrate such counter-examples. In the toy LDA vs. logistic regression experiments (Fig. 1), for low class separation, moderate sample size $n = 200$, and higher dimensions $d = 64, 256$, the LDA log-joint signal is less vulnerable than the LDA posterior-probability signal (green curve lying below the orange one). This happens because in this regime LDA learns the conditional decision boundary reasonably well, while the marginal density is poorly estimated. This is exactly the situation where the premise fails. As expected from our theory, the joint score does not dominate in this regime.

### G.5 FINDINGS UNDER MODEL MISSPECIFICATION

**Contamination model.** We introduce misspecification through Huber-style $\varepsilon$–contamination (Huber, 1992; Kasa & Rajan, 2023) by *replacing* each example (independently in train and test) with probability $\varepsilon$ by an isotropic high-variance draw that is independent of the label:

$$X \sim \begin{cases} \text{clean generator (core/spurious/noise)} & \text{w.p. } 1 - \varepsilon, \\ \mathcal{N}(0, \tau^2 I_d) & \text{w.p. } \varepsilon, \end{cases} \quad \text{with } \tau = \texttt{tau\_mult} \cdot \sigma_{\mathrm{noise}}.$$

We keep the label $y$ unchanged. In our runs we use $\varepsilon = 0.02$ and $\texttt{tau\_mult} = 10$, yielding empirical contamination rates $\approx 2.2\%$ in train and $\approx 2.0\%$ in test on average (diagnostics in the CSV).

**Protocol.** Except for the contamination replacement above, the setup matches the clean case: for each $(\mu, n, d, w)$ we train Logistic Regression and LDA; we fix $\sigma = 0.15$, $\sigma_{\mathrm{noise}} = 1.0$, and $B = 0$; we average over 5 seeds and summarize membership with AUROC and its direction-invariant

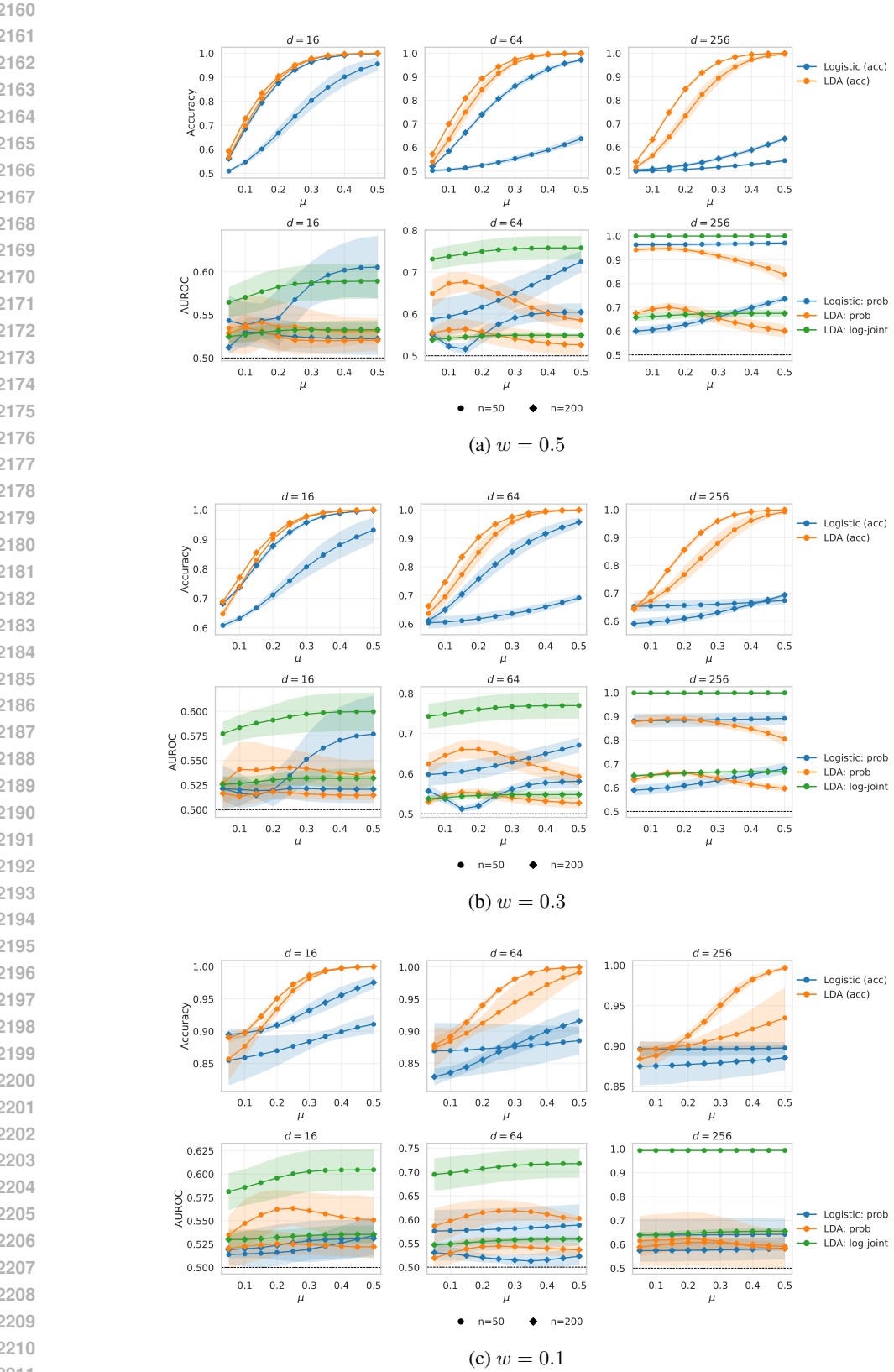

Figure 27: **Mean ± SEM across 5 seeds.** The three subfigures correspond to varying degree of imbalance, with $w = 0.5$ corresponding to the balanced case. Each subfigure shows: top row = test accuracy vs. $\mu$, bottom row = MIA (AUROC) vs. $\mu$; columns are $d \in \{16, 64, 256\}$. Markers denote $n_{\text{train}} \in \{50, 2000\}$. Within each panel, color denotes series (Accuracy: Logistic/LDA; AUROC: Logistic max-prob, LDA max-prob, LDA log-joint)

advantage $\text{AUROC} = \max\{\text{AUROC}, 1 - \text{AUROC}\}$ for the three scores LR/prob, LDA/prob, and LDA/log-joint. The results are given in Figure 28.

**Generative vs. discriminative under misspecification.** Contamination reverses LDA's clean-data sample-efficiency edge in accuracy—Logistic is typically better—because a few large-norm replacements strongly distort shared-covariance estimation even with shrinkage. However, exposing density scale remains risky: LDA/log-joint is the most susceptible membership score across most regimes we tested, particularly at high $d$ and small $n$.

The introduction of misspecification through contamination depresses accuracy overall and especially at higher $d$; accuracy increases with geometric separation $\mu$ and with sample size $n$. Under contamination the discriminative model is more resilient than LDA: averaged across the grid, Logistic attains $\sim 0.775$ vs. LDA $\sim 0.742$ mean accuracy. By dimension, accuracy drops from $(d{=}16)$ to $(d{=}256)$ for both methods (e.g., LDA: $0.76 \rightarrow 0.72$, Logistic: $0.83 \rightarrow 0.73$), consistent with inflated covariance estimates and leverage effects from large-norm points.

Contamination *amplifies* member–nonmember score gaps, with stronger effects at larger $d$, smaller $n$, and larger $\mu$. Both posterior-based signals rise with $d$ , and LDA/log-joint is consistently the most revealing . At very small sample sizes ($n{=}50$) the advantage is largest ; by $n{=}2000$ these fall back toward chance .

As we move from extreme imbalance ($w{=}0.1$) toward balance ($0.5$), accuracy decreases (less prior help), while membership susceptibility *increases* for all three signals (e.g., LR/prob mean AUROC $\approx 0.54 \rightarrow 0.61$, LDA/prob $\approx 0.54 \rightarrow 0.58$, LDA/log-joint $\approx 0.58 \rightarrow 0.59$), echoing the dampening effect of imbalance on confidence-based MIAs in the clean setting.

**Summary.** Replacing a small fraction of points by high-variance, label-independent outliers simultaneously hurts accuracy and strengthens membership signals, with the sharpest increases at larger $d$, smaller $n$, and larger $\mu$. While Logistic is more robust in accuracy, releasing generative *log-joint/likelihood* values (LDA/log-joint) is notably more revealing than posteriors, reinforcing the recommendation to avoid exposing such scores under potential contamination.

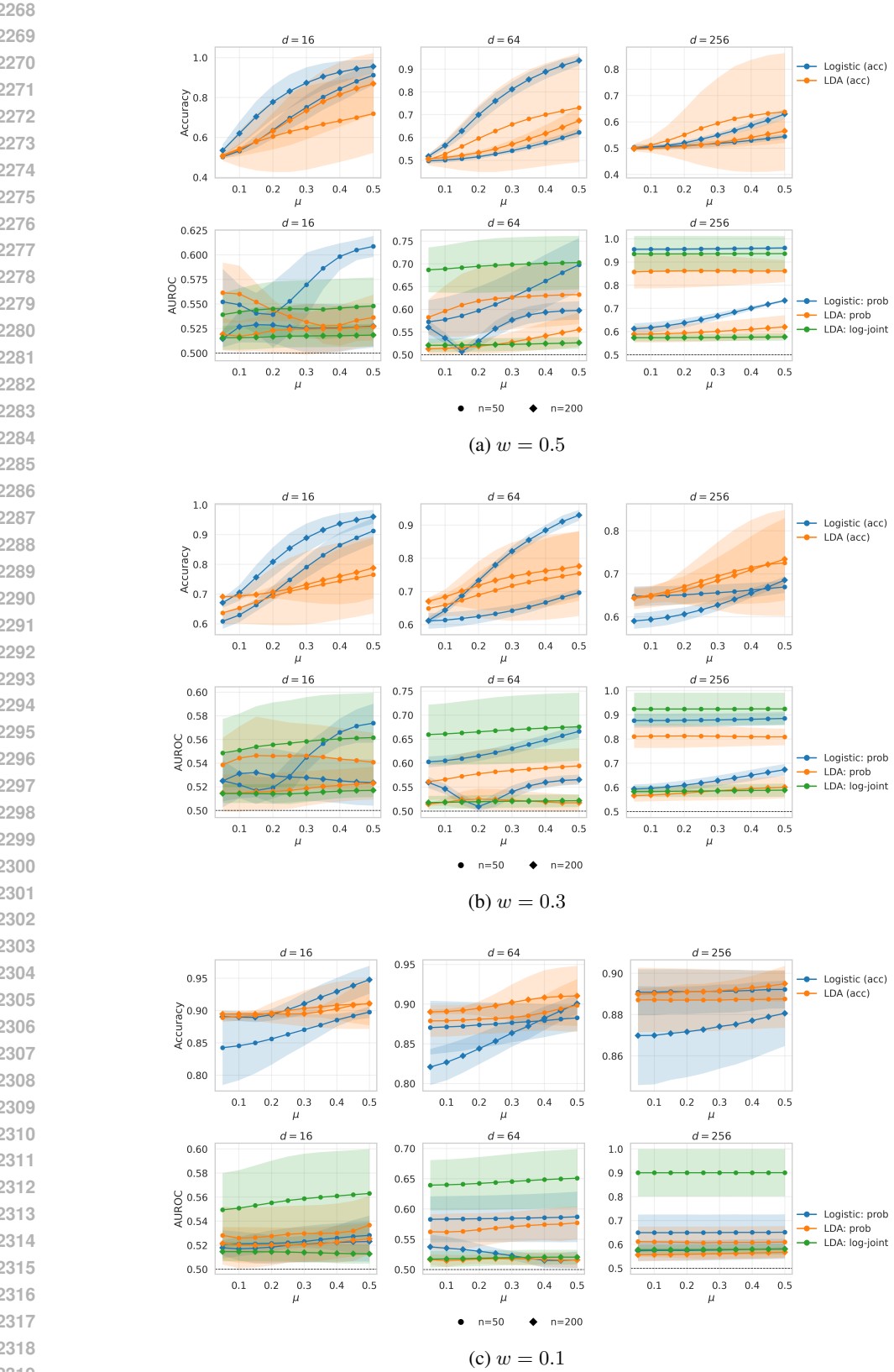

Figure 28: **Mean $\pm$ SEM across 5 seeds.** The three subfigures correspond to varying degree of imbalance, with $w = 0.5$ corresponding to the balanced case. Each subfigure shows: top row = test accuracy vs. $\mu$, bottom row = MIA (AUROC) vs. $\mu$; columns are $d \in \{16, 64, 256\}$. Markers denote $n_{\text{train}} \in \{50, 2000\}$. Within each panel, color denotes series (Accuracy: Logistic/LDA; AUROC: Logistic max-prob, LDA max-prob, LDA log-joint)

## H PRIVACY-UTILITY ANALYSIS

We conducted a comprehensive privacy-utility analysis examining how privacy vulnerabilities change with model architecture and training data characteristics. Our analysis focused on four key strategies: ENC (Encoder/DISC), AR (Autoregressive), MLM (Masked Language Model), and DIFF (Diffusion), evaluating their susceptibility to Gradient Boosting Machine (GBM) based membership inference attacks.

### H.1 METHODOLOGY

The analysis examined privacy-utility trade-offs across different model configurations, specifically investigating:

- **Model Size Impact**: Varying the number of transformer layers (1, 6, 12)
- **Training Data Size**: Different sample counts (128, 256, 512, 1024, 2048, 4096, Full Data)
- **Attack Methods**: GBM-based attacks using logits and probability distributions
- **Utility Metric**: F1 scores across multiple text classification datasets

### H.2 RESULTS

#### H.2.1 MODEL SIZE ANALYSIS

Figure 29 presents the privacy-utility trade-offs as a function of model size. Each point represents the average performance across datasets, with layer annotations (L1, L6, L12) indicating the model depth.

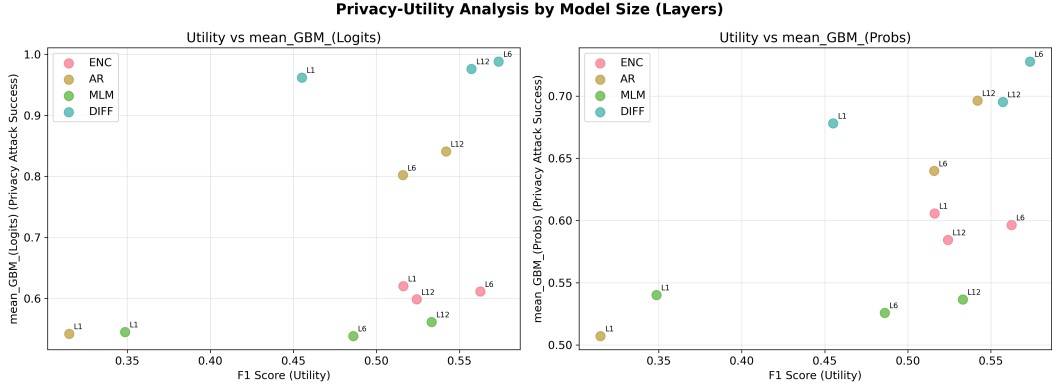

Figure 29: Privacy-utility trade-offs by model size across four strategies. Left panel shows GBM Logits attack success vs. F1 utility scores. Right panel shows GBM Probs attack success vs. F1 utility scores. Lower attack success indicates better privacy protection.

#### H.2.2 TRAINING SAMPLE SIZE ANALYSIS

Figure 30 illustrates how training data size affects the privacy-utility balance. Sample size annotations indicate the number of training examples used.

### H.3 KEY FINDINGS

#### H.3.1 STRATEGY PERFORMANCE RANKING

Our analysis reveals significant differences in privacy-utility characteristics across strategies:

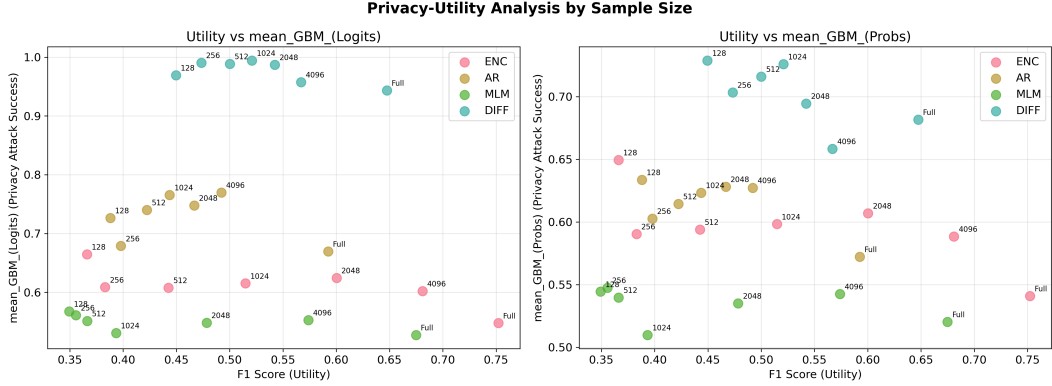

Figure 30: Privacy-utility trade-offs by training sample size across four strategies. Left panel shows GBM Logits attack success vs. F1 utility scores. Right panel shows GBM Probs attack success vs. F1 utility scores. Sample size annotations indicate training data volume.

**Utility Performance (F1 Scores):**

1. **ENC**: 0.534 (±0.233) – Best overall utility performance
2. **DIFF**: 0.529 (±0.171) – Second best with highest consistency
3. **AR**: 0.458 (±0.210) – Moderate utility with high variance
4. **MLM**: 0.456 (±0.238) – Lowest utility but improving with model size

**Privacy Vulnerability (Attack Success Rates):** For GBM Logits attacks (lower values indicate better privacy protection):

1. **MLM**: 0.548 (±0.056) – Best privacy protection
2. **ENC**: 0.610 (±0.086) – Good privacy protection
3. **AR**: 0.728 (±0.198) – Moderate vulnerability
4. **DIFF**: 0.976 (±0.058) – Highest vulnerability

### H.3.2 MODEL ARCHITECTURE IMPACT

The relationship between model size and privacy-utility trade-offs varies significantly across strategies:

- **ENC Strategy**: Demonstrates optimal balance with utility peaking at 6 layers (F1=0.562) while privacy protection improves with model depth. Attack success rates decrease from 0.620 to 0.599 (GBM Logits) as layers increase from 1 to 12.

- **MLM Strategy**: Shows the most favorable privacy characteristics with consistent protection across all model sizes. Utility improves substantially with depth ($0.349 \rightarrow 0.533$) while maintaining the lowest attack success rates.

- **AR Strategy**: Exhibits concerning behavior where utility gains ($0.315 \rightarrow 0.542$) come at severe privacy cost, with attack success rates increasing dramatically ($0.542 \rightarrow 0.841$) for larger models.

- **DIFF Strategy**: Despite achieving good utility, consistently shows the highest vulnerability to privacy attacks ($> 95\%$ success rate) across all configurations, making it unsuitable for privacy-sensitive applications.

### H.4 RECOMMENDATIONS

Based on our comprehensive analysis, we provide the following recommendations:

- **General Applications**: Use ENC strategy with 6-12 layers for optimal privacy-utility balance
- **Privacy-Critical Systems**: Deploy MLM strategy with 12 layers for maximum privacy protection
- **High-Risk Scenarios**: Avoid DIFF strategy due to severe privacy vulnerabilities
- **AR Strategy Caution**: Monitor privacy implications carefully when scaling AR models
- **Logit-Clipping** does reduce the susceptible of generative classifiers to MIA but it comes at the cost of reduced performance.

The analysis demonstrates that privacy and utility considerations must be carefully balanced when selecting model architectures and training strategies, with ENC and MLM strategies offering the most favorable trade-offs for privacy-preserving applications.

## I  LIMITATIONS

Despite providing the first systematic analysis of MIAs across *Discriminative*, *Generative*, and *Pseudo-Generative* text classifiers, our study has several limitations. First, our experiments are conducted under standard i.i.d. assumptions, and the results may not generalize to real-world scenarios involving distribution shifts, such as covariate or concept drift (Bickel et al., 2009; Roychowdhury et al., 2024), where both attack success and classifier behavior could differ substantially. Second, we limit our study to only black-box attacks; it would be interesting to study if the same findings translate to white-box attacks on generative classifiers, which we leave for future work. Third, we focus on transformer-based architectures with conventional fine-tuning, omitting emerging paradigms such as few-shot or prompt-based in-context learning (Sun et al., 2023; Gupta et al., 2023), as well as parameter-efficient adaptation techniques like LoRA (Hu et al., 2022), which may exhibit different privacy-utility trade-offs. Fourth, our analysis is restricted to text classification; multi-modal data—including tabular, visual, or audio modalities (Pattisapu et al., 2025; Lu et al., 2019; Kushwaha & Fuentes, 2023)—may yield distinct membership leakage patterns due to richer or correlated feature structures. Fourth, we primarily evaluate standard MIA strategies and do not explore fully adaptive adversaries that could exploit model-specific quirks, ensemble behaviors, or auxiliary side information. Finally, while we study training data volume as a factor influencing vulnerability, other aspects such as pretraining data composition, model calibration, or data augmentation strategies may also impact privacy risks but remain unexplored. These limitations suggest that while our findings provide foundational insights, extending analyses to diverse settings and adaptive attacks is necessary to fully understand and mitigate privacy risks in generative classification systems.

