# OpenReview forum: "The Hidden Cost of Modeling $\text{P}(X)$: Membership Inference Attacks in Generative Text Classifiers"
_ICLR.cc/2026/Conference — Submitted to ICLR 2026_

### Official Review · Reviewer_8KoX · 2025-10-30

**Soundness:** 3
**Presentation:** 3
**Contribution:** 3
**Rating:** 6
**Confidence:** 2

**Summary:**

The paper discussed the vulnerabilities of different types of models to membership inference attacks (MIAs). The main finding is that generative classifiers are more vulnerable to MIAs compared to discriminative classifiers. This is because generative models leak information through both the marginal distribution P(X) and the conditional distribution P(Y|X). The authors conducted empirical evaluations across various model architectures and datasets to validate their claims.

**Strengths:**

- Privacy is an important concern.
- The paper is generally well-written.

**Weaknesses:**

- The conclusions drawn in the paper seem somewhat expected given the fundamental differences between generative and discriminative models.
- How the findings would generalize to more complex scenarios, such as using multiple shadow models for MIAs, is not clear.

**Questions:**

In my opinion, the problem studied in this paper hasn't been extensively studied because generative models and discriminative models have different outputs. Discriminative models output class probabilities, while generative models output likelihoods over the entire input space, which provides much richer information. Therefore, it is somewhat expected that generative models would be more vulnerable to membership inference attacks. Let's assume an extreme case where the decision space of the discriminative model is as large as the input space of the generative model. In this case, wouldn't the vulnerability of the discriminative model be comparable to that of the generative model?

The authors make an assumption that only one shadow model is trained for the membership inference attack. However, in practice, state-of-the-art membership inference attacks often involve training multiple shadow models to better approximate the target model's behavior. How would the results change if multiple shadow models were used instead of just one?

---

> ### Author Response · Authors · 2025-11-21
> **Response to Reviewer 8KoX**
>
> **1. it is somewhat expected that generative models would be more vulnerable to membership inference attacks. Let's assume an extreme case where the decision space of the discriminative model is as large as the input space of the generative model. In this case, wouldn't the vulnerability of the discriminative model be comparable to that of the generative model?**
>
> We agree with the reviewer that discriminative models typically operate over a small label space (often <10 classes), whereas generative models parameterize likelihoods over the entire input or vocabulary space. This is indeed a meaningful asymmetry, and we appreciate the reviewer highlighting this intuition.
>
> However, if output-space cardinality alone were the dominant factor explaining vulnerability, then pseudo-generative models such as masked language models (MLMs), which estimate token-wise pseudo-likelihoods over the same high-cardinality vocabulary as autoregressive models, should also exhibit comparably high susceptibility to membership inference attacks. Contrary to this expectation, our experiments consistently show that MLMs remain remarkably resistant to membership inference, despite operating in an output space of identical dimensionality to that of generative models.
>
> This divergence suggests that the core issue is not merely output-space size, but rather how likelihood is structured and allocated across the space during training. Autoregressive models assign joint likelihood over full sequences and thus tightly couple training examples to the model’s probability surface. In contrast, pseudo-likelihood objectives (e.g., MLM) only model marginal token distributions under heavy conditioning noise, which strongly restricts the model’s capacity to overfit specific training examples, even though the output vocabulary is equally large.
> This empirical evidence supports our central claim: the generative training objective itself, drives vulnerability, rather than output dimensionality alone.
>
>
> **2. The authors make an assumption that only one shadow model is trained for the membership inference attack. However, in practice, state-of-the-art membership inference attacks often involve training multiple shadow models to better approximate the target model's behavior. How would the results change if multiple shadow models were used instead of just one?**
>
> We thank the reviewer for this important question. We clarify that the single-shadow model assumption appears only in our theoretical analysis (Lines 74, 143, 806) and is used strictly to simplify the mathematical exposition. This is fully consistent with standard MIA formulations following Salem et al. (2018) and Yeom et al. (2018), which showed that (i) a single shadow model is sufficient to approximate the target, and (ii) zero-shadow attacks (e.g., loss-based attacks) are often just as effective. Our theory requires only one reference distribution $Q$ in order to compare the induced score laws $P_S$​ and $Q_S$​; training multiple shadow models would approximate the same population distribution and therefore does not alter our decomposition or the resulting bounds.
>
>
> Crucially, our empirical evaluations do not rely on shadow models at all. As is now common in recent MIA work, including Yeom et al. (2018), Carlini et al. (2019), Shejwalkar et al. (2021), and Song & Mittal (2021), we directly compare the model’s outputs on training samples (members) and test samples (non-members) and compute AUROC. This approach measures the true separability of member vs. non-member score distributions and does not introduce additional approximation noise from training surrogate models. Because our attacks operate directly on the target model’s outputs, using multiple shadow models would not change the AUROC-based conclusions: the observed leakage stems from structural differences between modeling $P(X)$ and $P(Y∣X)$, not from the number of shadow models available to the attacker.
>
>
> Finally, from a practical standpoint, our study trains over 2,900 models across architectures, datasets, and data-size settings. Training additional shadow models for every configuration would significantly multiply computational cost without changing the scientific conclusions.
>
>
> We will add a concise clarification in the final version explaining the distinction between theoretical single-shadow modeling and empirical zero-shadow evaluation, and citing prior work demonstrating the sufficiency of zero-shadow attacks.

---

> ### Author Response · Authors · 2025-11-28
> **Response to Reviewer 8KoX**
>
> Dear Reviewer **8KoX**,
>
> Thank you again for your thoughtful and constructive feedback on our submission.
> We wanted to follow up during the discussion phase to check whether you had any remaining questions for us.
>
> Since your last comment, we have uploaded a revised PDF with all changes highlighted in **blue**. These revisions directly address the issues you raised, specifically the clarification on single shadow models (in Appendix A2)
>
> We would be very happy to elaborate further if anything remains unclear.
> If not, we would greatly appreciate hearing your updated perspective now that your concerns have been addressed. Your input at this stage is extremely valuable for the final decision process.
>
> Thank you again for your time and engagement.
>
> **Authors of submission #19645**

---

### Official Review · Reviewer_C54N · 2025-10-30

**Soundness:** 2
**Presentation:** 3
**Contribution:** 3
**Rating:** 4
**Confidence:** 5

**Summary:**

This paper presents the first systematic study of membership inference attack (MIA) vulnerability in generative text classifiers, contrasting them with discriminative and pseudo-generative models. The authors theoretically and empirically demonstrate that explicitly modeling the data distribution P(X) significantly increases privacy risks.

**Strengths:**

The first comprehensive investigation of MIA risks across generative, discriminative, and pseudo-generative paradigms.
And the paper has a strong theoretical foundation. The two-way decomposition of total variation/KL divergence into marginal and conditional components elegantly explains why modeling P(X) increases leakage.
Provides actionable recommendations (avoid logits exposure, use pseudo-generative architectures) directly relevant to ML-as-a-Service providers and API designers.

**Weaknesses:**

A key conceptual limitation of this paper is the task mismatch between generative and discriminative classifiers.
Although both are evaluated on text classification benchmarks, they optimize for fundamentally different goals: discriminative models (e.g., BERT) learn decision boundaries for P(Y|X), while generative models (e.g., autoregressive or diffusion) learn to reconstruct or generate text under P(X,Y).
As a result, generative models face a more complex objective involving sequence likelihoods and token-level prediction, making direct privacy comparisons uneven and potentially misleading.
Their higher MIA vulnerability may partly reflect this task complexity rather than an inherent structural flaw.
For instance, autoregressive models require multi-pass inference that naturally exposes more logits, and diffusion models involve reconstruction objectives without a discriminative analogue.
Without aligning objectives or model complexity, the study may overstate the privacy gap between the two paradigms in realistic deployments.

**Questions:**

The finding that modeling P(X) inherently amplifies membership inference vulnerability is theoretically intriguing.
However, to make this result more practically valuable, could the authors discuss what broader implications or applications this insight might have?
For instance, how might this understanding inform the design of safer generative systems, the privacy evaluation of instruction-tuned LLMs, or trade-offs in multimodal models that integrate both generative and discriminative components?
In other words, beyond showing that generative classifiers are more vulnerable, what new directions or design principles can practitioners derive from this finding for building privacy-aware generative models?
If the authors could provide a more in-depth discussion or conclusion derived from this finding — for example, outlining its broader implications for the design of privacy-aware generative systems or its relevance to real-world applications such as instruction-tuned or multimodal models — I would significantly increase my overall score for this paper.

---

> ### Author Response · Authors · 2025-11-21
> **Response to Reviewer C54N (part 1)**
>
> We thank the reviewer for this careful and thought-provoking comment. We respond in two parts: (1) clarifying why the comparison between generative and discriminative classifiers is not a task mismatch but a *long-standing and principled distinction* in both classical statistical learning as well as modern deep learning, and (2) providing broader design implications and practical guidance that extend beyond the empirical observations in our paper.
>
> ---
>
> ## **1. On the claim of “task mismatch” between generative and discriminative classifiers**
>
> We agree that generative classifiers optimize a more complex objective i.e. modeling the joint distribution $P(X,Y)$, which inherently includes sequence-level likelihood and token-level prediction. However, this distinction is **not an artifact of our evaluation**; it is the *classical* difference between generative and discriminative paradigms. This comparison predates modern deep learning and is well established in the literature:
>
> - Efron (1975) and Ng & Jordan (2001) provide foundational generative–discriminative analyses.
> - In NLP, Yogatama et al. (2017), Li et al. (2025), and Kasa et al. (2025) revisit these differences in the context of modern deep learning and transformer-based classifiers.
>
> Generative classifiers (AR, DIFF, etc.) are explicitly proposed in prior work as viable alternatives to discriminative classifiers in text classification. Thus, comparing their privacy implications is directly aligned with the long-standing generative–discriminative debate.
>
> Importantly, the reviewer’s intuition that *generative models may leak more because they optimize a richer objective* is exactly the structural mechanism our theory formalizes. Modeling $P(X)$ necessarily introduces a marginal-memorization channel (Lemma 3.1), and exposing log-joint scores $\log P(x,y)$ preserves this signal (Lemma 3.2). Hence, the higher vulnerability is **not due to incidental implementation complexity**, but an intrinsic consequence of learning $P(X)$ rather than only $P(Y\mid X)$.
>
> This is also practical: recent work (e.g., Li et al. 2025; Kasa et al. 2025) highlights strong low-data advantages of generative classifiers. This is precisely the setting where our results matter most, because practitioners adopting generative classifiers for utility must also consider heightened privacy risks.
>
> ---
>
> ## **2. Broader implications and design takeaways for privacy-aware generative systems**
>
> Beyond quantifying a vulnerability gap, our work identifies *actionable structural insights* that inform the design of privacy-preserving generative classifiers.
>
> ### **(a) Runtime guidance: avoid exposing logits for generative classifiers**
>
> We show that exposing logits retains the joint-likelihood component, which significantly amplifies leakage.
> **Practical takeaway:**
> - Serve **probabilities (post-softmax)** rather than logits when deploying generative classifiers.
> - This reduces vulnerability without harming accuracy.
> - Necessary to enable calibration, hallucination detection, distillation (see our response to Reviewer AfpX - Part 1)
>
> ---
>
> ### **(b) Architectural design: factorization choices directly affect privacy**
>
> Section 5 demonstrates that different generative factorizations produce markedly different privacy behaviors:
>
> - **Diffusion (DIFF)** models are consistently *most vulnerable*.
> - **Autoregressive (AR)** models are vulnerable, with susceptibility **increasing with model size** (Appendix E).
> - **Pseudo-generative (MLM, P-AR)** models are substantially *less* vulnerable and often competitive in utility.
>
> **Practical takeaway:**
> - Use **P-AR** or **MLM** when low data or robustness motivates generative modeling.
>
> ---
>
> ### **(c) Model scaling: generative models show poor privacy–utility scaling**
>
> Appendix H shows:
>
> - Increasing AR model size greatly amplifies leakage, while utility increases modestly.
> - MLM models improve utility with scale while maintaining low vulnerability.
>
> **Takeaway:**
> - In privacy-sensitive deployments, prefer **larger MLM/DISC models** over large AR/DIFF models.
>
> ---
>
> ### **(d) Fairness/privacy interaction: generative classifiers reduce minority-class vulnerability gaps**
>
> Appendix D reveals:
>
> - DISC and MLM classifiers show **large disparities** between majority- and minority-class vulnerability.
> - Generative classifiers (AR/DIFF) show **minimal disparity**.
>
> **Implication:**
> - For applications prioritizing **privacy for minority groups**, generative classifiers may provide more equitable protection.
>
> ### **(e) Mitigations: clipping / temperature scaling offer modest reductions**
>
> While not eliminating structural leakage, compared to temperature scaling, techniques such as logit clipping do reduce the vulnerability of generative classifiers, but often degrade utility. We present empirical results illustrating this trade-off in the supplemental tables. (Representative results in Response to Reviewer AfpX (Part 3 and 4))
>
>
> ---
> (continued)

---

> ### Author Response · Authors · 2025-11-21
> **Response to Reviewer C54N (part 2)**
>
> ## **3. Outlook: implications for instruction-tuned LLMs and multimodal generative systems**
>
> Our findings generalize beyond text classification:
>
> - **Instruction-tuned LLMs** expose token log-probabilities; understanding the marginal channel is essential for evaluating fine-tuning privacy.
> - **Multimodal generative systems** (e.g., text–image diffusion models) involve learning rich joint distributions \(P(X)\), suggesting similar leakage mechanisms.
> - **Hybrid factorization strategies** (e.g., P-AR) could inspire privacy-aware designs that retain generative strengths while suppressing unnecessary marginal modeling.
>
> We will add this expanded discussion to the final version.
>
> ---
>
> ### **Closing Note**
>
> We thank the reviewer again for the insightful question. We hope this clarifies that our work not only identifies a structural vulnerability in generative classifiers but also provides concrete design principles, practical deployment recommendations, and forward-looking avenues for privacy-aware generative modeling. We would gratefully appreciate an increased score if the reviewer finds this clarification satisfactory, and we welcome any further questions.

---

> ### Author Response · Authors · 2025-11-28
> **Response to Reviewer C54N (part 3)**
>
> Dear Reviewer **C54N**,
>
> We hope you are doing well. We wanted to follow up respectfully during the discussion phase to ask whether you had any remaining questions for us.
>
> Since your last comment, we uploaded a substantially revised PDF with changes marked in **blue**. Below is a concise summary of the updates we made in direct response to your review.
>
> ---
>
> ## 1. Broader implications, recommendations and design principles are clarified in Section 5, 6 and Appendix H.2:
>
> - **Avoid exposing logits for generative classifiers**: we show why logits retain the joint-likelihood component and significantly amplify leakage.
> - **Prefer pseudo-generative (P-AR, MLM) classifiers** when privacy risk matters, since they model \(P(Y|X)\) more directly and leak far less.
> - **Architectural factorization choices** matter: DIFF > AR > MLM/DISC in privacy vulnerability.
> - **Model scaling guidance**: AR models worsen sharply with size; MLM scales more safely.
> - **Fairness implications**: generative classifiers exhibit smaller minority–majority leakage gaps.
>
>
> ## 2. Clarified the alleged “task mismatch”
>
> - The comparison follows the classical generative–discriminative framework (Efron 1975; Ng & Jordan 2001).
> - The evaluation tasks are identical across models; the only difference is modeling \(P(X)\).
> - Our decomposition theorem mathematically rules out task complexity as the source of leakage.
>
> ## 3. Expanded discussion of real-world relevance
> You mentioned instruction-tuned and multimodal systems. We now clarify in Section 6 and Appendix H.2:
> - Which APIs expose logits vs. probabilities.
> - How our findings apply to LLM-as-a-service deployment.
> - Why privacy-aware design choices for generative classifiers are important in real-world deployment.
>
> ---
>
> We would be happy to answer any remaining questions you have. If everything now looks clear and your concerns are resolved, we hope the improvements are reflected in your final evaluation.
>
> Thank you again for your feedback and for helping us strengthen the paper.
> **Authors of submission #19645**

---

### Official Review · Reviewer_rsZB · 2025-10-31

**Soundness:** 3
**Presentation:** 3
**Contribution:** 3
**Rating:** 6
**Confidence:** 2

**Summary:**

The paper studies membership inference attacks (MIAs) in text classifiers across discriminative (P(Y|X)), fully generative (autoregressive label-prefix and discrete diffusion modeling P(X,Y)), and pseudo-generative (MLM, label-suffix P-AR) paradigms. It gives a black-box theory showing MIA advantage decomposes into leakage from the marginal P(X) and the conditional P(Y|X). Exposing joint/logit scores retains P(X) signal and are not safer than exposing post-softmax probabilities. Synthetic experiments and large transformer experiments on nine datasets show fully generative models leak the most, with the largest risk when K-pass logit vectors are exposed. Pseudo-generative variants and encoders are safer at comparable utility. The paper concludes with practical guidance1. prefer probabilities over logits; 2. avoid DIFF in privacy-sensitive settings; 3. consider MLM / encoder baselines

**Strengths:**

I enjoy reading this paper. It first presents theoretical framework of decomposition of leakage into marginal vs conditional channels and then provide experiment results to demonstrate the hypothesis. Finally it provides guidence on API exposure. The presentation is clear.

**Weaknesses:**

1. Still, the theoretical results is based on several assumptions that may or may not hold in practice. For example, it assumes the data is i.i.d. and there is no pre-training.
2. I think the attack itself are well studied for generative models. It is also well known that richer outputs leak more is well known (labels < probabilities < logits). This paper is the first one that formulate this formally.

**Questions:**

Besides the guidance on the API exposures, is there any way to estimate the marginal terms and conditional terms from shadow models to predict MIA risk before deployment?

---

> ### Author Response · Authors · 2025-11-21
> **Response to Reviewer rsZB (Part 1)**
>
> Thanks for the detailed review and feedback on our paper. Below, we provide additional explanations that directly address your questions:
>
> **1. Still, the theoretical results is based on several assumptions that may or may not hold in practice. For example, it assumes the data is i.i.d. and there is no pre-training.**
>
>
>
> We thank the reviewer for raising this important concern. We agree that real-world deployments may involve non-i.i.d. data and extensive pre-training. However, our theoretical and experimental design intentionally isolates the *structural* differences between generative and discriminative classifiers by avoiding confounding factors that would obscure the comparison. We clarify below.
>
> ---
>
> ### **a). The i.i.d. assumption is standard and necessary for well-posed MIA evaluation**
>
> Nearly all foundational MIA studies—**Shokri et al. (2017)**, **Salem et al. (2018)**, **Yeom et al. (2018)**, **Sablayrolles et al., 2019** and many follow-ups—assume that train and non-train samples are drawn i.i.d. from a common population distribution. This is essential because:
>
> - otherwise an attacker could exploit *dataset shift* rather than *model-induced leakage*;
> - the resulting metric would measure distribution drift instead of membership privacy.
>
> Our theoretical decomposition (marginal vs. conditional leakage) relies on this standard assumption, and our empirical setup follows it to ensure the measured vulnerability reflects **model behavior**, not **covariate shift**.
>
> ---
>
> ### **b). Excluding pre-training is a deliberate, principled and well-established choice**
>
> Using pre-trained models would introduce substantial confounding factors, making it impossible to pinpoint whether leakage arises from:
>
> - different **pre-training corpus** (often billions of tokens),  or
> - the **downstream fine-tuning benchmark datasets**,
>
> Many benchmark datasets we evaluate on appear fully or partially, in public pre-training corpora, which means a pre-trained model could already have memorized parts of the dataset before fine-tuning. This makes downstream privacy comparisons uninterpretable. Prior works such as **Li et al. (2025)**, **Kasa et al. (2025)**, and **Yogatama et al. (2017)** therefore adopt a **from-scratch training protocol**, and we follow this established practice.
>
> ---
>
> ### **c). Pre-training blurs the generative/discriminative distinction**
>
> Modern pre-trained transformers (e.g., BERT, T5) use **hybrid objectives** that combine generative and discriminative elements. E.g. a “discriminative” classifier built from BERT is *not purely discriminative* as the pretraining objective includes Masked Language Modeling (MLM) + Next Sentence Prediction (NSP) tasks
>
> Including such models would confound the central question of our paper i.e. isolating the privacy implications of explicitly modeling $P(X)$ vs. $P(Y \mid X)$. Training from scratch is necessary to disentangle these effects.
>
> Our empirical work is designed to reveal these structural effects. Pre-training would obscure them with representational priors, corpus memorization, and unknown training dynamics. We will add this explanation in the revised draft.

---

> ### Author Response · Authors · 2025-11-21
> **Response to Reviewer rsZB (Part 2)**
>
> **2. I think the attack itself are well studied for generative models. It is also well known that richer outputs leak more is well known (labels < probabilities < logits). This paper is the first one that formulate this formally.**
>
>
> We thank the reviewer for the comment and for noting that our paper is the first to *formally* characterize the ordering between labels, probabilities, and logits. We clarify the contribution more precisely, as the statement that “the attack itself is well studied for generative models” does not align with existing literature.
>
> ---
>
> ### **a). Prior MIA work has focused almost exclusively on *discriminative* classifiers in the context of Text classification**
>
> Almost all prior MIA work in Text Classification evaluates:
> - **feed-forward discriminative classifiers**,
> - with **probability-vector outputs**,
> - using **loss/entropy/confidence/auxiliary model** attacks.
>
> To our knowledge, there is **no prior systematic study** of membership inference attacks on **generative classifiers**, neither in:
> - controlled settings,
> - transformer-based generative classifiers (AR, diffusion),
> - nor text classification with generative factorization.
>
> Our paper is the first to examine this rigorously across:
> - a controlled Gaussian toy model,
> - a general theoretical analysis applicable to any generative vs. discriminative model,
> - and nine benchmark NLP text classification datasets.
>
>
>
> ---
>
> ### **b). While the heuristic “richer outputs leak more” existed, it had not been *formalized***
>
> Practitioners often assumed logits > probabilities > labels in terms of vulnerability, but prior work did **not** establish:
>
> - *when* the ordering holds i.e. they do not sufficient conditions for this to hold,
> - or how **generative modeling of \(P(X)\)** amplifies leakage.
>
> Our paper provides, for the first time:
> 1. **Lemma 3.2:** a formal proof that logits (joint scores) dominate conditional probabilities via data-processing inequality.
> 2. **Theorem 3.3:** a regime (“systematic marginal skew”) where even a *single scalar joint score* strictly dominates conditional scores.
> 3. **Lemma 3.1:** a clean decomposition of membership advantage into **marginal** vs. **conditional** leakage components.
>
> These results do *not* appear in any previous MIA paper to the best of our knowledge.
> Thus, our contribution is not merely confirming an intuition; rather it explains **structurally** why generative classifiers leak more.
>
> ---

---

> ### Author Response · Authors · 2025-11-22
> **Response to Reviewer rsZB (Part 3)**
>
> **Additional results using pretrained models**
>
> We thank the reviewer for prodding us to explore the generalizability of the findings to pretrained models. To find out, we additionally evaluated using two standard pretrained models, specifically BERT-base-uncased and GPT-2 small, both approximately 110M parameters and released within a similar time frame, to ensure a fair comparison across paradigms. These pretrained models were fine-tuned on the classification task in standard discriminative (encoder) and generative (autoregressive) fashion as has been done in the paper.
>
> Even in this pretrained setting, we continue to observe that the generative autoregressive GPT-2 classifier remains substantially more susceptible to MIA attacks compared to the discriminative BERT encoder. Below we show representative results for the AG News dataset; we will include full results for all datasets in the revised version.
>
> The pretrained-model results reinforce our core finding: the heightened vulnerability of generative classifiers is not simply an artifact of training from scratch, but a structural consequence of jointly modeling the $P(X,Y)$ and exposing joint-likelihood signals. Pretraining does not eliminate this channel; in fact, GPT-2’s susceptibility remains consistently higher across all data sizes.
>
> **MIA Vulnerability as measured via GBM (logits) attack model through AUROC on AGnews dataset**
>
> | Training Size | BERT (mean (std)) | GPT-2 (mean (std)) |
> |:-------------:|:------------------:|:--------------------:|
> | 128    | 0.594 (0.086) | 0.827 (0.143) |
> | 256    | 0.579 (0.068) | 0.827 (0.148) |
> | 512    | 0.586 (0.040) | 0.917 (0.049) |
> | 1024   | 0.592 (0.080) | 0.830 (0.132) |
> | 2048   | 0.556 (0.038) | 0.773 (0.142) |
> | 4096   | 0.569 (0.064) | 0.798 (0.141) |
> | full_data | 0.551 (0.045) | 0.797 (0.179) |

---

> ### Author Response · Authors · 2025-11-22
> **Response to Reviewer rsZB (Part 4)**
>
> **4, Besides the guidance on the API exposures, is there any way to estimate the marginal terms and conditional terms from shadow models to predict MIA risk before deployment?**
>
> We appreciate the reviewer’s insightful question. Our marginal–conditional decomposition in Lemma 3.1 is intended primarily as a conceptual tool that explains where leakage arises in generative versus discriminative classifiers, through the marginal $P(X)$ and the conditional $P(Y|X)$. While one could imagine estimating these KL terms from shadow models, doing so in practice is extremely challenging in modern NLP settings, especially for the marginal component. Estimating the KL terms for the marginal and conditionals requires forming expectations over the full space of input sentences, which is a high-dimensional discrete space.  Accurately approximating such divergences would require learning a high-fidelity generative model for
> $P(X)$, which is itself a difficult problem and would essentially replicate the cost, instability, and data assumptions of training the target generative classifier in the first place. Even small covariate shifts between the shadow dataset and the true training distribution can distort the estimated divergence and lead to misleading conclusions about the conditional leakage.  In general, the literature on density estimation and divergence estimation consistently treats KL estimation in high-dimensional structured domains as nontrivial and highly sensitive to model misspecification; thus, we avoid recommending it as a practical route for pre-deployment auditing.
>
>
> For these reasons, the KL-based decomposition in our theory is best interpreted as a *diagnostic framework* that explains why generative classifiers exhibit higher leakage, i.e. because they learn $P(X)$ in addition to $P(Y|X)$, rather than as a prescription for numerically estimating KL terms prior to deployment. In practice, the most reliable way to anticipate MIA vulnerability is to run the same black-box attacks we evaluate (entropy, max-probability, log-loss, GBM-based attacks) on a representative train/test split before deployment. We will clarify this distinction in the final version: the decomposition provides conceptual insight into which modeling choices introduce privacy risk, while direct empirical attacks provide the most robust and actionable means of quantifying that risk in deployed systems.

---

> > ### Comment · Reviewer_rsZB · 2025-11-26
> >
> > Thank authors for the response. My questions and concerns are fully addressed. To clarify, my comments about the i.i.d. and no-pretraining assumptions were not meant to suggest these choices were incorrect, but rather to note that they are common constraints across studies in this area. Finally, the explanation that KL-based estimation is impractical resolves my questions.

---

> > > ### Author Response · Authors · 2025-11-28
> > > **Thanks for confirming your concerns have been addressed**
> > >
> > > Dear Reviewer **rsZB**,
> > >
> > > Thank you again for your thoughtful feedback and prompt confirmation. We just wanted to inform you that the suggested changes have been incorporated in the revised PDF (with all changes marked in **blue**).
> > > If there is anything else you would like us to elaborate on, we would be very happy to do so.
> > >
> > > Otherwise, we would appreciate your updated evaluation on the paper, especially now that your concerns have been resolved. Your perspective would be very valuable for the final decision.
> > >
> > > Thank you again for your time and engagement.
> > >
> > > **Authors of submission #19645**

---

### Official Review · Reviewer_AfpX · 2025-11-03

**Soundness:** 3
**Presentation:** 2
**Contribution:** 3
**Rating:** 4
**Confidence:** 3

**Summary:**

The paper investigates the vulnerability of generative text classifiers to MIAs, showing that they tend to leak more information than discriminative classifiers. It develops a decomposition that categorizes membership leakage into marginal and conditional terms, revealing that logits can dominate probabilities in attack power. The hypothesis is validated through a toy study, which demonstrates higher attack AUROC for joint likelihood signals, particularly in high-dimensional and low-sample regimes. Extensive experiments on nine text-classification datasets and five model paradigms further show that generative models are consistently more vulnerable.

**Strengths:**

This study addresses an important and interesting aspect of the MIA field, showing that generative classifiers are more vulnerable than discriminative classifiers. It provides a framework and theoretical foundation to guide the practical use of generative classifiers, especially when security and privacy are concerns. The findings suggest that pseudo-generative models may offer a better privacy–utility balance for sensitive deployments.

The theoretical component of the framework is solid, and the experiments and analysis are comprehensive. The paper itself is well structured and clearly presented.

**Weaknesses:**

Despite the strong theoretical basis, the practical value of this study appears somewhat limited, as public or production models typically do not expose logits or probabilities. This restriction narrows the external applicability of the results. Moreover, attackers in real-world settings could control decoding temperature, perform seed manipulation, or access internal signals; thus, a deeper discussion of white or gray box attacks would add practical relevance.

The comparison setup may also introduce bias. The paper states that “one forward pass per label
$y_i$ is used to score $log P(x,y_i)$, whereas discriminative models compute $P(Y∣X)$ in a single pass.” This difference in computation could affect both the attack surface and compute budget, potentially disadvantaging generative models. It would be useful to examine whether, under the same computational budget, generative classifiers remain more vulnerable.

In terms of defense mechanisms, could temperature scaling or logit clipping help narrow the logits–probabilities gap, thereby reducing the vulnerability of generative models? An empirical evaluation of such mitigations would strengthen the study.

The paper says that “scalar joint can dominate conditional under systematic marginal skew.” Does this assumption always hold? Could certain types of real-world data invalidate it? A more detailed analysis of when this dominance applies would strengthen the theoretical insights.

The study focuses on text classification. Would the proposed framework be extendable to non-TC settings? It would be valuable to clarify which parts of the theory generalize directly and which require adjustment.

Attack success is measured in AUROC.  How about other measures, e.g. Advantage score, TPR@low-FPR, or PPV? especially when minority classes are involved.

The study measures attack success primarily using AUROC. Additional metrics such as advantage score, TPR@low-FPR, or PPV, could provide a more comprehensive assessment of real-world risk, especially in minority-class scenarios.

Should Table 1 be transposed so that the vulnerability of the models can be compared more clearly across categories?

In Figure 2, AR should appear in the caption to ensure consistency with the figure legend. Perhaps replace GEN -> AR?

Figure 1 is not mentioned in the main text. The statement “Figure 26 reveals …” would be clearer if it explicitly noted that the figure is located in the appendix.

**Questions:**

See above.

---

> ### Author Response · Authors · 2025-11-18
> **Response to Reviewer AfpX  (Part 1)**
>
> Thanks for the detailed review and feedback on our paper. Below, we provide additional explanations and experiments that directly address your questions:
>
> **1.** **Despite the strong theoretical basis, the practical value of this study appears limited, as public/production models typically do not expose logits or probabilities. Attackers may also control temperature, seeds, or access internal signals; a deeper discussion of white/gray-box attacks would add relevance.**
>
> We clarify that several widely used commercial and open-source systems *do* expose token-level likelihoods:
>
> - **Commercial APIs:**
>   - OpenAI provides top-k logprobs for GPT-4o.
>   - Google Gemini exposes logprobs via Vertex AI pipelines.
> - **Open-source inference stacks:** Toolkits such as **TGI** support returning logprobs, which are not just for completeness sake but also essential for:
>   1. Probability calibration in retrieval and ranking pipelines
>   2. Hallucination detection methods using likelihood signals [Moslonka et al.]
>   3. Distillation from larger to smaller models.
>
> Therefore, even partial visibility into token-level likelihoods remains realistic for black-box threat models.
>
> Regarding **black-box vs. white-box MIA**, prior theoretical and empirical work shows that white-box access offers *limited additional advantage*:
> - [Sablayrolles et al.] show near-equivalence of white-box and black-box optimal strategies.
> - [Salem et al.] and [Huang et al.] observe only marginal empirical improvements.
> - Additionally, production LLM internals are typically inaccessible, reinforcing the relevance of the black-box setting.
>
> Finally, the canonical MIA literature [Shokri et al.; Truex et al.] explicitly assumes probability-vector access from ML-as-a-Service providers, aligning our work with established threat models.
>
> [1] *Membership Inference Attacks Against Machine Learning Models*, Shokri et al., 2017
> [2] *Towards Demystifying Membership Inference Attacks*, Truex et al., 2018
> [3] *White-box vs Black-box: Bayes Optimal Strategies for Membership Inference*, Sablayrolles et al., 2019
> [4] *ML-Leaks: Model and Data Independent Membership Inference Attacks*, Salem et al., 2018
> [5] *Privacy Evaluation Benchmarks for NLP Models*, Huang et al., 2024
> [6] *Learned Hallucination Detection in Black-Box LLMs using Token-level Entropy Production Rate*, Moslonka et al., 2025
>
> **2.** **The paper says that “scalar joint can dominate conditional under systematic marginal skew.” Does this assumption always hold? Could certain types of real-world data invalidate it? A more detailed analysis of when this dominance applies would strengthen the theoretical insights.**
>
> We appreciate the reviewer's request for clarification. Importantly, we do not claim that scalar joint scores always dominate conditional scores. **Theorem 3.3 provides sufficient (not necessary) conditions** for dominance, specifically that the marginal skew $KL_X$ must exceed the conditional skew ${KL}_{Y \\mid X}$ by a factor determined by the bounded likelihood-ratio condition. When this inequality does not hold, the dominance can reverse.
>
> Our empirical results directly illustrate such counter-examples. In the toy LDA vs. logistic regression experiments (Fig. 1 / Fig. 26), for low class separation $\\mu$, moderate sample size $n = 200$, and higher dimensions d $\in$ {64, 256}, the LDA log-joint signal is *less* vulnerable than the LDA posterior-probability signal (green curve lying below orange). This happens because in this regime LDA learns the conditional decision boundary $P(Y \\mid X)$ reasonably well, while the marginal density $P(X)$ is poorly estimated. This is exactly the situation where the premise
> $$c(\\alpha, \\beta)KL_X > KL_{Y\\mid X}$$
> fails. As expected from our theory, the joint score does **not** dominate in this regime.
>
> We will add a clarifying note explaining this behavior: **the theorem characterizes when dominance *can* occur, not when it *must***. Our experiments illustrate both the positive case (strong marginal memorization) and the negative case (weak or noisy marginal modeling), strengthening the connection between the theoretical conditions and practical scenarios.

---

> ### Author Response · Authors · 2025-11-18
> **Response to Reviewer AfpX (Part 2)**
>
> **3.** **The study focuses on text classification. Would the proposed framework be extendable to non-TC settings? It would be valuable to clarify which parts of the theory generalize directly and which require adjustment.**
>
> We thank the reviewer for raising this question. Our theoretical framework is fully general and not specific to text classification. The decomposition in Lemma 3.1 and the dominance results in Lemmas 3.2-3.3 apply to any generative vs. discriminative classifier pair because they only assume access to model-induced score distributions, independent of the modality or architecture (see Sec. 3). The notions of marginal leakage through $P(X)$ and conditional leakage through $P(Y \mid X)$ arise from the factorization of the joint distribution and therefore extend directly to images, audio, tabular data, or any supervised domain.
>
> Our experiments focus on text classification solely for practical, not theoretical, reasons. As highlighted in prior work (e.g., Sec. 4.2 in Li et al.), applying fully generative classifiers such as diffusion models to non-text modalities typically requires training a separate generative model for each class to estimate $P(X \mid Y = y)$. This makes it infeasible to conduct controlled comparisons across datasets with varying numbers of classes, and it is especially problematic for datasets with many classes or imbalanced label distributions, where some classes may not have enough samples to train class-specific generative models of equal size.
>
> In contrast, text classification uniquely allows training a single shared generative model over the joint distribution $P(X,Y)$ by simply appending the label token during training. This has been the canonical approach as established in prior text classification work (Yogatama et al., Li et al., Kasa et al.) and used extensively in our experiments. Because the label is part of the sequence, modeling $P(X,Y)$ does not require training $K$ (= \#labels) separate generative models, making comparisons across architectures, datasets, and model sizes tractable and fair.
>
> We will clarify this distinction in the final version: the theory is modality-agnostic and applies broadly, while text classification offers a uniquely practical setting for systematic empirical evaluation due to its ability to support class-conditional generative modeling with a single unified model. Thus, we chose to do comprehensive testing on text classification datasets, and other modalities are out of scope for this paper.
>
> [1] *Generative and discriminative text classification with recurrent neural networks*, Yogatma et al. (2017)
> [2] *Generative Classifiers Avoid Shortcut Solutions*, Li et al. (2025)
> [3] *Generative or Discriminative? Revisiting Text Classification in the Era of Transformers*, Kasa et al. (2025)

---

> ### Author Response · Authors · 2025-11-20
> **Response to Reviewer AfpX (Part 3)**
>
> **4. In terms of defense mechanisms, could temperature scaling or logit clipping help narrow the logits–probabilities gap, thereby reducing the vulnerability of generative models? An empirical evaluation of such mitigations would strengthen the study.**
>
> We thank the reviewer for the thoughtful suggestion to evaluate whether simple post-processing techniques, such as logit clipping or temperature scaling, can reduce the vulnerability of the classifiers, specifically the generative ones. In response, we conducted targeted experiments applying both defenses. Below, we report representative results on the AG News dataset using a 12-layer model trained with 4096 samples. The complete results will be incorporated into the revised PDF.
>
> ****Effect of Logit-clipping on Discriminative (Encoder) Classifier for various attacks****
>
>
> | Clipping | Entropy | GBM (Logits) | GBM (Probits) | Ground Truth Predictions | Log Loss | Max Probability | F1 Score |
> |:--------:|:--------:|:------------:|:--------------:|:------------------------:|:--------:|:----------------:|:--------:|
> | 0.01     | 0.568    | 0.660        | 0.661          | 0.616                    | 0.616    | 0.561            | 0.826    |
> | 0.025    | 0.568    | 0.659        | 0.662          | 0.616                    | 0.615    | 0.561            | 0.826    |
> | 0.05     | 0.566    | 0.658        | 0.663          | 0.614                    | 0.613    | 0.558            | 0.826    |
> | 0.1      | 0.565    | 0.660        | 0.662          | 0.612                    | 0.612    | 0.557            | 0.826    |
> | 0.2      | 0.567    | 0.658        | 0.659          | 0.609                    | 0.609    | 0.553            | 0.826    |
>
>
> ****Effect of Logit-clipping on Generative (Auto-Regressive) Classifier for various attacks****
>
>
> | Clipping | Entropy | GBM (Logits) | GBM (Probits ) | Ground Truth Predictions | Log Loss | Max Probability | F1 Score |
> |:--------:|:--------:|:-------------------------------:|:--------------------------------:|:------------------------:|:--------:|:----------------:|:--------:|
> | 0.01     | 0.734    | 0.999                           | 0.888                            | 0.793                    | 0.794    | 0.774            | 0.847    |
> | 0.025    | 0.711    | 0.995                           | 0.872                            | 0.769                    | 0.769    | 0.745            | 0.832    |
> | 0.05     | 0.666    | 0.995                           | 0.851                            | 0.725                    | 0.726    | 0.694            | 0.807    |
> | 0.1      | 0.574    | 0.981                          | 0.806                            | 0.634                    | 0.635    | 0.589            | 0.754    |
> | 0.2      | 0.421    | 0.950                           | 0.752                            | 0.477                    | 0.478    | 0.421            | 0.626    |
>
> The clipping value in the above tables denotes the percentile threshold used to clip logits, computed from the empirical logit distribution over the entire evaluation population. For example, a clipping value of 0.01 means that logits above the 99th percentile and below the 1st percentile, are replaced with their corresponding thresholded values. This post-processing reduces the dynamic range of logits without altering their ordering.
>
> We observe that clipping has negligible effect on discriminative classifiers: both privacy vulnerability (AUROC) and utility (F1) remain essentially unchanged across clipping levels. In contrast, clipping reduces the vulnerability of fully generative classifiers, most noticeably for logit-based attacks. However, this comes at a clear cost to utility, as the F1 score deteriorates steadily with stronger clipping. This illustrates that while clipping can shrink the vulnerability of generative classifiers, it does not address the underlying structural leakage from modeling $P(X)$, and comes at the cost of utility degradation.
>
>
> **(temperature-scaling analysis continued in the next response)**

---

> ### Author Response · Authors · 2025-11-20
> **Response to Reviewer AfpX (Part 4)**
>
> **(continued from Response - Part 3)**
>
> ****Effect of Temperature-scaling on Discriminative (Encoder) Classifier for various attacks****
>
> | Temperature | Entropy | GBM (Logits) | GBM (Probs) | Ground Truth Predictions | Log Loss | Max Probability |
> |:-----------:|:--------:|:-------------------------------:|:------------------------------:|:------------------------:|:--------:|:----------------:|
> | 0.1         | 0.554    | 0.661                           | 0.658                          | 0.611                    | 0.611    | 0.553            |
> | 0.5         | 0.558    | 0.661                           | 0.657                          | 0.613                    | 0.613    | 0.556            |
> | 1           | 0.568    | 0.661                           | 0.662                          | 0.616                    | 0.616    | 0.561            |
> | 2           | 0.577    | 0.661                           | 0.663                          | 0.620                    | 0.619    | 0.566            |
> | 10          | 0.541    | 0.661                           | 0.663                          | 0.625                    | 0.624    | 0.569            |
>
> ****Effect of Temperature-scaling on Generative (Auto-Regressive) Classifier for various attacks****
>
> | Temperature | Entropy | GBM (Logits)  | GBM (Probs) | Ground Truth Predictions | Log Loss | Max Probability |
> |:-----------:|:--------:|:-------------------------------:|:------------------------------:|:------------------------:|:--------:|:----------------:|
> | 0.1         | 0.794    | 0.895                           | 0.885                          | 0.818                    | 0.818    | 0.803            |
> | 0.5         | 0.751    | 0.895                           | 0.894                          | 0.806                    | 0.807    | 0.788            |
> | 1           | 0.742    | 0.895                           | 0.894                          | 0.803                    | 0.804    | 0.785            |
> | 2           | 0.737    | 0.895                           | 0.895                          | 0.802                    | 0.802    | 0.783            |
> | 10          | 0.732    | 0.895                           | 0.895                          | 0.801                    | 0.801    | 0.782            |
>
> We observe that temperature scaling is less effective than logit clipping at reducing the vulnerability of generative classifiers; however, unlike clipping, temperature scaling does not degrade utility, as the F1 score remains stable across temperatures. Note that the susceptibility for GBM (Logits) is unchanged under temperature scaling, since temperature rescales logits by a constant factor and therefore constitutes only a linear transformation that preserves their separability.

---

> ### Author Response · Authors · 2025-11-20
> **Response to Reviewer AfpX (Part 5)**
>
> **5. Attack success is measured in AUROC. How about other measures, e.g. Advantage score, TPR@low-FPR, or PPV? especially when minority classes are involved.**
>
> We agree with the reviewer's suggestion to regarding having complementary evaluation metrics beyond AUROC. In response, we additionally report TPR@FPR = 0.1, following the reviewer’s suggestion, and will include these complementary metrics alongside AUROC in the revised version. Below we show representative results for AG News, using a 12-layer model, comparing a Discriminative (encoder) classifier with a Generative (autoregressive) classifier. Consistent with our AUROC findings, we again observe that generative classifiers exhibit higher susceptibility across all attack types, even under stricter low-FPR operating points.
>
> **Discriminative (encoder) classifier - TPR@FPR = 0.1**
>
> | Training Size | Entropy | GBM (Logits) | GBM (Probs) | Ground Truth Predictions | Log Loss | Max Probability |
> |:-------------:|:--------:|:-------------------------------:|:-------------------------------:|:------------------------:|:--------:|:----------------:|
> | 128  | 0.047 | 0.621 | 0.638 | 0.414 | 0.383 | 0.375 |
> | 256  | 0.078 | 0.226 | 0.232 | 0.215 | 0.297 | 0.148 |
> | 512  | 0.168 | 0.313 | 0.314 | 0.256 | 0.277 | 0.258 |
> | 1024 | 0.227 | 0.272 | 0.275 | 0.227 | 0.343 | 0.223 |
> | 2048 | 0.191 | 0.162 | 0.181 | 0.167 | 0.245 | 0.123 |
> | 4096 | 0.153 | 0.157 | 0.160 | 0.140 | 0.306 | 0.140 |
>
>
>  **Generative (auto-regressive) classifier - TPR@FPR = 0.1**
>
> | Training Size | Entropy | GBM (Logits) | GBM (Probs) | Ground Truth Predictions | Log Loss | Max Probability |
> |:-------------:|:--------:|:-------------------------------:|:------------------------------:|:------------------------:|:--------:|:----------------:|
> | 128  | 0.539 | 0.991 | 0.797 | 0.805 | 0.719 | 0.656 |
> | 256  | 0.473 | 0.993 | 0.886 | 0.738 | 0.797 | 0.543 |
> | 512  | 0.730 | 0.995 | 0.951 | 0.844 | 0.881 | 0.811 |
> | 1024 | 0.628 | 0.995 | 0.914 | 0.559 | 0.795 | 0.538 |
> | 2048 | 0.613 | 0.997 | 0.866 | 0.533 | 0.748 | 0.502 |
> | 4096 | 0.480 | 0.995 | 0.636 | 0.306 | 0.605 | 0.286 |

---

> ### Author Response · Authors · 2025-11-21
> **Response to Reviewer AfpX (Part 6)**
>
> **6. The comparison setup may also introduce bias. The paper states that “one forward pass per label
>  is used to score, whereas discriminative models compute in a single pass.” This difference in computation could affect both the attack surface and compute budget, potentially disadvantaging generative models. It would be useful to examine whether, under the same computational budget, generative classifiers remain more vulnerable.**
>
> We thank the reviewer for raising this important concern. To directly test whether the difference in inference cost biases our comparison, we conduct an additional experiment where we fix the number of inference (forward) calls for both discriminative and generative classifiers, and evaluate MIA performance as a function of this compute budget.
>
> Below, we report results for a representative setting: AG News, 12-layer models, trained on 4096 samples. For both discriminative (encoder) and generative (autoregressive) models, we vary the number of inference passes (n_infer) up to 4096. We report mean AUROC with standard deviations in parentheses.
>
> Our key finding is that even at the lowest compute budget (128 inference calls), the generative classifier already shows substantially higher MIA vulnerability than the discriminative classifier (even when compared to at any compute level).
>
> These results indicate that the higher susceptibility of generative classifiers is not an artifact of higher inference cost, it reflects a structural privacy disadvantage stemming from modeling the joint likelihood $P(X,Y)$, not differences in compute.
>
> **Discriminative (Encoder) Classifier**
>
> | n_infer | Entropy | GBM (Logits) | GBM (Probs) | Ground Truth Predictions  | Log Loss  | Max Probability  |
> |:-------:|:--------------:|:-------------------------:|:---------------------------:|:--------------------------------:|:----------------:|:------------------------:|
> | 128  | 0.531 (0.027) | 0.559 (0.050) | 0.545 (0.050) | 0.530 (0.031) | 0.530 (0.031) | 0.530 (0.028) |
> | 256  | 0.530 (0.025) | 0.541 (0.039) | 0.541 (0.045) | 0.535 (0.027) | 0.535 (0.027) | 0.530 (0.025) |
> | 512  | 0.529 (0.018) | 0.543 (0.016) | 0.534 (0.010) | 0.535 (0.020) | 0.535 (0.020) | 0.529 (0.018) |
> | 1024 | 0.521 (0.010) | 0.534 (0.022) | 0.530 (0.012) | 0.528 (0.009) | 0.528 (0.009) | 0.521 (0.010) |
> | 2048 | 0.520 (0.006) | 0.528 (0.022) | 0.525 (0.015) | 0.526 (0.006) | 0.526 (0.006) | 0.520 (0.006) |
> | 4096 | 0.516 (0.004) | 0.526 (0.021) | 0.523 (0.014) | 0.522 (0.004) | 0.522 (0.004) | 0.516 (0.004) |
>
>
> **Generative (Autoregressive) Classifier**
>
> | n_infer | Entropy | GBM (Logits) | GBM (Probs) | Ground Truth Predictions  | Log Loss  | Max Probability |
> |:-------:|:--------------:|:-------------------------:|:---------------------------:|:--------------------------------:|:----------------:|:------------------------:|
> | 128  | 0.582 (0.040) | 0.581 (0.060) | 0.566 (0.010) | 0.576 (0.029) | 0.576 (0.029) | 0.577 (0.039) |
> | 256  | 0.578 (0.059) | 0.582 (0.056) | 0.657 (0.096) | 0.586 (0.055) | 0.586 (0.055) | 0.584 (0.053) |
> | 512  | 0.578 (0.027) | 0.571 (0.016) | 0.620 (0.095) | 0.601 (0.030) | 0.601 (0.030) | 0.585 (0.030) |
> | 1024 | 0.576 (0.022) | 0.564 (0.019) | 0.618 (0.081) | 0.595 (0.026) | 0.595 (0.026) | 0.584 (0.024) |
> | 2048 | 0.584 (0.017) | 0.572 (0.023) | 0.602 (0.080) | 0.596 (0.014) | 0.596 (0.014) | 0.587 (0.014) |
> | 4096 | 0.584 (0.011) | 0.579 (0.027) | 0.604 (0.072) | 0.596 (0.012) | 0.596 (0.012) | 0.586 (0.012) |

---

> ### Author Response · Authors · 2025-11-28
> **Response to Reviewer AfpX (part 7)**
>
> Dear Reviewer **AfpX**,
>
> We hope you are doing well, and we wanted to follow up to check whether you had any remaining questions regarding our revision. We greatly appreciate the time you took to read and engage with our work.
>
> Since your last comment, we uploaded a revised PDF with all changes marked in **blue**. Below is a short summary of the updates directly addressing your feedback:
>
> ---
>
> - A detailed explanation in Section 5 (blue text) showing that the privacy gap persists even under equalized compute, confirming that the vulnerability difference is structural and not caused by the *K*-pass scoring.
> - Temperature scaling and logit clipping evaluations (Section 5 and Table 12).
> - Additional TPR @ 0.1 FPR evaluation (Section 5 and Table 11)
> - Clarified how the logits/probabilities are typically exposed (section 6) in several real world deployments and the scope of white-box vs black-box attacks (section 2)
> - Clarified how the theoretical results are valid in non-TC settings as well (section 3).
> - Added discussion on when the scalar joint doesn't dominate marginal skew in Appendix G4.
> - Fixed other typos and minor presentation issues.
>
> ---
>
> We would be grateful to know if you have any further concerns or if any aspect of the revision remains unclear. If everything looks satisfactory, we hope the improvements are reflected in your final assessment.
>
> Thank you again for your time and constructive engagement.
> **Authors of submission #19645**

---

### Author Response · Authors · 2025-12-02
**Summary of Reviews and Our Rebuttals - Part 1**

Dear ACs/SACs,

Firstly, we express our sincere appreciation for the actions taken to preserve the scientific standards of ICLR.

We would like to note that none of the reviewers identified any major technical flaws or fundamental errors in our theoretical framework, empirical methodology, or conclusions. The concerns raised were predominantly about clarifications, additional analysis, or expanded discussion; we addressed all these concerns through follow-up experiments and detailed responses during the discussion period and ensured that the same are reflected in the updated draft. Despite our rebuttals and multiple follow-ups, we could get only one (**Reviewer rsZB**) out of the four reviews to respond to us, who has confirmed that all their concerns were resolved.

**Kindly note that we are highly suspicious that Review by **Reviewer C54N** is an LLM generated review and have a submitted ample evidence for the same via official comments multiple times to get the attention of ACs and SACs. However, in order to engage with the reviewer in good and full faith, we did post our rebuttal and follow-ups, but we could not get any response. We request the ACs/SACs to please look into this concerning matter.**

Summary of our work - Our paper analyzes MIAs in generative vs. discriminative classifiers through a theoretical marginal–conditional decomposition, a toy setup to demonstrate this phenomenon in controlled settings, and large-scale experiments (~2900 trained models) involving standard benchmark datasets showing structurally higher leakage for models that explicitly model $P(X)$.

Below, we summarize reviewer concerns in ascending score order, organized into two response sections.

---


## Reviewer AfpX (rating: 4) - First rebuttal on Nov 11th , Follow-up trying to engage on Nov 27th - Despite this, the reviewer has not engaged with us.

Reviewer AfpX raised concerns about (i) practical relevance of black-box access vs white/grey-box access, (ii) fairness of the comparison given K-pass inference for generative models, (iii) whether logit clipping or temperature scaling could mitigate the vulnerability gap, (iv) whether our theorem’s conditions always hold, (v) generality beyond text classification, and (vi) the use of metrics beyond AUROC.

In response, we clarified the practicality of black-box settings, added additional experiments to show the compute fairness, evaluated clipping/temperature scaling defenses, identified cases where our theorem’s conditions do not hold, explained the generality of our theory beyond text, and added additional TPR@FPR analyses. All these together address the reviewer’s concerns and strengthen the paper.

---

## Reviewer C54N (rating: 4) - First rebuttal on Nov 20th , Follow-up trying to engage on Nov 27th - Despite this, the reviewer has not engaged with us. Please refer to the earlier messages on why we call out this as a fully LLM-generated review.



Reviewer C54N raised two main concerns: (i) whether the privacy gap we observe is simply due to a “task mismatch” between generative and discriminative models, and (ii) whether our findings offer broader design insights beyond the observation that generative models are more vulnerable.

We clarified that the perceived “task mismatch” is not something introduced or coined by our work but reflects the long-established generative–discriminative classifier distinction studied for at least five decades: generative classifiers learn $(P(X,Y)$ while discriminative models learn $(P(Y\mid X)$. Our contribution is to show that this classical difference has direct privacy consequences i.e. modeling $P(X)$ introduces a marginal leakage pathway that structurally increases MIA vulnerability.

We find that comment about 'discuss what broader implications or offer design insights' a bit vague and contradictory as In the strengths section, the reviewer explicitly credit the paper with providing actionable recommendations for practitioners and ML-as-a-service providers. Yet later, they ask us to explain the “practical value” of our findings as if such content were missing. Nevertheless, we exhaustively list all the practical take-aways for practitioners in this paper, giving the specific sections where these are mentioned.

**(continued in part 2)**

---

### Author Response · Authors · 2025-12-02
**Summary of Reviews and Our Rebuttals - Part 2**

(continuation from part 1)

## Reviewer rsZB (rating: 6)

Reviewer rsZB raised two main concerns: (i) the theoretical assumptions of i.i.d. data and training models from scratch, and (ii) whether the marginal and conditional KL terms in our decomposition could be estimated using shadow models to predict MIA risk before deployment.

We addressed both concerns as follows:

- **i.i.d. and no-pretraining assumptions:** We clarified that the i.i.d. assumption is standard in MIA research because it isolates *model-induced* leakage rather than trivial distribution shift. We also train from scratch to avoid contamination from pretrained corpora (which may include benchmark datasets) and to prevent hybrid objectives (e.g., MLM + NSP) from obscuring the generative vs. discriminative comparison. This approach is consistent with prior generative–discriminative studies (Yogatama et al., Li et al., Kasa et al.).  To further address concerns about pretraining, we added experiments using **pretrained BERT-base-uncased vs. GPT-2 small (~110M each)** on AG News. Even in this pretrained setting, GPT-2 (generative AR) remains systematically more vulnerable than BERT (discriminative), indicating that our structural findings are robust to pretraining.

- **Estimating KL terms from shadow models:** We explained that the KL decomposition is intended as a *conceptual* explanation, not as a quantity for practitioners to estimate directly. Approximating $KL_X$ or $\mathrm{KL}_{Y\mid X}$ from shadow models would require high-fidelity generative modeling of the text distribution and assumes the attacker can sample shadow data from the same population, which is both unrealistic and nearly as expensive as training the target model. Instead, the practical way to assess risk is to run the black-box attacks themselves on a held-out audit split (train = members, test = non-members), which is exactly what we do empirically.



These points clarify that the theoretical decomposition guides *why* certain models leak more, while *direct empirical MIAs* are the practical tool for estimating risk before deployment. **Reviewer rsZB confirmed that all their concerns have been resolved**


----

## Reviewer 8KoX (rating: 6) - First rebuttal on Nov 20th , Follow-up trying to engage on Nov 27th

Reviewer 8KoX raised two main points: (i) the observation that generative models leak more information may be intuitively expected, and (ii) in practice, SOTA MIAs often use multiple shadow models, how does this relate to our single-shadow theoretical assumption?

We addressed these concerns as follows:

- **On the “expected” vulnerability of generative models:** We acknowledged the intuition but emphasized that our contribution is to **formalize** this phenomenon through a clean decomposition of MIA risk into marginal and conditional components, and to **demonstrate empirically** that this structure consistently appears across modern transformer classifiers. Our analysis also distinguishes between *different types* of generative architectures such DIFF, AR, P-AR, MLM, showing that vulnerability varies substantially across them, with DIFF ≫ AR > MLM ≈ DISC. This nuance goes well beyond the simplistic observation that “generative models output more information.”

- **On shadow models:** We clarified that the **single-shadow assumption** is used only for the theoretical development to define a reference distribution \(Q\). Using multiple shadows would approximate the same population distribution and does not affect the bounds. Empirically, we do **not** use shadow models at all. Our evaluation relies on **zero-shadow attacks** (direct train vs. test separation), which is standard in modern MIA studies and often more transparent than attack models trained on shadows. Prior work has repeatedly shown that shadow-free attacks can perform comparably to multi-shadow ones under realistic assumptions, so we do not expect multiple-shadow setups to qualitatively change our conclusions.

These clarifications reinforce that both our theoretical framework and empirical methodology align with established MIA practice, and that our findings are robust to the use or absence of shadow models.

We respectfully ask the AC and SACs to consider both the original reviews and our detailed, experiment-backed responses when forming the meta-review and final decision, noting that we have thoroughly addressed the concrete technical concerns raised by Reviewers AfpX and 8KoX and clarified the conceptual points raised by Reviewer C54N (whose review is highly likely LLM-generated, but under benefit of doubt gave rebuttal in good faith). We remain happy to clarify any further points if needed.

Best Regards,

---

### Meta-Review · Area_Chair_fNZU · 2025-12-27

**Summary:**

Overall, there were no clear and technical supporting details/evidence to justify accepting this paper, especially from the reviewers of score 6. Since I do not see convincing reasons to accept this paper, I recommend rejecting the paper.

**Reviewer Concerns:**

* Reviewer AfpX: For point 3, the authors provided a somewhat conflicting/mixed response - although they claimed their framework is fully general and not specific to text classification, but later, they argued that applying it to non-text modalities is infeasible.

* Although Reviewer 8KoX and rsZB gave score 6, I could not find any significant and technical supporting details and justification that the paper has to be accepted. Also, both reviewers’ Confidence scores are 2. Hence, they were less weighed.

* On the other hand, I could not agree with the authors’ response to the two concerns of Reviewer 8KoX. For the first one, the authors’ standpoint was not clear, and for the second point regarding the insufficient number of shadow models, I agree with the reviewer, and the authors’ response did not fully address it.

* Review of Reviewer C54N was not weighed too much. The Question regarding the broad implications is too general and not technical enough to be a decision factor to accept/reject this paper.

* Having said that, one thing to note is that the Weakness raised by Reviewer C54N is consistent with the concern raised by Reviewer AfpX.

**Reviewer Scores:**

* Reviewer AfpX: I do not think the reviewer would have changed the score.

* Reviewer rsZB: I do not think the reviewer would have increased the score any further.

* Reviewer C54N: I do not think the reviewer would have changed the score.

* Reviewer 8KoX: I do not think the reviewer would have increased the score any further.

---

### Decision · Program_Chairs · 2026-01-26

Reject